# Maximum Entropy Heterogeneous-Agent Reinforcement Learning

**Jiarong Liu**[1][♯][*] **Yifan Zhong**[1,2][*]**, Siyi Hu**[3]**,**
**Haobo Fu**[4]**, Qiang Fu**[4]**, Xiaojun Chang**[3]**, Yaodong Yang**[1][†]
[1]Institute for AI, Peking University, [2]National Key Laboratory of General AI, BIGAI,
[3]University of Technology Sydney, [4]Tencent AI Lab

## Abstract

*Multi-agent reinforcement learning* (MARL) has been shown effective for cooperative games in recent years. However, existing state-of-the-art methods face challenges related to sample complexity, training instability, and the risk of converging to a suboptimal Nash Equilibrium. In this paper, we propose a unified framework for learning *stochastic* policies to resolve these issues. We embed cooperative MARL problems into probabilistic graphical models, from which we derive the maximum entropy (MaxEnt) objective for MARL. Based on the MaxEnt framework, we propose *Heterogeneous-Agent Soft Actor-Critic* (HASAC) algorithm. Theoretically, we prove the monotonic improvement and convergence to *quantal response equilibrium* (QRE) properties of HASAC. Furthermore, we generalize a unified template for MaxEnt algorithmic design named *Maximum Entropy Heterogeneous-Agent Mirror Learning* (MEHAML), which provides any induced method with the same guarantees as HASAC. We evaluate HASAC on six benchmarks: Bi-DexHands, Multi-Agent MuJoCo, StarCraft Multi-Agent Challenge, Google Research Football, Multi-Agent Particle Environment, and Light Aircraft Game. Results show that HASAC consistently outperforms strong baselines, exhibiting better sample efficiency, robustness, and sufficient exploration.

## 1 Introduction

Cooperative multi-agent reinforcement learning (MARL) is a complex problem characterized by the difficulty of coordinating individual agent policy improvements to enhance overall performance of the entire team. As a result, traditional independent learning methods in MARL often lead to poor convergence properties (30; 3). To alleviate these difficulties, the *centralized training decentralized execution* (CTDE) paradigm (5; 19) assumes that global states and teammates' actions and policies are accessible during the training phase. This approach leads to the development of competent multi-agent policy gradient algorithms (40; 37; 38; 43; 42) as well as value decomposition algorithms (27; 24; 41; 26; 33) . Furthermore, heterogeneous-agent mirror learning (HAML) (15) provides a template for rigorous algorithmic design, which guarantees any induced algorithm of monotonic improvement of joint objective and convergence to *Nash equilibrium* (NE) (22).

Despite the theoretical soundness of the HAML framework, HAML-derived algorithms suffer from two main challenges. First, these methods face challenges attributed to either sample complexity or training instability. On-policy algorithms, including HAPPO and HATRPO (13), require new sample data for each gradient step, which becomes very expensive as task complexity and agent numbers increase. Off-policy algorithms, on the other hand, observe training instability and hyperparameter sensitivity (45). Moreover, HAML-derived algorithms suffer from insufficient exploration, which can lead to suboptimal NE convergence. This is primarily due to the standard MARL objective they maximize, where there always exists a deterministic convergence solution (15; 28) and stochasticity is not inherently encouraged. Since the presence of multiple NEs is a frequently observed phenomenon in many multi-agent games, these methods can fail to explore sufficiently and prematurely converge to a suboptimal NE, as we will show in Section 3.2.

A possible solution to these challenges is to let agents learn *stochastic* behaviors in a sample-efficient way. Similar to the case of single-agent RL (7), in cooperative MARL problems, stochastic policies enable effective exploration of the reward landscape, mastery of multiple ways of performing the

---

[*]Equal contribution. [♯]Work done during internship at Institute for AI, Peking University. [†]Corresponding to
<yaodong.yang@pku.edu.cn>. See our page at https://sites.google.com/view/meharl.

task, and robustness when facing prediction errors (46). Unfortunately, while a number of methods have achieved great success in single-agent RL settings (46; 7; 9; 17), solving such stochastic policy learning problems in cooperative MARL is challenging. Existing CTDE methods offer no convergence guarantee for learning stochastic policies in general cases. Recently, FOP (44) applies maximum entropy framework to multi-agent settings via value decomposition. However, FOP only provides convergence to the global optimum when a task satisfies the Individual-Global-Optimal (IGO) assumption, which limits its applicability in general cooperative MARL problems.

In this paper, we propose the first theoretically-justified actor-critic framework for learning stochastic policies in cooperative MARL. Firstly, we model cooperative MARL as a probabilistic graphical inference problem (Figure 2), where stochastic policies arise as optimal solutions. Performing variational inference in this model leads us to derive the maximum entropy (MaxEnt) MARL objective. To maximize this objective, we introduce *heterogeneous-agent soft policy iteration* (HASPI) which ensures the properties of monotonic improvement and convergence to *quantal response equilibrium* (QRE), which is the solution concept corresponding to stochastic policies in game theory (20; 6). The key insight behind this theory is the *joint soft policy decomposition proposition*. Based on HASPI, we derive the *heterogeneous-agent soft actor-critic* (HASAC) algorithm. Furthermore, we generalize the HASPI procedure to the *Maximum Entropy Heterogeneous-Agent Mirror Learning* (MEHAML) template, which offers a unified solution to MaxEnt MARL problems and provides the same theoretical guarantees for *any* induced methods as HASAC. We test HASAC on six benchmarks: Multi-Agent MuJoCo (MAMuJoCo) (4), Bi-DexHands (2), StarCraft Multi-Agent Challenge (SMAC) (25), Google Research Football (GRF) (16), Multi-Agent Particle Environment (MPE) (19), and Light Aircraft Game (LAG) (23). Across the majority of benchmark tasks, HASAC consistently outperforms strong baselines, exhibiting the advantages of stochastic policies, namely improved robustness, higher sample efficiency, and sufficient exploration.

## 2 RELATED WORK

Multi-agent policy gradient (MAPG) methods have been shown effective for multi-agent cooperation tasks (5; 19). Yu et al. (42) discovers the effectiveness of PPO in multi-agent scenarios and introduces MAPPO. It inspires CoPPO (39), which preserves monotonic improvement property with a simultaneous update scheme, HAPPO / HATRPO (13), which proves monotonic improvement and NE convergence property with a sequential update scheme, and A2PO (34), which preserves per-agent monotonic improvement property. Guarantees of HAPPO / HATRPO are enhanced by HAML (15), which abstracts a general theoretical framework and leads to several practical algorithms. While these methods are effective on challenging benchmarks, we show in Section 3.2 that due to the standard objective they optimize, they tend to converge rapidly to a suboptimal NE when in proximity to it. Notably, the idea of NE can be considered as a notion of local optimum in cooperative MARL settings, and has been studied in many prior works (29; 36; 13; 15). To alleviate suboptimal NE convergence problem, we propose to learn stochastic policies, which maximize the MaxEnt MARL objective that we derive from probabilistic graphical models. We adopt QRE (20; 6; 11) as the solution concept in MaxEnt MARL framework, which generalizes NE when payoffs are perturbed by additional noise.

MaxEnt algorithms have achieved great success in single-agent RL. SQL (7) and SAC (8) learn optimal MaxEnt policies through soft Q-iteration and soft policy iteration respectively. They refresh SOTA performance, showcasing the robustness and effective exploration of stochastic policy. Levine (17) reviews these algorithms from a control as inference perspective. Unfortunately, learning MaxEnt policies with theoretical guarantees remains a challenge in cooperative MARL settings. MASQL (36) adopts a multi-agent actor-critic architecture similar to MADDPG (19), extending SQL to multi-agent settings without providing any theoretical guarantees. FOP (44), on the other hand, is a decomposed actor-critic method (4; 35), which utilizes the decomposed critic instead of the centralized critic to learn individual policies. It factorizes the optimal joint policy of MaxEnt MARL under the IGO assumption (see Appendix A.2 for details), which leads to poor performance in complex scenarios. To overcome the constraint of IGO, MACPF (32) learns optimal joint policy during training phase and distills independent policies from the optimal joint policy to fulfill decentralized execution. Such a procedure can be considered as offline imitating learning, where independent policies strive to mimic the behaviors produced by the optimal joint policy, but they still lack the guarantee of converging to the optimum. In contrast, our approach is the first MaxEnt actor-critic method with theoretical guarantee, presenting an improvement to HAML-based algorithms. We augment the objective with entropy, propose HASPI, prove its monotonic improvement and QRE (20; 6; 11) convergence property without restrictive assumption, derive HASAC, and establish MEHAML template.

## 3    PRELIMINARIES

### 3.1    COOPERATIVE MULTI-AGENT REINFORCEMENT LEARNING

We consider a cooperative Markov game (18) formulated by a tuple $\langle \mathcal{N}, \mathcal{S}, \boldsymbol{\mathcal{A}}, r, P, \gamma, d \rangle$. Here, $\mathcal{N} = \{1, \ldots, n\}$ denotes the set of $n$ agents, $\mathcal{S}$ is the finite state space, $\boldsymbol{\mathcal{A}} = \prod_{i=1}^{n} \mathcal{A}^i$ is the joint action space, where $\mathcal{A}^i$ denotes the finite action space of agent $i$, $r : \mathcal{S} \times \boldsymbol{\mathcal{A}} \to \mathbb{R}$ is the joint reward function, $P : \mathcal{S} \times \boldsymbol{\mathcal{A}} \times \mathcal{S} \to [0, 1]$ is the transition probability function, $\gamma \in [0, 1)$ is the discount factor, and $d \in \mathcal{P}(X)$ (where $\mathcal{P}(X)$ denotes the set of probability distributions over a set $X$) is initial state distribution. In this work, we use the notation $\mathbb{P}(X)$ to denote the power set of a set $X$ and $\mathrm{Sym}(n)$ to denote the set of permutations of integers $\{1, \ldots, n\}$, known as the symmetric group. At time step $t \in \{1, \ldots, T\}$, each agent $i \in \mathcal{N}$ is at state $\mathrm{s}_t \in \mathcal{S}$ and then takes independent actions $\mathrm{a}_t^i \sim \pi^i(\cdot^i | \mathrm{s}_t)$, where $\pi^i$ is the policy of agent $i$. Let $\mathbf{a}_t = \left(\mathrm{a}_t^1, \ldots, \mathrm{a}_t^n\right) \in \boldsymbol{\mathcal{A}}$ denotes the joint action and $\boldsymbol{\pi}(\cdot | \mathrm{s}_t) = \prod_{i=1}^{n} \pi^i\left(\cdot^i | \mathrm{s}_t\right)$ denotes the joint policy. We denote the policy space of agent $i$ as $\Pi^i \triangleq \left\{ \times_{s \in \mathcal{S}} \pi^i\left(\cdot^i | s\right) | \forall s \in \mathcal{S} \right\}$, and the joint policy space as $\Pi \triangleq \left(\Pi^1, \ldots, \Pi^n\right)$. The agents receive a joint reward $r\left(\mathrm{s}_t, \mathbf{a}_t\right)$ and move to the next state $\mathrm{s}_{t+1} \sim P\left(\cdot | \mathrm{s}_t, \mathbf{a}_t\right)$. The initial state distribution $d$, the joint policy $\boldsymbol{\pi}$, and the transition kernel $P$ induce a marginal state distribution at time $t$, denoted by $\rho_{\boldsymbol{\pi}}^t$. We define the (unnormalized) marginal state distribution $\rho_{\boldsymbol{\pi}} \triangleq \sum_{t=1}^{T} \rho_{\boldsymbol{\pi}}^t$. The standard joint objective of all agents is to maximize the expected total reward, defined as[1]

$$J_{\mathrm{std}}(\boldsymbol{\pi}) = \mathbb{E}_{\mathrm{s}_{1:T} \sim \rho_{\boldsymbol{\pi}}^{1:T}, \mathbf{a}_{1:T} \sim \boldsymbol{\pi}} \left[ \sum_{t=1}^{T} r\left(\mathrm{s}_t, \mathbf{a}_t\right) \right]. \tag{1}$$

### 3.2    LIMITATIONS OF EXISTING COOPERATIVE MARL METHODS

Existing MAPG methods maximizing the standard joint objective (Equation 1), such as MAPPO and HAPPO, may fail to find the optimal NE and converge prematurely. We analyze this problem by considering a singe-state 2-agent cooperative matrix game as shown in Figure 1.

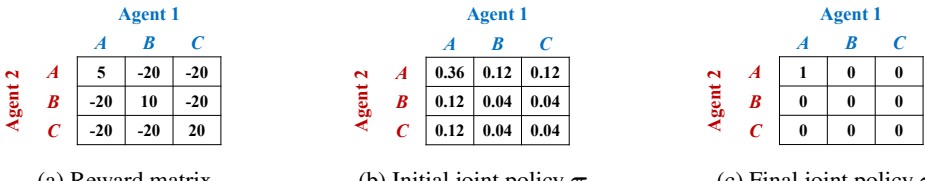

(a) Reward matrix      (b) Initial joint policy $\boldsymbol{\pi}$      (c) Final joint policy $\boldsymbol{\pi}$

Figure 1: A single-state 2-agent cooperative matrix game. (a) is the reward matrix of joint actions. (b) represents the initial joint policy $\boldsymbol{\pi}$ formed by both agents taking the individual policy $\pi = \{0.6, 0.2, 0.2\}$. (c) represents the final joint policy $\boldsymbol{\pi}$ that MAPPO and HAPPO converge to, deterministically choosing action $(A, A)$.

Agents each choose from three possible actions $\{A, B, C\}$ and receive a joint reward. In this case, $(A, A), (B, B), (C, C)$ are three different NEs with rewards of $5, 10, 20$, with $(C, C)$ being the optimal NE. To simulate common scenarios where finding the global optima is challenging and, due to the learning in the early stages, agents are updated towards some reasonably good local optima initially, we consider a setting where agents assign a higher probability to action $(A, A)$, as shown in Figure 1b. By exact calculation (see Appendix B), we find traditional algorithms like MAPPO and HAPPO converge rapidly towards the suboptimal point $(A, A)$, failing to identify the Pareto-optimal equilibria $(C, C)$, as shown in Figure 1c, which is also corroborated by our experiment (Section 5.1).

The explanation is that maximizing the standard joint objective (Equation 1) leads to convergence towards the deterministic policies exploiting the local optimum they have explored so far (28). Specifically, the standard objective discourages any unilateral deviation from the local optimum as it will result in a decrease in the joint reward. As a result, agents tend to deterministically converge to the suboptimal NE. To address this issue, we propose to learn stochastic behaviors, which always preserve the probability of selecting currently underexplored actions, and will eventually converge to the QRE where action probability is proportional to the exponential of action value. We will show in the experimental section that our method could successfully converge to the Pareto-optimal equilibrium $(C, C)$, even when it has a higher probability of choosing action $(A, A)$ initially.

---

[1]We write $a^i$, $\boldsymbol{a}$, and $s$ when we refer to the action, joint action, and state as to values, and $\mathrm{a}^i$, $\mathbf{a}$, and $\mathrm{s}$ as to random variable.

## 4 METHOD

In this section, we establish *maximum entropy heterogeneous-agent reinforcement learning* - a framework for learning stochastic policies in cooperative MARL settings, which alleviates the suboptimal convergence problem mentioned in Section 3.2. We name it *heterogeneous-agent* (HA) as it is a substantial improvement to HARL (45), its Proposition 1 builds upon the prior advantage decomposition lemma (13), and it is generally applicable to HA settings. This framework encompasses three key components, including the derivation of MaxEnt MARL objective from a probabilistic inference perspective in Section 4.1, the HASPI procedure and HASAC algorithm in Section 4.2, and the unified MEHAML template for theoretically-justified algorithmic design in Section 4.3.

### 4.1 MAXIMUM ENTROPY MULTI-AGENT REINFORCEMENT LEARNING

To formalize the idea of learning stochastic policies $\pi(\cdot|s_t)$, we embed cooperative MARL problem into the PGM (Figure 2). Following Levine (17), we introduce an additional optimality variable $\mathcal{O}_t$, which takes on binary values indicating the optimality of joint actions taken by all agents. Specifically, $\mathcal{O}_t = 1$ denotes that time step $t$ is optimal, and we model it as $p(\mathcal{O}_t = 1|s_t, \boldsymbol{a}_t) \propto \exp(r(s_t, \boldsymbol{a}_t))$. Then we use structured variational inference to approximate the posterior distribution $p(\tau|\mathcal{O}_{1:T} = 1) \propto \left[ p(s_1) \prod_{t=1}^{T} p(s_{t+1}|s_t, \boldsymbol{a}_t) \right] \exp\left( \sum_{t=1}^{T} r(s_t, \boldsymbol{a}_t) \right)$ over trajectory $\tau$ with the distribution $q(\tau) = q(s_1) \prod_{t=1}^{T} q(s_{t+1}|s_t, \boldsymbol{a}_t) q(\boldsymbol{a}_t|s_t)$, where we fix the environment dynamics $q(s_1) = p(s_1)$ and $q(s_{t+1}|s_t, \boldsymbol{a}_t) = p(s_{t+1}|s_t, \boldsymbol{a}_t)$ to avoid risk-seeking behaviors (17). This

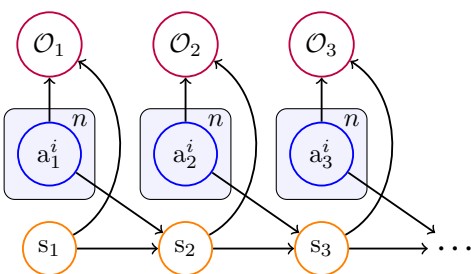

Figure 2: The probabilistic graphical model for cooperative MARL.

inference procedure leads to the *maximum entropy* (MaxEnt) objective of MARL (see Appendix C):

$$J(\boldsymbol{\pi}) = \mathbb{E}_{s_{1:T} \sim \rho_{\boldsymbol{\pi}}^{1:T}, \mathbf{a}_{1:T} \sim \boldsymbol{\pi}} \left[ \sum_{t=1}^{T} \left( r(s_t, \boldsymbol{a}_t) + \alpha \sum_{i=1}^{n} \mathcal{H}\left( \pi^i(\cdot|s_t) \right) \right) \right], \quad (2)$$

where $\alpha$ is the temperature constant that trades off between reward and entropy maximization, and when $\alpha = 0$, the objective is reduced to standard MARL.

Augmenting standard MARL objective with an entropy term (Equation 2) aligns with the solution concept of *quantal response equilibrium* (QRE) proposed by McKelvey & Palfrey (20), which is a generalization of the standard notion of *Nash equilibrium* (NE) in game theory. In a QRE, payoffs are perturbed by additive disturbances (entropy term in our case) so that players do not deterministically choose the strategy with the highest observed payoff, but rather assign the probability mass in its strategies according to every strategy's payoff (6). The following Theorem 1 shows that the QRE policies of MaxEnt objective can be represented as Boltzmann distributions:

**Theorem 1 (Representation of QRE).** *A joint policy $\boldsymbol{\pi}_{QRE} \in \boldsymbol{\Pi}$ is a QRE if none of the agents can increase the maximum entropy objective (Equation 2) by unilaterally altering its policy, i.e.,*

$$\forall i \in \mathcal{N}, \forall \pi^i \in \Pi^i, J\left( \pi^i, \boldsymbol{\pi}_{QRE}^{-i} \right) \leq J\left( \boldsymbol{\pi}_{QRE} \right).$$

*Then the QRE policies are given by*

$$\forall i \in \mathcal{N}, \pi_{QRE}^i\left( a^i|s \right) := \frac{\exp\left( \alpha^{-1} \mathbb{E}_{\mathbf{a}^{-i} \sim \boldsymbol{\pi}_{QRE}^{-i}} \left[ Q_{\boldsymbol{\pi}_{QRE}}\left( s, a^i, \mathbf{a}^{-i} \right) \right] \right)}{\sum_{b^i \in \mathcal{A}^i} \exp\left( \alpha^{-1} \mathbb{E}_{\mathbf{a}^{-i} \sim \boldsymbol{\pi}_{QRE}^{-i}} \left[ Q_{\boldsymbol{\pi}_{QRE}}\left( s, b^i, \mathbf{a}^{-i} \right) \right] \right)}, \quad (3)$$

*where the soft Q-functions are defined as follows,*

$$Q_{\boldsymbol{\pi}}(s, \boldsymbol{a}) = r(s, \boldsymbol{a}) + \mathbb{E}_{\mathbf{a}_{1:\infty} \sim \boldsymbol{\pi}, s_{1:\infty} \sim P} \left[ \sum_{t=1}^{\infty} \gamma^t \left( r(s_t, \boldsymbol{a}_t) + \alpha \sum_{i=1}^{n} \mathcal{H}\left( \pi^i(\cdot|s_t) \right) \right) \Big| s_0 = s, \mathbf{a}_0 = \boldsymbol{a} \right]. \quad (4)$$

Proof can be found in Appendix D. Theorem 1 illustrates the connection between QRE policies and an energy-based model, where the term $\frac{1}{\alpha} Q_{\boldsymbol{\pi}_{QRE}}(s, \boldsymbol{a})$ serves as the negative energy. When $\alpha > 0$,

the QRE policies (Equation 3) are no longer deterministic but rather can represent all the ways of performing a task. This suggests that the inclusion of entropy term makes the policy $\pi_{\text{QRE}}$ stochastic, enabling more effective exploration of the environment (21), which is consistent with our initial goal of learning stochastic policies in MARL settings.

## 4.2 HETEROGENEOUS-AGENT SOFT ACTOR-CRITIC

In this subsection, we develop *heterogeneous-agent soft policy iteration* (HASPI) to maximize MaxEnt objective (Equation 2), which alternates between joint soft policy evaluation and heterogeneous-agent soft policy improvement, and then derive HASAC based on this theory.

### 4.2.1 HETEROGENEOUS-AGENT SOFT POLICY ITERATION

In joint soft policy evaluation step of HASPI, we compute soft Q-value from any $Q(s, \boldsymbol{a}) : \mathcal{S} \times \boldsymbol{\mathcal{A}} \to \mathbb{R}$ by repeatedly applying a soft Bellman backup operator $\Gamma_{\boldsymbol{\pi}}$ given by:

$$\Gamma_{\boldsymbol{\pi}} Q(s, \boldsymbol{a}) \triangleq r(s, \boldsymbol{a}) + \gamma \mathbb{E}_{s' \sim P} \left[ V\left(s'\right) \right], \tag{5}$$

$$\text{where} \quad V(s) = \mathbb{E}_{\mathbf{a} \sim \boldsymbol{\pi}} \left[ Q(s, \mathbf{a}) + \alpha \sum_{i=1}^{n} \mathcal{H}\left(\pi^i\left(\cdot^i|s\right)\right) \right]. \tag{6}$$

is the soft value function. We can obtain the soft Q-function of any joint policy $\boldsymbol{\pi}$ as shown in Lemma 4.1. Notably, the same method for updating soft Q-function has been proposed in FOP (44) since it is the straightforward application of soft Bellman equation.

**Lemma 4.1** (**Joint Soft Policy Evaluation**). *Consider the soft Bellman backup operator $\Gamma_{\boldsymbol{\pi}}$ and a mapping $Q_0 : \mathcal{S} \times \boldsymbol{\mathcal{A}} \to \mathbb{R}$ with $|\boldsymbol{\mathcal{A}}| < \infty$, and define $Q_{k+1} = \Gamma_{\boldsymbol{\pi}} Q_k$. Then the sequence $Q_k$ will converge to the joint soft Q-function $\boldsymbol{\pi}$ as $k \to \infty$.*

Proof can be found in E.1. In policy improvement step, we show that the joint policy $\boldsymbol{\pi}$ can be updated based on individual policy updates. To this end, we first introduce the following definition.

**Definition 1.** *Let $i_{1:m} = \{i_1, \ldots, i_m\} \subseteq \mathcal{N}$ be an ordered subset of agents, and let $-i_{1:m}$ refer to its complement. We write $i_k$ when we refer to the $k^{th}$ agent in the ordered subset. Correspondingly, the multi-agent soft Q-function is defined as*

$$Q_{\boldsymbol{\pi}}^{i_{1:m}}\left(s, \boldsymbol{a}^{i_{1:m}}\right) \triangleq \mathbb{E}_{\mathbf{a}^{-i_{1:m}} \sim \boldsymbol{\pi}^{-i_{1:m}}} \left[ Q_{\boldsymbol{\pi}}\left(s, \boldsymbol{a}^{i_{1:m}}, \mathbf{a}^{-i_{1:m}}\right) + \alpha \sum_{i \in -i_{1:m}} \mathcal{H}\left(\pi^i\left(\cdot^i|s\right)\right) \right]. \tag{7}$$

In the case where $m = n$, $Q_{\boldsymbol{\pi}}^{i_{1:n}}\left(s, \boldsymbol{a}^{i_{1:n}}\right)$ takes the form $Q_{\boldsymbol{\pi}}(s, \boldsymbol{a})$, representing the joint soft Q-function. When $m = 0$, *i.e.*, $i_{1:m} = \emptyset$, the function represents the soft value function $V_{\boldsymbol{\pi}}(s)$.

With this notation defined, we introduce a pivotal proposition that shows the joint soft policy update can be decomposed into a multiplication of sequential local policy updates.

**Proposition 1** (**Joint Soft Policy Decomposition**). *Let $\pi$ be a joint policy, and $i_{1:n} \in \text{Sym}(n)$ be an agent permutation. Suppose that, for each state $s$ and every $m = 1, \ldots, n$,*

$$\pi_{new}^{i_m} = \arg \min_{\pi^{i_m} \in \Pi^{i_m}} \mathrm{D}_{\mathrm{KL}} \left( \pi^{i_m}\left(\cdot^{i_m}|s\right) \| \frac{\exp\left( \mathbb{E}_{\mathbf{a}^{i_{1:m-1}} \sim \boldsymbol{\pi}_{new}^{i_{1:m-1}}} \left[ \frac{1}{\alpha} Q_{\boldsymbol{\pi}_{old}}^{i_{1:m}}\left(s, \mathbf{a}^{i_{1:m-1}}, \cdot^{i_m}\right) \right] \right)}{\mathbb{E}_{\mathbf{a}^{i_{1:m-1}} \sim \boldsymbol{\pi}_{new}^{i_{1:m-1}}} \left[ Z_{\boldsymbol{\pi}_{old}}\left(s, \mathbf{a}^{i_{1:m-1}}\right) \right]} \right), \tag{8}$$

*where $\mathbf{a}^{i_{1:m-1}}$ is drawn from the policy $\boldsymbol{\pi}_{new}^{i_{1:m-1}}(\cdot|s)$ and the partition function $Z_{\boldsymbol{\pi}_{old}}\left(s, \mathbf{a}^{i_{1:m-1}}\right)$ normalizes the distribution. Then the joint policy satisfies the following:*

$$\boldsymbol{\pi}_{new} = \arg \min_{\boldsymbol{\pi} \in \boldsymbol{\Pi}} \mathrm{D}_{\mathrm{KL}} \left( \boldsymbol{\pi}\left(\cdot|s\right) \| \frac{\exp\left( \frac{1}{\alpha} Q_{\boldsymbol{\pi}_{old}}\left(s, \cdot\right) \right)}{Z_{\boldsymbol{\pi}_{old}}\left(s\right)} \right). \tag{9}$$

Proof can be found in E.2. Proposition 1 holds significance due to the crucial insight it provides, suggesting that a MaxEnt MARL problem can be considered as a sum of $n$ MaxEnt RL problems. It indicates an effective *heterogeneous-agent* approach to improving the joint soft policy in multi-agent learning, where each agent optimizes individual KL-divergence sequentially leading to the optimization of joint soft policy. To formally extend the above process into a policy improvement procedure with theoretical guarantees of monotonic improvement and convergence to QRE, we propose the *heterogeneous-agent soft policy improvement* as formalized below.

**Lemma 4.2 (Heterogeneous-Agent Soft Policy Improvement).** *Let $i_{1:n} \in \mathrm{Sym}(n)$ be an agent permutation, and for every $m = 1, \ldots, n$, let policy $\pi_{old}^{i_m} \in \Pi^{i_m}$ and $\pi_{new}^{i_m}$ be the optimizer of the minimization problem defined in Equation 8. Then $Q_{\boldsymbol{\pi}_{new}}(s, \boldsymbol{a}) \geq Q_{\boldsymbol{\pi}_{old}}(s, \boldsymbol{a})$ for all $(s, \boldsymbol{a}) \in \mathcal{S} \times \boldsymbol{\mathcal{A}}$ with $|\boldsymbol{\mathcal{A}}| < \infty$ and $J(\boldsymbol{\pi}_{new}) \geq J(\boldsymbol{\pi}_{old})$.*

Proof can be found in E.3. Lemma 4.2 guarantees that soft Q-function and MaxEnt objective monotonically increase at each policy improvement step. Next, we propose *heterogeneous-agent soft policy iteration*, which alternates between joint soft policy evaluation and heterogeneous-agent soft policy improvement, and prove that joint policy $\boldsymbol{\pi}$ converges to a QRE.

**Theorem 2 (Heterogeneous-Agent Soft Policy Iteration).** *For any joint policy $\boldsymbol{\pi} \in \boldsymbol{\Pi}$, if we repeatedly apply joint soft policy evaluation and heterogeneous-agent soft policy improvement from $\pi^i \in \Pi^i$. Then the joint policy $\boldsymbol{\pi} = \prod_{i=1}^n \pi^i$ converges to $\boldsymbol{\pi}_{QRE}$ in Theorem 1.*

Proof can be found in E.4. To obtain such theoretical results, the updating approach in Proposition 1 plays a pivotal role. Lemma 4.2 ensures that, through the updates in Proposition 1, soft Q-function increases monotonically, leading to convergence of the policies. Eventually, none of the agents is motivated to make an update (Equation 8) at convergence, thereby establishing a QRE.

### 4.2.2 PRACTICAL ALGORITHM

In practice, large continuous domains require us to derive a practical approximation to the procedure above. We will use function approximators for both centralized soft Q-function $Q_\theta(s_t, \mathbf{a}_t)$ and tractable decentralized policies $\pi_{\phi^{i_m}}^{i_m}(a_t^{i_m}|s_t)$, for each agent $i_m$, parameterized respectively by $\theta$ and $\phi^{i_m}$, and alternate between optimizing both networks with stochastic gradient descent.

The centralized soft Q-function parameters can be trained to minimize the Bellman residual

$$J_Q(\theta) = \mathbb{E}_{(s_t, \mathbf{a}_t) \sim \mathcal{D}} \left[ \frac{1}{2} \left( Q_\theta(s_t, \mathbf{a}_t) - \left( r(s_t, \mathbf{a}_t) + \gamma \mathbb{E}_{s_{t+1} \sim P} [V_{\bar{\theta}}(s_{t+1})] \right) \right)^2 \right],$$

where the soft value function is implicitly parameterized through the soft Q-function parameters via Equation 6. Then we draw a permutation $i_{1:n} \in \mathrm{Sym}(n)$ and sequentially update the policy of each agent $i_m$ according to Equation 8. The policy parameters can be learned by directly minimizing the expected KL-divergence in Equation 8 disregarding the constant log-partition function

$$J_{\pi^{i_m}}(\phi^{i_m}) = \mathbb{E}_{s_t \sim \mathcal{D}} \left[ \mathbb{E}_{\mathbf{a}_t^{i_{1:m-1}} \sim \boldsymbol{\pi}_{\phi_{new}^{i_{1:m-1}}}^{i_{1:m-1}}, a_t^{i_m} \sim \pi_{\phi^{i_m}}^{i_m}} \left[ \alpha \log \pi_{\phi^{i_m}}^{i_m}(a_t^{i_m}|s_t) - Q_{\boldsymbol{\pi}_{old};\theta}^{i_{1:m}}(s_t, \mathbf{a}_t^{i_{1:m-1}}, a_t^{i_m}) \right] \right].$$
(10)

We refer to the above procedure as HASAC and Appendix F for its full pseudocode.

### 4.3 MAXIMUM ENTROPY HETEROGENEOUS-AGENT MIRROR LEARNING

In addition to updating policies by directly minimizing the KL-divergence in Equation 8, we aim to propose a generalized HASPI procedure that provides a range of solutions to MaxEnt MARL problem. To this end, we start by introducing the necessary definitions of the operators proposed in HAML (15): the drift functional (HADF) $\mathfrak{D}_{\boldsymbol{\pi}}^i(\hat{\pi}^i|s, \bar{\pi}^{j_{1:m}})$ which, intuitively, is a notion of distance between $\pi^i$ and $\hat{\pi}^i$, given that agents $j_{1:m}$ just updated to $\bar{\pi}^{j_{1:m}}$; the neighborhood operator $\mathcal{U}_{\boldsymbol{\pi}}^i(\pi^i)$ which forms a region around the policy $\pi^i$; as well as a sampling distribution $\beta_{\boldsymbol{\pi}} \in \mathcal{P}(\mathcal{S})$ that is continuous in $\boldsymbol{\pi}$ (see detailed definitions in Appendix A.3). As shown in (15; 14), these operators allow for effective abstraction of standard RL and MARL methods due to their generality.

We present the generalized HASPI using these operators. In heterogeneous-agent soft policy improvement step, we redefine the operator that agents optimize as follows,

**Definition 2.** *Let $i \in \mathcal{N}, j^{1:m} \in \mathbb{P}(-i)$, and $\mathfrak{D}^i$ be a HADF of agent $i$. The maximum entropy heterogeneous-agent mirror operator (MEHAMO) integrates the soft Q-function as*

$$\left[ \mathcal{M}_{\mathfrak{D}^i, \bar{\pi}^{j_{1:m}}}^{(\hat{\pi}^i)} V_{\boldsymbol{\pi}} \right](s) \triangleq \mathbb{E}_{\mathbf{a}^{j_{1:m}} \sim \bar{\pi}^{j_{1:m}}, a^i \sim \hat{\pi}^i} \left[ Q_{\boldsymbol{\pi}}^{j_{1:m}, i}(s, \mathbf{a}^{j_{1:m}}, a^i) - \alpha \log \hat{\pi}^i(a^i|s) \right] - \mathfrak{D}_{\boldsymbol{\pi}}^i(\hat{\pi}^i|s, \bar{\pi}^{j_{1:m}}).$$

Then we propose MEHAML Algorithm template 1 to formalize *generalized* heterogeneous-agent soft policy iteration. Notably, HAML is a special instance of our template when $\alpha = 0$.

---

**Algorithm 1:** Maximum Entropy Heterogeneous-Agent Mirror Learning

---

Initialise a joint policy $\boldsymbol{\pi}_0 = \left(\pi_0^1, \ldots, \pi_0^n\right)$;

**for** $k = 0, 1, \cdots$ **do**

    Compute the soft Q-function $Q_{\boldsymbol{\pi}_k}(s, \boldsymbol{a})$ for all state-(joint)action pairs $(s, \boldsymbol{a})$;

    Update $Q_{\boldsymbol{\pi}_k}(s, \boldsymbol{a})$ according to the soft Bellman equation (Equation 5);

<p align="center"><em style="color:blue">Joint Soft Policy Evaluation</p>

    Draw a permutaion $i_{1:n}$ of agents at random;

    **for** $m = 1 : n$ **do**

        Make an update $\pi_{k+1}^{i_m} = \underset{\pi^{i_m} \in \mathcal{U}_{\boldsymbol{\pi}_k}^{i_m}\left(\pi_k^{i_m}\right)}{\arg\max} \mathbb{E}_{\mathrm{s} \sim \beta_{\boldsymbol{\pi}_k}} \left[\left[\mathcal{M}_{\mathfrak{D}^{i_m}, \boldsymbol{\pi}_{k+1}^{i_{1:m-1}}}^{\left(\pi^{i_m}\right)} V_{\boldsymbol{\pi}_k}\right](\mathrm{s})\right]$;

<p align="center"><em style="color:blue">(Generalized) Heterogeneous-Agent Soft Policy Improvement</p>

    **end**

**end**

**Output:** A limit-point joint policy $\boldsymbol{\pi}_\infty$

---

Despite the presence of a drift penalty and a neighborhood constraint, optimizing the *MEHAMO* sequentially is sufficient to guarantee the same desired properties as HASPI. We establish the complete list of the core MEHAML properties in Theorem 3, which confirms that any method derived from Algorithm 1 has the desired properties of monotonic improvement of the MaxEnt objective and QRE convergence (detailed proof can be found in Appendix G).

**Theorem 3** (**The Core Theorem of MEHAML**). *Let $\boldsymbol{\pi}_0 \in \Pi$, and the sequence of joint policies $(\boldsymbol{\pi}_k)_{k=0}^\infty$ be obtained by a MEHAML algorithm 1 induced by $\mathfrak{D}^i, \mathcal{U}^i, \forall i \in \mathcal{N}$, and $\beta_{\boldsymbol{\pi}}$. Then, the joint policies induced by the algorithm enjoy the following list of properties (1) Attain the monotonic improvement property $J\left(\boldsymbol{\pi}_{k+1}\right) \geq J\left(\boldsymbol{\pi}_k\right)$, (2) Their value functions converge to a quantal response value function $\lim_{k \to \infty} V_{\boldsymbol{\pi}_k} = V^{QRE}$, (3) Their expected returns converge to a quantal response return $\lim_{k \to \infty} J\left(\boldsymbol{\pi}_k\right) = J^{QRE}$, (4) Their $\omega$-limit set consists of quantal response equilibria.*

*Proof sketch.* We divide the proof into four steps. In *Step 1*, we show that the sequence of soft value function $(V_{\boldsymbol{\pi}_k})_{k \in \mathbb{N}}$ increases monotonically and converges to a limit point. In *Step 2*, we show the existence of limit points $\bar{\boldsymbol{\pi}}$ of $(\boldsymbol{\pi}_k)_{k \in \mathbb{N}}$, and that they are fixed points of the MEHAML update. In the most important *Step 3*, we prove that $\bar{\boldsymbol{\pi}}$ is also a fixed point of soft policy iteration by leveraging the concavity of MEHAMO. *Step 4* finalizes the proof, proving that fixed points of soft policy iteration are QRE policies.

By selecting appropriate HADFs and neighborhood operators that satisfy the definitions, Algorithm 1 has the potential to generate various theoretically-justified algorithms to solve MaxEnt MARL problem. The drifts $\mathfrak{D}_{\boldsymbol{\pi}}^i \left(\hat{\pi}^i | s, \bar{\boldsymbol{\pi}}^{j_{1:m}}\right)$ can serve as soft constraints, such as KL-divergence, controlling the distance between $\hat{\pi}^i$ and $\pi^i$ when the agents $j_{1:m}$ have just updated to $\bar{\boldsymbol{\pi}}^{j_{1:m}}$. Additionally, the neighborhood operators $\mathcal{U}^i$ can generate small policy-space subsets, serving as hard constraints, then the resultant policy improvement will remain within a small range due to the fact that $\pi^i \in \mathcal{U}_{\boldsymbol{\pi}}^i\left(\pi^i\right), \forall i \in \mathcal{N}, \pi^i \in \Pi^i$. Therefore, algorithms equipped with appropriate HADFs and neighborhoods can learn stochastic policies in a stable and coordinated manner. In summary, MEHAML provides a template for generating theoretically sound, stable, monotonically improving algorithms that enable agents to learn stochastic policies to solve multi-agent cooperation tasks.

## 5 EXPERIMENTS

To demonstrate the advantages of the stochastic policies learned by HASAC, we conduct comprehensive experiments on the matrix game in Section 3.2 and six benchmarks — MAMuJoCo (4), Bi-DexHands (2), SMAC (25), GRF (16), MPE (19), and LAG (23)[2]. It is important to note that while HASAC is originally designed for continuous actions, we employ a Gumbel-Softmax (10) to ensure that HASAC would work for discrete actions. **Compared to existing SOTA MAPG methods, HASAC achieves the best performance in 31 out of 35 tasks across all benchmarks.** In addition to final performance, experimental results (see full experimental details and hyperparameter in Appendix

---

[2]HASAC achieves best performance on fully cooperative MPE and LAG tasks. See Appendix H.2.5 and H.2.6 for full results.

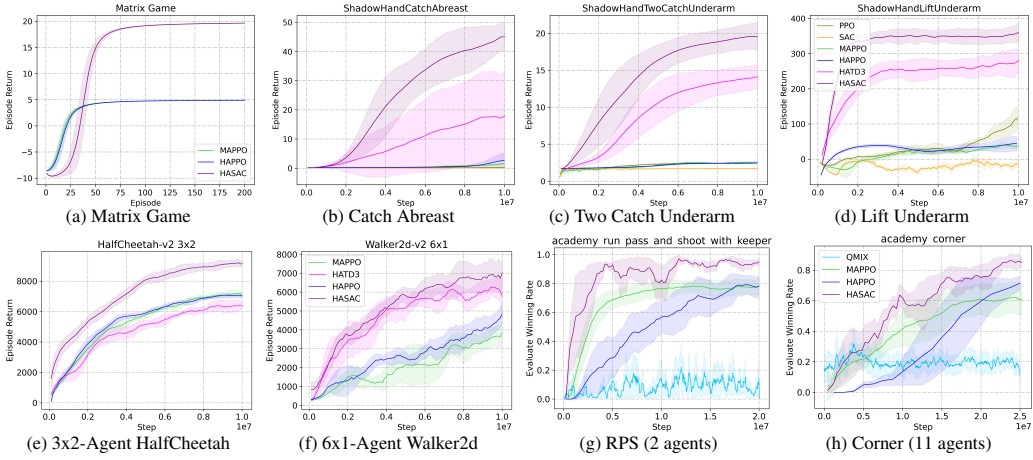

Figure 3: Performance comparisons on selected tasks of multiple benchmarks.

| Map | Difficulty | HASAC | HAPPO | HATRPO | MAPPO | QMIX | FOP | Timesteps |
|---|---|---|---|---|---|---|---|---|
| 8m_vs_9m | Hard | **97.5(1.2)** | 83.8(4.1) | 92.5(3.7) | 87.5(4.0) | 92.2(1.0) | 23.4(1.6) | 1e7 |
| 5m_vs_6m | Hard | **90.0(3.9)** | 77.5(7.2) | 75.0(6.5) | 75.0(18.2) | 77.3(3.3) | 46.9(5.3) | 1e7 |
| 3s5z | Hard | **100.0(0.0)** | 97.5(1.2) | 93.8(1.2) | 96.9(0.7) | 89.8(2.5) | 93.0(0.8) | 1e7 |
| 10m_vs_11m | Hard | 95.0(3.1) | 87.5(6.7) | **98.8(0.6)** | 96.9(4.8) | 95.3(2.2) | 12.5(6.2) | 1e7 |
| MMM2 | Super Hard | **97.5(2.4)** | 88.8(2.0) | **97.5(6.4)** | 93.8(4.7) | 87.5(2.5) | 37.5(28.1) | 2e7 |
| 3s5z_vs_3s6z | Super Hard | **82.5(4.1)** | 66.2(3.1) | 72.5(14.7) | 70.0(10.7) | **87.5(12.6)** | 0.0(0.0) | 2e7 |
| corridor | Super Hard | 90.0(10.8) | 92.5(13.9) | 88.8(2.7) | **97.5(1.2)** | 82.8(4.4) | 0.0(0.0) | 2e7 |
| 6h_vs_8z | Super Hard | **95.0(3.1)** | 76.2(3.1) | 78.8(0.6) | 85.0(2.0) | 7.0(27.0) | 0.0(0.0) | 4e7 |

Table 1: Median evaluate winning rate and standard deviation on eight SMAC maps for different methods. All values within 1 standard deviation of the maximum score rate are marked in bold.

H) show the following advantages of stochastic policies: **(1)** *HASAC exhibits similar performance across different random seeds and higher training stability* , **(2)** *HASAC shows higher learning speed compared to existing algorithms*, and **(3)** *HASAC improves agents' exploration, which facilitates policies to escape from suboptimal equilibria and converge towards a higher reward equilibrium.*

## 5.1 EXPERIMENTAL RESULTS

**Matrix Game.** We show the results of HASAC, HAPPO, MAPPO over 200 learning episodes in matrix game (Figure 1a). HASAC escapes local optima and achieves the Pareto optimum due to the learned stochastic policies, while the other two methods fall into the suboptimal NE (Figure 3a).

**Bi-DexHands.** Bi-DexHands offers numerous bimanual manipulation tasks that match various human skill levels. In the challenging Catch Abreast, Two Catch Underarm, and Lift Underarm tasks (Figure 3b, 3c, and 3d), all on-policy methods fail within 10m steps due to low sample efficiency. HATD3 (45), on the other hand, exhibits very high variability and prematurely converges to local optima. In contrast, HASAC outperforms the other five methods by a large margin. It also exhibits higher convergence speed and improved robustness, demonstrating the benefits of stochastic policies.

**Multi-Agent MuJoCo.** We compare our method to HAPPO, MAPPO, and HATD3 in ten MAMu-JoCo tasks (see Appendix H.2.2). Figure 3e, 3f, and the other results in Appendix H.2.2 demonstrate that HASAC enjoys superior performance over the three rivals both in terms of reward values and learning speed. It's worth noting that, although both HATD3 and HASAC are off-policy algorithms, we generally observe that HASAC learns faster with higher sample efficiency compared to HATD3. This is because the exploration mechanism of HATD3 involves adding Gaussian noise to a determin-istic policy, resulting in lower exploration efficiency, requiring more samples to explore the reward landscape. In contrast, HASAC, due to its inherent encouragement of exploration within stochastic policies, can effectively explore the entire reward landscape, thus learning better behaviors rapidly.

**StarCraft Multi-Agent Challenge.** We evaluate our method on four hard and four super-hard maps. As shown in Table 1, HASAC achieves over 90% win rates in 7 out of 8 maps and outperforms other strong baselines in most maps. Notably, in particularly challenging tasks such as 5m_vs_6m, 3s5z_vs_3s5z, and 6h_vs_8z, we observe that HAPPO and HATRPO would converge towards suboptimal NE. In addition, FOP is unable to learn meaningful joint policy in super-hard tasks due to its reliance on the IGO assumption. By contrast, HASAC consistently achieves superior performance

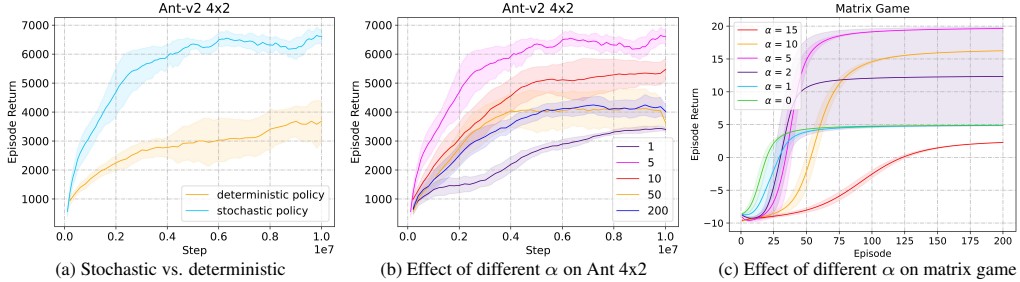

Figure 4: Performance comparison between HASAC with different hyperparameters on Ant-v2 4x2 task and matrix game. (a) The comparison indicates that stochastic policy can lead to better equilibrium and stabilize training. (b) & (c) HASAC converges to different QRE with different temperature parameter $\alpha$.

|  | **Agent 1** | | |
|---|---|---|---|
|  | **A** | **B** | **C** |
| **A** | 1 | 0 | 0 |
| **B** | 0 | 0 | 0 |
| **C** | 0 | 0 | 0 |

(a) HASAC: $\alpha = 1$

|  | **Agent 1** | | |
|---|---|---|---|
|  | **A** | **B** | **C** |
| **A** | 0 | 0 | 0 |
| **B** | 0 | 0 | 0 |
| **C** | 0 | 0 | 1 |

(b) HASAC: $\alpha = 5$

|  | **Agent 1** | | |
|---|---|---|---|
|  | **A** | **B** | **C** |
| **A** | 0.0025 | 0.0025 | 0.02 |
| **B** | 0.0025 | 0.0025 | 0.02 |
| **C** | 0.02 | 0.02 | 0.91 |

(c) HASAC: $\alpha = 10$

|  | **Agent 1** | | |
|---|---|---|---|
|  | **A** | **B** | **C** |
| **A** | 0.01 | 0.01 | 0.1 |
| **B** | 0.01 | 0.01 | 0.1 |
| **C** | 0.1 | 0.1 | 0.56 |

(d) HASAC: $\alpha = 15$

Figure 5: Different temperature $\alpha$ leads to convergence of different QRE.

and shows the ability to identify higher reward equilibria due to its extensive exploration. We also observe that HASAC has better stability and higher learning speed across most maps.

**Google Research Football.** We compare HASAC with QMIX, MAPPO, and HAPPO. As shown in Figure 3g and 3h, we generally observe that both MAPPO and HAPPO tend to converge to a non-optimal NE on the two challenging tasks, while HASAC exhibits the ability to attain a higher reward equilibrium by learning stochastic policies which effectively enhance exploration.

## 5.2 ABLATION STUDY

We investigate the benefits of stochastic policies learned by HASAC and *empirically* show how different temperature $\alpha$ values affect the stochasticity of policies, leading to different QRE convergence.

**Stochasticity.** HASAC learns stochastic policies through maximizing the MaxEnt objective (Equation 2). We compare it to a deterministic variant which utilizes deterministic policies with fixed Gaussian exploration noise to maximize standard MARL objective. The results in Figure 4a show that HASAC achieves a higher reward equilibrium and demonstrates better stability compared to the deterministic variant, which exhibits high variance across the different runs. These findings highlight the importance of learning stochastic policies, which can improve robustness, facilitate escape from suboptimal equilibria, and converge to higher reward equilibrium.

**Analysis of temperature $\alpha$.** We further show the effect of temperature $\alpha$ on the stochasticity of policies. As illustrated in Figure 4b (we report $\alpha^{-1}$ in legend), 4c, and 5, when $\alpha$ is large, policies predominantly emphasize maximizing entropy, leading to poor performance due to the failure to exploit the reward signal. Conversely, when $\alpha$ is small, MaxEnt objective almost degrades to standard objective, leading to suboptimal equilibrium due to inadequate exploration. A proper $\alpha$ achieves a trade-off between exploration and exploitation, eventually resulting in better performance. To obtain appropriate $\alpha$, we implement both fixed and auto-tuned (see Appendix H.1 for details) $\alpha$ in practice.

## 6 CONCLUSION

In this paper, we propose *maximum entropy heterogeneous-agent reinforcement learning* (MEHARL) — a unified framework for learning stochastic policies in MARL. This framework comprises three key components, including the PGM derivation of MaxEnt MARL, the HASAC algorithm with monotonic improvement and QRE convergence properties, and the unified MEHAML template that provides any induced MaxEnt method with the same theoretical guarantees as HASAC. To demonstrate the advantages of stochastic policies, we evaluate HASAC on both discrete and continuous control tasks, confirming its superior performance and improved robustness and exploration. For future work, we aim to explore appropriate drift functionals and neighborhood operators to design more principled MaxEnt MARL algorithms that can further enhance performance and stability in multi-agent cooperation tasks.

**Acknowledgements.** This work is sponsored by National Natural Science Foundation of China (62376013) and Beijing Municipal Science & Technology Commission (Z231100007423015).

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

# Appendix

## Table of Contents

# A  PRELIMINARIES

## A.1  TABLE OF ACRONYMS

Table 2 lists the main acronyms used in this paper.

Table 2: The acronyms used in this paper.

| Acronym | Meaning |
|---|---|
| A2PO | Agent-by-agent Policy Optimization |
| CoPPO | Coordinated Proximal Policy Optimization |
| CTDE | Centralized training decentralized execution |
| FOP | Factorizing Optimal Joint Policy |
| GRF | Google Research Football |
| HA | Heterogeneous-Agent |
| HADF | Heterogeneous-agent drift functional |
| HAML | Heterogeneous-Agent Mirror Learning |
| HAPPO | Heterogeneous-Agent Proximal Policy Optimization |
| HARL | Heterogeneous-Agent Reinforcement Learning |
| HASAC | Heterogeneous-Agent Soft Actor-Critic |
| HASPI | Heterogeneous-agent soft policy iteration |
| HATD3 | Heterogeneous-Agent Twin Delayed Deep Deterministic Policy Gradient |
| HATRPO | Heterogeneous-Agent Trust Region Policy Optimization |
| IGO | Individual-Global-Optimal |
| LAG | Light Aircraft Game |
| MACPF | Multi-Agent Conditional Policy Factorization |
| MADDPG | Multi-Agent Deep Deterministic Policy Gradient |
| MAMuJoCo | Multi-Agent MuJoCo |
| MAPG | Multi-agent policy gradient |
| MAPPO | Multi-Agent Proximal Policy Optimization |
| MARL | Multi-agent reinforcement learning |
| MASQL | Multi-Agent Soft Q-Learning |
| MaxEnt | Maximum entropy |
| MEHAML | Maximum Entropy Heterogeneous-Agent Mirror Learning |
| MEHAMO | Maximum Entropy Heterogeneous-Agent Mirror Operator |
| MEHARL | Maximum Entropy Heterogeneous-Agent Reinforcement Learning |
| MPE | Multi-Agent Particle Environment |
| NE | Nash equilibrium |
| PGM | Probabilistic Graphical Model |
| PPO | Proximal Policy Optimization |
| QRE | Quantal response equilibrium |
| RL | Reinforcement learning |
| RPS | Run pass and shoot with keeper |
| SAC | Soft Actor-Critic |
| SMAC | StarCraft Multi-Agent Challenge |
| SQL | Soft Q-Learning |

## A.2  INDIVIDUAL-GLOBAL-OPTIMAL

The Individual-Global-Optimal (**IGO**) assumption is introduced in (44) and can be stated as below.

**Definition 3** (IGO). *For an optimal joint policy* $\boldsymbol{\pi}_\star(\boldsymbol{a}|s) : \mathcal{S} \times \boldsymbol{\mathcal{A}} \to [0, 1]$*, if there exist individual optimal policies* $[\pi^i_\star(\boldsymbol{a}^i|s) : \mathcal{S} \times \mathcal{A}^i \to [0, 1]]^n_{i=1}$*, such that the following holds:*

$$\boldsymbol{\pi}_\star(\boldsymbol{a}|s) = \prod_{i=1}^{n} \pi^i_\star(\boldsymbol{a}^i|s),$$

*then, we say that* $[\pi^i]$ *satisfy* **IGO** *for* $\boldsymbol{\pi}$ *under s.*

The main limitation of IGO arises from the fact that during training, each agent's policy is updated based solely on its local Q-function, thereby ignoring the actions of other agents. As discussed by Wang et al. (32), when multiple agents need to collaborate in decision-making, they may fail to reach a consensus, updating their policies in the direction that increases their local Q-functions, but eventually resulting in a decrease in the global Q-function.

## A.3 DEFINITIONS AND ASSUMPTIONS

Throughout the proofs, we make the following regularity assumption introduced by Kuba et al. (13):

**Assumption 1.** *There exists $\eta \in \mathbb{R}$, such that $0 < \eta \ll 1$, and for every agent $i \in \mathcal{N}$, the policy space $\Pi^i$ is $\eta$-soft; that means that for every $\pi^i \in \Pi^i, s \in \mathcal{S}$, and $a^i \in \mathcal{A}^i$, we have $\pi^i \left(a^i | s\right) \geq \eta$.*

In the following, we provide the essential definitions of the two key components, originally proposed by Kuba et al. (15), that serve as the building blocks of the MEHAML framework. Additionally, we present the definitions of soft advantage function and a notion of distance that will be utilized in the proof of Lemma A.2.

**Definition 4.** *Let $i \in \mathcal{N}$, a **heterogeneous-agent drift functional** (HADF) $\mathfrak{D}^i$ of $i$ consists of a map, which is defined as*

$$\mathfrak{D}^i : \boldsymbol{\Pi} \times \boldsymbol{\Pi} \times \mathbb{P}(-i) \times \mathcal{S} \to \left\{ \mathfrak{D}^i_{\boldsymbol{\pi}}\left(\cdot | s, \bar{\boldsymbol{\pi}}^{j_{1:m}}\right) : \mathcal{P}\left(\mathcal{A}^i\right) \to \mathbb{R} \right\},$$

*such that for all arguments, under notation $\mathfrak{D}^i_{\boldsymbol{\pi}}\left(\hat{\pi}^i | s, \bar{\boldsymbol{\pi}}^{j_{1:m}}\right) \triangleq \mathfrak{D}^i_{\boldsymbol{\pi}}\left(\hat{\pi}^i\left(\cdot^i | s\right) | s, \bar{\boldsymbol{\pi}}^{j_{1:m}}\right),$*

1. *$\mathfrak{D}^i_{\boldsymbol{\pi}}\left(\hat{\pi}^i | s, \bar{\boldsymbol{\pi}}^{j_{1:m}}\right) \geq \mathfrak{D}^i_{\boldsymbol{\pi}}\left(\pi^i | s, \bar{\boldsymbol{\pi}}^{j_{1:m}}\right) = 0$ (non-negativity),*

2. *$\mathfrak{D}^i_{\boldsymbol{\pi}}\left(\hat{\pi}^i | s, \bar{\boldsymbol{\pi}}^{j_{1:m}}\right)$ has all Gâteaux derivatives zero at $\hat{\pi}^i = \pi^i$ (zero gradient),*

*We say that the HADF is positive if $\mathfrak{D}^i_{\boldsymbol{\pi}}\left(\hat{\pi}^i | \bar{\boldsymbol{\pi}}^{j_{1:m}}\right) = 0, \forall s \in \mathcal{S}$ implies $\hat{\pi}^i = \pi^i$, and trivial if $\mathfrak{D}^i_{\boldsymbol{\pi}}\left(\hat{\pi}^i | \bar{\boldsymbol{\pi}}^{j_{1:m}}\right) = 0, \forall s \in \mathcal{S}$ for all $\boldsymbol{\pi}, \bar{\boldsymbol{\pi}}^{j_{1:m}}$, and $\hat{\pi}^i$.*

**Definition 5.** *Let $i \in \mathcal{N}$. We say that, $\mathcal{U}^i : \boldsymbol{\Pi} \times \Pi^i \to \mathbb{P}\left(\Pi^i\right)$ is a neighborhood operator if $\forall \pi^i \in \Pi^i$, $\mathcal{U}^i_{\boldsymbol{\pi}}\left(\pi^i\right)$ contains a closed ball, i.e., there exists a state-wise monotonically non-decreasing metric $\chi : \Pi^i \times \Pi^i \to \mathbb{R}$ such that $\forall \pi^i \in \Pi^i$ there exists $\delta^i > 0$ such that $\chi\left(\pi^i, \bar{\pi}^i\right) \leq \delta^i \implies \bar{\pi}^i \in \mathcal{U}^i_{\boldsymbol{\pi}}\left(\pi^i\right)$.*

**Definition 6.** *Let $i_{1:m} = \{i_1, \ldots, i_m\} \subseteq \mathcal{N}$ and $j_{1:k} = \{j_1, \ldots, j_k\} \subseteq \mathcal{N}$ be disjoint ordered subsets of agents, and let $Q^{i_{1:m}}_{\boldsymbol{\pi}}\left(s, \boldsymbol{a}^{i_{1:m}}\right)$ be the multi-agent soft Q-function. Then the multi-agent soft advantage function is defined as*

$$A^{i_{1:m}}_{\boldsymbol{\pi}}\left(s, \boldsymbol{a}^{j_{1:k}}, \boldsymbol{a}^{i_{1:m}}\right) \triangleq Q^{j_{1:k}, i_{1:m}}_{\boldsymbol{\pi}}\left(s, \boldsymbol{a}^{j_{1:k}}, \boldsymbol{a}^{i_{1:m}}\right) - Q^{j_{1:k}}_{\boldsymbol{\pi}}\left(s, \boldsymbol{a}^{j_{1:k}}\right). \tag{11}$$

**Definition 7.** *Let $X$ be a finite set and $p : X \to \mathbb{R}, q : X \to \mathbb{R}$ be two maps. Then, the notion of **distance** between $p$ and $q$ that we adopt is given by $\|p - q\| \triangleq \max_{x \in X} |p(x) - q(x)|$.*

## A.4 PROOFS OF USEFUL LEMMAS

**Lemma A.1** (Multi-Agent Advantage Decomposition). *Let $\pi$ be a joint policy, and $i_1, \ldots, i_m$ be an arbitrary ordered subset of agents. Then, for any state $s$ and joint action $\boldsymbol{a}^{i_{1:m}}$,*

$$A^{i_{1:m}}_{\boldsymbol{\pi}}\left(s, \boldsymbol{a}^{i_{1:m}}\right) = \sum_{j=1}^{m} A^{i_j}_{\boldsymbol{\pi}}\left(s, \boldsymbol{a}^{i_{1:j-1}}, a^{i_j}\right). \tag{12}$$

*Proof.* (The lemma is proposed in (12) and we quote the proof from Kuba et al. (12)) By the definition of multi-agent soft advantage function,

$$\begin{aligned} A^{i_{1:m}}_{\boldsymbol{\pi}}\left(s, \boldsymbol{a}^{i_{1:m}}\right) &= Q^{i_{1:m}}_{\boldsymbol{\pi}}\left(s, \boldsymbol{a}^{i_{1:m}}\right) - V_{\boldsymbol{\pi}}(s) \\ &= \sum_{j=1}^{m} \left[ Q^{i_{1:j}}_{\boldsymbol{\pi}}\left(s, \boldsymbol{a}^{i_{1:j}}\right) - Q^{i_{1:j-1}}_{\boldsymbol{\pi}}\left(s, \boldsymbol{a}^{i_{1:j-1}}\right) \right] = \sum_{j=1}^{m} A^{i_j}_{\boldsymbol{\pi}}\left(s, \boldsymbol{a}^{i_{1:j-1}}, a^{i_j}\right), \end{aligned}$$

which finishes the proof. $\square$

The continuity of $Q_\pi$ is a crucial requirement for proving Theorem 3 later. Now we first prove that the inclusion of an additional entropy term does not affect the continuity of $Q_\pi$, which is the state-action value function in the single-agent setting, where $\pi$ denotes the policy of a single agent. And finally, we generalize the result to $Q_{\boldsymbol{\pi}}$ in MARL.

**Lemma A.2** (Continuity of $Q_\pi$). *Let $\pi$ be a policy. Then $Q_\pi(s, a)$ is continuous in $\pi$.*

*Proof.* Let $\pi$ and $\hat{\pi}$ be two policies. Then we have

$$|Q_\pi(s,a) - Q_{\hat{\pi}}(s,a)|$$

$$= \left| \left( r(s,a) + \gamma \sum_{s'} P(s'|s,a) \left( \sum_{a'} \pi(a'|s')Q_\pi(s',a') - \alpha \sum_{a'} \pi(a'|s') \log \pi(a'|s') \right) \right) \right.$$

$$\left. - \left( r(s,a) + \gamma \sum_{s'} P(s'|s,a) \left( \sum_{a'} \hat{\pi}(a'|s')Q_{\hat{\pi}}(s',a') - \alpha \sum_{a'} \hat{\pi}(a'|s') \log \hat{\pi}(a'|s') \right) \right) \right|$$

$$= \gamma \left| \sum_{s'} P(s'|s,a) \left( \sum_{a'} [\pi(a'|s')Q_\pi(s',a') - \hat{\pi}(a'|s')Q_{\hat{\pi}}(s',a')] \right. \right.$$

$$\left. \left. -\alpha \sum_{a'} [\pi(a'|s') \log \pi(a'|s') - \hat{\pi}(a'|s') \log \hat{\pi}(a'|s')] \right) \right|$$

$$\leq \gamma \sum_{s'} P(s'|s,a) \left( \sum_{a'} |\pi(a'|s')Q_\pi(s',a') - \hat{\pi}(a'|s')Q_{\hat{\pi}}(s',a')| \right.$$

$$\left. + \alpha \sum_{a'} |\pi(a'|s') \log \pi(a'|s') - \hat{\pi}(a'|s') \log \hat{\pi}(a'|s')| \right)$$

$$= \gamma \sum_{s'} P(s'|s,a) \left( \sum_{a'} |\pi(a'|s')Q_\pi(s',a') - \hat{\pi}(a'|s')Q_\pi(s',a') \right.$$

$$+ \hat{\pi}(a'|s')Q_\pi(s',a') - \hat{\pi}(a'|s')Q_{\hat{\pi}}(s',a')|$$

$$\left. + \alpha \sum_{a'} |(\pi(a'|s') - \hat{\pi}(a'|s')) \log \pi(a'|s') + \hat{\pi}(a'|s')(\log \pi(a'|s') - \log \hat{\pi}(a'|s'))| \right)$$

$$\leq \gamma \sum_{s'} P(s'|s,a) \left( \sum_{a'} (|\pi(a'|s')Q_\pi(s',a') - \hat{\pi}(a'|s')Q_\pi(s',a')| \right.$$

$$+ |\hat{\pi}(a'|s')Q_\pi(s',a') - \hat{\pi}(a'|s')Q_{\hat{\pi}}(s',a')|)$$

$$\left. + \alpha \sum_{a'} (|\pi(a'|s') - \hat{\pi}(a'|s')| |\log \pi(a'|s')| + |\hat{\pi}(a'|s')| |\log \pi(a'|s') - \log \hat{\pi}(a'|s')|) \right)$$

$$= \gamma \sum_{s'} P(s'|s,a) \left( \sum_{a'} |\pi(a'|s') - \hat{\pi}(a'|s')| \cdot |Q_\pi(s',a')| + \sum_{a'} \hat{\pi}(a'|s') |Q_\pi(s',a') - Q_{\hat{\pi}}(s',a')| \right.$$

$$\left. + \alpha \sum_{a'} (|\pi(a'|s') - \hat{\pi}(a'|s')| |\log \pi(a'|s')| + |\hat{\pi}(a'|s')| |\log \pi(a'|s') - \log \hat{\pi}(a'|s')|) \right)$$

$$\leq \gamma \sum_{s'} P(s'|s,a) \left( \sum_{a'} \|\pi - \hat{\pi}\| \cdot Q_{\max} + \sum_{a'} \hat{\pi}(a'|s') \|Q_\pi - Q_{\hat{\pi}}\| \right.$$

$$\left. + \alpha \sum_{a'} (\|\pi - \hat{\pi}\| \cdot \log_{\max} \pi + \hat{\pi}(a'|s') \|\log \pi - \log \hat{\pi}\|) \right)$$

$$\leq \gamma Q_{\max} \cdot |\mathcal{A}| \cdot \|\pi - \hat{\pi}\| + \gamma \|Q_\pi - Q_{\hat{\pi}}\| + \alpha \gamma \log_{\max} \pi \cdot |\mathcal{A}| \cdot \|\pi - \hat{\pi}\| + \alpha \gamma \|\log \pi - \log \hat{\pi}\|$$

Hence, we get

$$\|Q_\pi - Q_{\hat{\pi}}\| \leq \gamma Q_{\max} \cdot |\mathcal{A}| \cdot \|\pi - \hat{\pi}\| + \gamma \|Q_\pi - Q_{\hat{\pi}}\| + \alpha \gamma \log_{\max} \pi \cdot |\mathcal{A}| \cdot \|\pi - \hat{\pi}\| + \alpha \gamma \|\log \pi - \log \hat{\pi}\|,$$

which implies

$$\|Q_\pi - Q_{\hat\pi}\| \leq \frac{\gamma \cdot |\mathcal{A}| \cdot (Q_{\max} + \alpha \log_{\max} \pi) \cdot \|\pi - \hat\pi\| + \alpha\gamma\|\log \pi - \log \hat\pi\|}{1 - \gamma}$$

By continuity of $\pi$ and $\log \pi$, for any arbitrary $\epsilon > 0$, we can find $\delta_1 > 0$ such that $\|\pi - \hat\pi\| < \delta_1$ implies $\|\pi - \hat\pi\| < \frac{(1-\gamma)\epsilon}{2\gamma \cdot |\mathcal{A}| \cdot (Q_{\max} + \alpha \log_{\max} \pi)}$ and $\delta_2 > 0$ such that $\|\pi - \hat\pi\| < \delta_2$ implies $\|\log \pi - \log \hat\pi\| < \frac{(1-\gamma)\epsilon}{2\alpha\gamma}$. Taking $\delta = \min(\delta_1, \delta_2)$, when $\|\pi - \hat\pi\| < \delta$ we get $\|Q_\pi - Q_{\hat\pi}\| < \epsilon$, which finishes the proof. $\qquad\square$

**Corollary 1.** *From Lemma A.2 we obtain that the following functions are continuous in $\pi$ :*

(1) *the state value function $V_\pi(s) = \sum_a (\pi(a|s)Q_\pi(s,a) - \pi(a|s)\log \pi(a|s))$,*

(2) *the advantage function $A_\pi(s,a) = Q_\pi(s,a) - V_\pi(s)$,*

(3) *and the expected total reward $J(\pi) = \mathbb{E}_{s \sim \rho_0}[V_\pi(s)]$.*

**Corollary 2** (Continuity in MARL). *All the results about continuity in $\pi$ extend to MARL. Policy $\pi$ can be replaced with joint policy $\boldsymbol{\pi}$; as $\boldsymbol{\pi}$ is Lipschitz-continuous in agent $i$ 's policy $\pi^i$, the above continuity results extend to continuity in $\pi^i$. Thus, we will quote them in our proofs for MARL.*

## B  EXACT CALCULATION OF MATRIX GAME

In this section, we provide an exact calculation of the matrix game in Section 3.2. The initial policies of the matrix game represent the scenario where due to the exploration and learning in the early stages, agents may have explored a locally optimal solution (in this example action $(A, A)$) and assigned a high probability to it. The matrix game aims to show that both MAPPO and HAPPO will converge towards this local optimum in expectation, while HASAC, owing to its optimization of the maximum entropy objective, has the potential to escape this local optimum and converge towards a superior solution.

We start by analyzing MAPPO. For both parameter-sharing and non-parameter-sharing versions of MAPPO, the policy is optimizing the following objective:

$$\pi_{\text{new}}^i = \arg\max_{\pi^i} \mathbb{E}_{s \sim \rho_{\boldsymbol{\pi}_{\text{old}}}, \mathbf{a} \sim \boldsymbol{\pi}_{\text{old}}} \left[ \frac{\pi^i(\mathbf{a}^i|s)}{\pi_{\text{old}}^i(\mathbf{a}^i|s)} A_{\boldsymbol{\pi}_{\text{old}}}(s, \mathbf{a}) \right], i = 1, 2 \tag{13}$$

Note that we omit ratio-clipping in this tabular case, as this is simpler and does not affect the final result. In this 2-agent single-state matrix game, $s \sim \rho_{\boldsymbol{\pi}_{\text{old}}}$ can be ignored as there is only one state. Value function $V_{\boldsymbol{\pi}_{\text{old}}}(s)$ can be calculated exactly:

$$
\begin{aligned}
V_{\boldsymbol{\pi}_{\text{old}}}(s) &= \mathbb{E}_{\mathbf{a} \sim \boldsymbol{\pi}_{\text{old}}}\left[Q_{\boldsymbol{\pi}_{\text{old}}}(s, \mathbf{a})\right] \\
&= \mathbb{E}_{\mathbf{a} \sim \boldsymbol{\pi}_{\text{old}}}\left[r(s, \mathbf{a})\right] \\
&= 0.36 \times 5 + 0.04 \times 10 + 0.04 \times 20 + 4 \times 0.12 \times (-20) + 2 \times 0.04 \times (-20) \\
&= -8.2
\end{aligned}
$$

Similarly, the advantage function $A_{\boldsymbol{\pi}_{\text{old}}}(s, \mathbf{a})$ can be calculated exactly as shown in Figure 6.

**Agent 1**

| | | A | B | C |
|---|---|---|---|---|
| **Agent 2** | A | 13.2 | -11.8 | -11.8 |
| | B | -11.8 | 18.2 | -11.8 |
| | C | -11.8 | -11.8 | 28.2 |

Figure 6: Advantage values $A_{\boldsymbol{\pi}_{\text{old}}}(s, \mathbf{a})$ for all joint actions in the first iteration.

Expanding objective 13, we get

$$
\begin{aligned}
\pi_{\text{new}}^i = \arg\max_{\pi^i} \ & (0.36 \times 13.2 + 0.12 \times (-11.8) \times 2) \times \frac{\pi^i(\mathbf{a}^i = A|s)}{0.6} \\
& + (0.12 \times (-11.8) + 0.04 \times 18.2 + 0.04 \times (-11.8)) \times \frac{\pi^i(\mathbf{a}^i = B|s)}{0.2} \\
& + (0.12 \times (-11.8) + 0.04 \times (-11.8) + 0.04 \times (28.2)) \times \frac{\pi^i(\mathbf{a}^i = C|s)}{0.2} \\
= \arg\max_{\pi^i} \ & 3.2 \times \pi^i(\mathbf{a}^i = A|s) + (-5.8) \times \pi^i(\mathbf{a}^i = B|s) + (-3.8) \times \pi^i(\mathbf{a}^i = C|s)
\end{aligned}
$$

This will encourage $\pi^i$ to assign a higher probability to action $A$ and lower probabilities to action $B$ and $C$. The updated policies will further encourage choosing action $A$ in subsequent iterations. Finally, the policies will converge to the deterministic policies $\pi^i = (1, 0, 0), i = 1, 2$.

For HAPPO, the agents update their policies sequentially. Without loss of generality, we assume they update in the order of Agent 1 and Agent 2. Agent 1 updates its policy according to objective 13 and increases the probability of action $A$. Let $\pi_{\text{new}}^1 = (p_1, p_2, p_3)$, s.t. $p_1 > 0.6, p_2 < 0.2, p_3 < 0.2, \sum_{j=1}^3 p_j = 1$. Agent 2 updates its policy according to the following objective:

$$\pi_{\text{new}}^2 = \arg\max_{\pi^2} \mathbb{E}_{s \sim \rho_{\boldsymbol{\pi}_{\text{old}}}, \mathbf{a} \sim \boldsymbol{\pi}_{\text{old}}} \left[ \frac{\pi^2(\mathrm{a}^2|\mathrm{s})}{\pi_{\text{old}}^2(\mathrm{a}^2|\mathrm{s})} \frac{\pi_{\text{new}}^1(\mathrm{a}^1|\mathrm{s})}{\pi_{\text{old}}^1(\mathrm{a}^1|\mathrm{s})} A_{\boldsymbol{\pi}_{\text{old}}}(\mathrm{s}, \mathbf{a}) \right]$$

$$= \arg\max_{\pi^2} (0.6 \times p_1 \times 13.2 + 0.6 \times p_2 \times (-11.8) + 0.6 \times p_3 \times (-11.8)) \times \frac{\pi^2(\mathrm{a}^2 = A|s)}{0.6}$$

$$+ (0.2 \times p_1 \times (-11.8) + 0.2 \times p_2 \times 18.2 + 0.2 \times p_3 \times (-11.8)) \times \frac{\pi^2(\mathrm{a}^2 = B|s)}{0.2}$$

$$+ (0.2 \times p_1 \times (-11.8) + 0.2 \times p_2 \times (-11.8) + 0.2 \times p_3 \times (28.2)) \times \frac{\pi^2(\mathrm{a}^2 = C|s)}{0.2}$$

$$= \arg\max_{\pi^2} (13.2 \times p_1 - 11.8 \times p_2 - 11.8 \times p_3) \times \pi^2(\mathrm{a}^2 = A|s)$$

$$(-11.8 \times p_1 + 18.2 \times p_2 - 11.8 \times p_3) \times \pi^2(\mathrm{a}^2 = B|s)$$

$$(-11.8 \times p_1 - 11.8 \times p_2 + 28.2 \times p_3) \times \pi^2(\mathrm{a}^2 = C|s)$$

Since $(13.2 \times p_1 - 11.8 \times p_2 - 11.8 \times p_3) > 3.2, (-11.8 \times p_1 + 18.2 \times p_2 - 11.8 \times p_3) < -5.8, (-11.8 \times p_1 - 11.8 \times p_2 + 28.2 \times p_3) < -3.8$, this will encourage assigning higher probability to action $A$ and lower probability to action $B$ and $C$. Similar to MAPPO, the updated policies will further encourage choosing action $A$ in subsequent iterations. Finally, the policies converge to $\pi^i = (1, 0, 0), i = 1, 2$.

By contrast, HASAC is possible to escape the local optimum and converge to better stochastic policies, as we show below. Without loss of generality, we assume the update order is Agent 1 and Agent 2. We first consider the update of Agent 1, who optimizes the following objective:

$$\pi_{\text{new}}^1 = \arg\max_{\pi^1} \mathbb{E}_{\mathrm{a}^1 \sim \pi^1, \mathrm{a}^2 \sim \pi_{\text{old}}^2} \left[ Q_{\boldsymbol{\pi}_{\text{old}}}(s, \mathrm{a}^1, \mathrm{a}^2) - \alpha \log \pi^1(\mathrm{a}^1|s) \right]$$

$$= \arg\max_{\pi^1} \mathbb{E}_{\mathrm{a}^1 \sim \pi^1, \mathrm{a}^2 \sim \pi_{\text{old}}^2} \left[ r(s, \mathrm{a}^1, \mathrm{a}^2) - \alpha \log \pi^1(\mathrm{a}^1|s) \right]$$

Parametrizing $\pi^1 = (p_1, p_2, p_3)$, we get

$$\pi_{\text{new}}^1 = \arg\max_{\pi^1} -5p_1 - 14p_2 - 12p_3 - \alpha p_1 \log p_1 - \alpha p_2 \log p_2 - \alpha p_3 \log p_3,$$

$$\text{s.t. } p_1 + p_2 + p_3 = 1$$

The solutions corresponding to different $\alpha$ are listed in the left part of Table 3.

| | After the first update | | | Convergent (both) | | |
|---|---|---|---|---|---|---|
| $\alpha$ | $p_1$ | $p_2$ | $p_3$ | $p_1$ | $p_2$ | $p_3$ |
| 1 | 0.9990 | 0.0001 | 0.0009 | 1.0000 | 0.0000 | 0.0000 |
| 2 | 0.9603 | 0.0107 | 0.0290 | 1.0000 | 0.0000 | 0.0000 |
| 5 | 0.7083 | 0.1171 | 0.1747 | 0.9849 | 0.0075 | 0.0076 |
| 10 | 0.5254 | 0.2136 | 0.2609 | 0.0221 | 0.0224 | 0.9555 |
| 15 | 0.4596 | 0.2522 | 0.2882 | 0.1278 | 0.1354 | 0.7368 |
| 20 | 0.4269 | 0.2722 | 0.3009 | 0.2514 | 0.2790 | 0.4697 |

Table 3: Exact calculation results for HASAC update. On the left we show the policy after the first update; on the right we show the policies of $\pi^1$ and $\pi^2$ after convergence.

As $\alpha$ increases, the policy is encouraged to be *stochastic*, trading off between stochasticity and reward maximization. With appropriate $\alpha$, the policies will be able to assign higher probabilities to action $B$

and $C$, thereby retaining the exploration of them. It then encourages more exploration of action $B$ and $C$ in subsequent updates. Finally, the convergent policies predominantly choose action $C$ while still being stochastic, as shown in the right part of Table 3.

Thus, we show the benefit of stochastic policies and that in expectation, with appropriate $\alpha$, HASAC is capable of escaping local optimum and converging to better stochastic policies. We note that the exact calculation results are generally consistent with the empirical results in Section 5.2 (this can be easily checked by computing the joint action probabilities and comparing with Table 5). An observation is that when $\alpha = 5$, theoretically it is not sufficient for escaping local optimum but empirically it already results in learning the optimal actions $(C, C)$. This may be due to the fact that in practical implementation, HASAC learns a centralized Q-function by sampling from the replay buffer, which introduces some randomness that causes slight deviations between the experimental results and the exact calculated ones.

## C    DERIVATION OF MAXENT MARL OBJECTIVE

In this section, we derive the maximum entropy objective of MARL (Equation 2) by performing variational inference in Figure 2. In structured variational inference, our objective is to approximate some distribution $p(\mathbf{y})$ with another, usually simpler distribution $q(\mathbf{y})$. In our case, we aim to approximate $p(\tau)$, given by

$$p(\tau) = \left[ p(s_1) \prod_{t=1}^{T} p(s_{t+1}|s_t, \mathbf{a}_t) \right] \exp\left( \sum_{t=1}^{T} r(s_t, \mathbf{a}_t) \right), \tag{14}$$

via the distribution

$$q(\tau) = q(s_1) \prod_{t=1}^{T} q(s_{t+1}|s_t, \mathbf{a}_t) q(\mathbf{a}_t|s_t) = q(s_1) \prod_{t=1}^{T} q(s_{t+1}|s_t, \mathbf{a}_t) \prod_{m=1}^{n} q^{i_m}\left( a_t^{i_m}|s_t \right), \tag{15}$$

where the joint policy $q(\mathbf{a}_t|s_t)$ follows a fully independent factorization $q(\mathbf{a}_t|s_t) = \prod_{m=1}^{n} q^{i_m}\left( a_t^{i_m}|s_t \right)$ as we assume that agents are independent of each other under CTDE paradigm. To avoid risk-seeking behavior, we fix the environment dynamics, *i.e.*, $q(s_1) = p(s_1)$ and $q(s_{t+1}|s_t, \mathbf{a}_t) = p(s_{t+1}|s_t, \mathbf{a}_t)$. In structured variational inference, approximate inference is performed by optimizing the variational lower bound (also called the evidence lower bound). In our case, the evidence is that $\mathcal{O}_t = 1, \forall t \in \{1, \ldots, T\}$ and the posterior is conditioned on the initial state $s_1$. The variational lower bound is given by

$$\begin{aligned}
\log p(\mathcal{O}_{1:T}) &= \log \int \int p(\mathcal{O}_{1:T}, s_{1:T}, \mathbf{a}_{1:T}) ds_{1:T} d\mathbf{a}_{1:T} \\
&= \log \int \int p(\mathcal{O}_{1:T}, s_{1:T}, \mathbf{a}_{1:T}) \frac{q(s_{1:T}, \mathbf{a}_{1:T})}{q(s_{1:T}, \mathbf{a}_{1:T})} ds_{1:T} d\mathbf{a}_{1:T} \\
&= \log \mathbb{E}_{(s_{1:T}, \mathbf{a}_{1:T}) \sim q(s_{1:T}, \mathbf{a}_{1:T})} \left[ \frac{p(\mathcal{O}_{1:T}, s_{1:T}, \mathbf{a}_{1:T})}{q(s_{1:T}, \mathbf{a}_{1:T})} \right] \\
&\geq \mathbb{E}_{(s_{1:T}, \mathbf{a}_{1:T}) \sim q(s_{1:T}, \mathbf{a}_{1:T})} \left[ \log p(\mathcal{O}_{1:T}, s_{1:T}, \mathbf{a}_{1:T}) - \log q(s_{1:T}, \mathbf{a}_{1:T}) \right],
\end{aligned}$$

where the inequality on the last line is obtained via Jensen's inequality. Substituting the definitions of $p(\tau)$ and $q(\tau)$ from Equations 14 and 15, and according to $q(s_{t+1}|s_t, \mathbf{a}_t) = p(s_{t+1}|s_t, \mathbf{a}_t)$, the bound reduces to

$$\log p(\mathcal{O}_{1:T}) \geq \mathbb{E}_{(s_{1:T}, \mathbf{a}_{1:T}) \sim q(s_{1:T}, \mathbf{a}_{1:T})} \left[ \sum_{t=1}^{T} \left( r(s_t, \mathbf{a}_t) - \sum_{m=1}^{n} \log q^{i_m}\left( a_t^{i_m}|s_t \right) \right) \right], \tag{16}$$

Optimizing this lower bound with respect to the policy $q(\mathbf{a}_t|s_t)$ corresponds exactly to the following maximum entropy objective:

$$J(\boldsymbol{\pi}) = \mathbb{E}_{s_{1:T} \sim \rho_{\boldsymbol{\pi}}^{1:T}, \mathbf{a}_{1:T} \sim \boldsymbol{\pi}} \left[ \sum_{t=1}^{T} \left( r(s_t, \mathbf{a}_t) + \alpha \sum_{i=1}^{n} \mathcal{H}\left( \pi^i(\cdot|s_t) \right) \right) \right], \tag{2}$$

where we multiply a temperature parameter $\alpha$ to the entropy term to assign relative importance to entropy and reward maximization (9).

# D PROOFS OF REPRESENTATION OF QRE

**Theorem 1 (Representation of QRE).** *A joint policy $\boldsymbol{\pi}_{QRE} \in \boldsymbol{\Pi}$ is a QRE if none of the agents can increase the maximum entropy objective (Equation 2) by unilaterally altering its policy, i.e.,*

$$\forall i \in \mathcal{N}, \forall \pi^i \in \Pi^i, J\left(\pi^i, \boldsymbol{\pi}_{QRE}^{-i}\right) \leq J\left(\boldsymbol{\pi}_{QRE}\right).$$

*Then the QRE policies are given by*

$$\forall i \in \mathcal{N}, \pi_{QRE}^i\left(a^i|s\right) := \frac{\exp\left(\alpha^{-1}\mathbb{E}_{\mathbf{a}^{-i}\sim\boldsymbol{\pi}_{QRE}^{-i}}\left[Q_{\boldsymbol{\pi}_{QRE}}\left(s, a^i, \mathbf{a}^{-i}\right)\right]\right)}{\sum_{b^i\in\mathcal{A}^i}\exp\left(\alpha^{-1}\mathbb{E}_{\mathbf{a}^{-i}\sim\boldsymbol{\pi}_{QRE}^{-i}}\left[Q_{\boldsymbol{\pi}_{QRE}}\left(s, b^i, \mathbf{a}^{-i}\right)\right]\right)}, \tag{3}$$

*where the soft Q-functions are defined as follows,*

$$Q_{\boldsymbol{\pi}}(s, \boldsymbol{a}) = r\left(s, \boldsymbol{a}\right) + \mathbb{E}_{\mathbf{a}_{1:\infty}\sim\boldsymbol{\pi}, \mathbf{s}_{1:\infty}\sim P}\left[\sum_{t=1}^{\infty}\gamma^t\left(r\left(\mathbf{s}_t, \mathbf{a}_t\right) + \alpha\sum_{i=1}^{n}\mathcal{H}\left(\pi^i\left(\cdot|\mathbf{s}_t\right)\right)\right)\Bigg|s_0 = s, \mathbf{a}_0 = \boldsymbol{a}\right]. \tag{4}$$

*Proof.* First, we consider the following constrained policy optimization problem to agent $i$: for a given state $s \in \mathcal{S}$,

$$\max_{\pi^i} \mathbb{E}_{\mathbf{a}^i\sim\pi^i, \mathbf{a}^{-i}\sim\boldsymbol{\pi}^{-i}}\left[Q_{\boldsymbol{\pi}}(s, \mathbf{a})\right] - \alpha\sum_{j=1}^{n}\sum_{a^j\in\mathcal{A}^j}\pi^j\left(a^j|s\right)\log\pi^j\left(a^j|s\right),$$

$$s.t. \sum_{a^i\in\mathcal{A}^i}\pi^i\left(a^i|s\right) = 1.$$

We consider its associated Lagrangian function as follows,

$$\mathcal{L}(\pi^i, \lambda) = \mathbb{E}_{\mathbf{a}^i\sim\pi^i, \mathbf{a}^{-i}\sim\boldsymbol{\pi}^{-i}}\left[Q_{\boldsymbol{\pi}}(s, \mathbf{a})\right] - \alpha\sum_{j=1}^{n}\sum_{a^j\in\mathcal{A}^j}\pi^j\left(a^j|s\right)\log\pi^j\left(a^j|s\right) + \lambda\big(\sum_{a^i\in\mathcal{A}^i}\pi^i\left(a^i|s\right) - 1\big)$$

$$= \sum_{\boldsymbol{a}\in\boldsymbol{\mathcal{A}}}\prod_{j=1}^{n}\pi^j\left(a^j|s\right)Q_{\boldsymbol{\pi}}(s, \boldsymbol{a}) - \alpha\sum_{j=1}^{n}\sum_{a^j\in\mathcal{A}^j}\pi^j\left(a^j|s\right)\log\pi^j\left(a^j|s\right) + \lambda\big(\sum_{a^i\in\mathcal{A}^i}\pi^i\left(a^i|s\right) - 1\big).$$

Then we consider the derivative of $\mathcal{L}(\pi^i, \lambda)$ with respect to $\pi^i$, we obtain

$$\frac{\partial\mathcal{L}(\pi^i, \lambda)}{\partial\pi^i\left(a^i|s\right)} = \sum_{\boldsymbol{a}^{-i}\in\boldsymbol{\mathcal{A}}^{-i}}\prod_{j\neq i}\pi^j\left(a^j|s\right)Q_{\boldsymbol{\pi}}(s, a^i, \boldsymbol{a}^{-i}) - \alpha\log\pi^i\left(a^i|s\right) - \alpha + \lambda.$$

Let $\frac{\partial\mathcal{L}(\pi^i, \lambda)}{\partial\pi^i(a^i|s)} = 0$, we know the optimal policy $\pi_\star^i$ satisfies the following condition,

$$\pi_\star^i\left(a^i|s\right) = \exp(\alpha^{-1}\mathbb{E}_{\mathbf{a}^{-i}\sim\boldsymbol{\pi}^{-i}}\left[Q_{\boldsymbol{\pi}}\left(s, a^i, \mathbf{a}^{-i}\right)\right])\exp(\frac{\lambda}{\alpha} - 1).$$

Furthermore, since $\sum_{a^i\in\mathcal{A}^i}\pi_\star^i\left(a^i|s\right) = 1$, we know optimal lagrange multiplier $\lambda_\star$ satisfies

$$\exp(1 - \frac{\lambda_\star}{\alpha}) = \sum_{a^i\in\mathcal{A}^i}\exp(\alpha^{-1}\mathbb{E}_{\mathbf{a}^{-i}\sim\boldsymbol{\pi}^{-i}}\left[Q_{\boldsymbol{\pi}}\left(s, a^i, \mathbf{a}^{-i}\right)\right]),$$

*i.e.,*

$$\lambda_\star = \left(1 - \log\sum_{a^i\in\mathcal{A}^i}\exp(\alpha^{-1}\mathbb{E}_{\mathbf{a}^{-i}\sim\boldsymbol{\pi}^{-i}}\left[Q_{\boldsymbol{\pi}}\left(s, a^i, \mathbf{a}^{-i}\right)\right])\right)\alpha.$$

Finally, we obtain the optimal policy as follows,

$$\pi_\star^i\left(a^i|s\right) = \frac{\exp(\alpha^{-1}\mathbb{E}_{\mathbf{a}^{-i}\sim\boldsymbol{\pi}^{-i}}\left[Q_{\boldsymbol{\pi}}\left(s, a^i, \mathbf{a}^{-i}\right)\right])}{\sum_{b^i\in\mathcal{A}^i}\exp(\alpha^{-1}\mathbb{E}_{\mathbf{a}^{-i}\sim\boldsymbol{\pi}^{-i}}\left[Q_{\boldsymbol{\pi}}\left(s, b^i, \mathbf{a}^{-i}\right)\right])}.$$

Hence, when each agent can not increase the maximum entropy objective by unilaterally changing its policy, policies take the following form,

$$\forall i \in \mathcal{N}, \pi_{\text{QRE}}^i \left(a^i | s\right) = \underset{\pi^i(\cdot | s)}{\arg\max} \, \mathbb{E}_{\mathbf{a}^i \sim \pi^i, \mathbf{a}^{-i} \sim \boldsymbol{\pi}_{\text{QRE}}^{-i}} \left[ Q_{\boldsymbol{\pi}_{\text{QRE}}}(s, \mathbf{a}) \right] - \alpha \sum_{j=1}^{n} \sum_{a^j \in \mathcal{A}^j} \pi^j \left(a^j | s\right) \log \pi^j \left(a^j | s\right)$$

$$= \pi_\star^i \left(a^i | s\right) = \frac{\exp(\alpha^{-1} \mathbb{E}_{\mathbf{a}^{-i} \sim \boldsymbol{\pi}_{\text{QRE}}^{-i}} \left[ Q_{\boldsymbol{\pi}_{\text{QRE}}} \left(s, a^i, \mathbf{a}^{-i}\right) \right])}{\sum_{b^i \in \mathcal{A}^i} \exp(\alpha^{-1} \mathbb{E}_{\mathbf{a}^{-i} \sim \boldsymbol{\pi}_{\text{QRE}}^{-i}} \left[ Q_{\boldsymbol{\pi}_{\text{QRE}}} \left(s, b^i, \mathbf{a}^{-i}\right) \right])},$$

which finishes the proof. $\square$

# E   PROOFS OF HETEROGENEOUS-AGENT SOFT POLICY ITERATION

In this section, we introduce heterogeneous-agent soft policy iteration and prove its properties of monotonic improvement and convergence to the QRE policies.

## E.1   PROOF OF JOINT SOFT POLICY EVALUATION

For *joint soft policy evaluation*, we will repeatedly apply soft Bellman operator $\Gamma_{\boldsymbol{\pi}}$ to $Q(s, \boldsymbol{a})$ until convergence, where:

$$\Gamma_{\boldsymbol{\pi}} Q(s, \boldsymbol{a}) \triangleq r(s, \boldsymbol{a}) + \gamma \mathbb{E}_{s' \sim P}\left[V\left(s'\right)\right] \tag{17}$$

$$V(s) = \mathbb{E}_{\mathbf{a} \sim \boldsymbol{\pi}}\left[Q(s, \mathbf{a}) + \alpha \sum_{i=1}^{n} \mathcal{H}\left(\pi^{i}\left(\cdot^{i}|s\right)\right)\right]. \tag{18}$$

In this way, as shown in lemma 4.1, we can get $Q_{\boldsymbol{\pi}}$ for any joint policy $\boldsymbol{\pi}$.

**Lemma 4.1** (**Joint Soft Policy Evaluation**). *Consider the soft Bellman backup operator $\Gamma_{\boldsymbol{\pi}}$ and a mapping $Q_0 : \mathcal{S} \times \mathcal{A} \to \mathbb{R}$ with $|\mathcal{A}| < \infty$, and define $Q_{k+1} = \Gamma_{\boldsymbol{\pi}} Q_k$. Then the sequence $Q_k$ will converge to the joint soft Q-function $\boldsymbol{\pi}$ as $k \to \infty$.*

*Proof.* We define the reward with entropy term as $r_{\boldsymbol{\pi}}(s, \boldsymbol{a}) \triangleq r(s, \boldsymbol{a}) + \mathbb{E}_{s' \sim P}\left[\sum_{i=1}^{n} \mathcal{H}\left(\pi^{i}\left(\cdot^{i}|s'\right)\right)\right]$. We can then express the update rule as:

$$Q(s, \boldsymbol{a}) \leftarrow r_{\boldsymbol{\pi}}(s, \boldsymbol{a}) + \gamma \mathbb{E}_{s' \sim P, \mathbf{a}' \sim \boldsymbol{\pi}}\left[Q(s', \mathbf{a}')\right]$$

and apply the standard convergence results for policy evaluation (28). $\qquad \square$

## E.2   PROOF OF JOINT SOFT POLICY DECOMPOSITION

After we get $Q_{\boldsymbol{\pi}}(s, \boldsymbol{a})$, we draw a permutation $i_{1:n} \in \mathrm{Sym}(n)$ and update each agent's policy $\pi^{i_m}$ according to the following optimization proposition:

**Proposition 1** (**Joint Soft Policy Decomposition**). *Let $\pi$ be a joint policy, and $i_{1:n} \in \mathrm{Sym}(n)$ be an agent permutation. Suppose that, for each state $s$ and every $m = 1, \dots, n$,*

$$\pi_{new}^{i_m} = \arg\min_{\pi^{i_m} \in \Pi^{i_m}} \mathrm{D}_{\mathrm{KL}}\left(\pi^{i_m}\left(\cdot^{i_m}|s\right) \left\| \frac{\exp\left(\mathbb{E}_{\mathbf{a}^{i_{1:m-1}} \sim \boldsymbol{\pi}_{new}^{i_{1:m-1}}}\left[\frac{1}{\alpha} Q_{\boldsymbol{\pi}_{old}}^{i_{1:m}}\left(s, \mathbf{a}^{i_{1:m-1}}, \cdot^{i_m}\right)\right]\right)}{\mathbb{E}_{\mathbf{a}^{i_{1:m-1}} \sim \boldsymbol{\pi}_{new}^{i_{1:m-1}}}\left[Z_{\boldsymbol{\pi}_{old}}\left(s, \mathbf{a}^{i_{1:m-1}}\right)\right]}\right), \tag{8}$$

*where $\mathbf{a}^{i_{1:m-1}}$ is drawn from the policy $\boldsymbol{\pi}_{new}^{i_{1:m-1}}(\cdot|s)$ and the partition function $Z_{\boldsymbol{\pi}_{old}}\left(s, \mathbf{a}^{i_{1:m-1}}\right)$ normalizes the distribution. Then the joint policy satisfies the following:*

$$\boldsymbol{\pi}_{new} = \arg\min_{\boldsymbol{\pi} \in \boldsymbol{\Pi}} \mathrm{D}_{\mathrm{KL}}\left(\boldsymbol{\pi}\left(\cdot|s\right) \left\| \frac{\exp\left(\frac{1}{\alpha} Q_{\boldsymbol{\pi}_{old}}\left(s, \cdot\right)\right)}{Z_{\boldsymbol{\pi}_{old}}\left(s\right)}\right). \tag{9}$$

*Proof.* First, we use $L_{\pi_{old}^{i_m}}^{i_m}\left(\pi^{i_m}(\cdot|s)\right)$ to denote

$$\mathrm{D}_{\mathrm{KL}}\left(\pi^{i_m}\left(\cdot^{i_m}|s\right) \left\| \frac{\exp\left(\mathbb{E}_{\mathbf{a}^{i_{1:m-1}} \sim \boldsymbol{\pi}_{new}^{i_{1:m-1}}}\left[\frac{1}{\alpha} Q_{\boldsymbol{\pi}_{old}}^{i_{1:m}}\left(s, \mathbf{a}^{i_{1:m-1}}, \cdot^{i_m}\right)\right]\right)}{\mathbb{E}_{\mathbf{a}^{i_{1:m-1}} \sim \boldsymbol{\pi}_{new}^{i_{1:m-1}}}\left[Z_{\boldsymbol{\pi}_{old}}\left(s, \mathbf{a}^{i_{1:m-1}}\right)\right]}\right).$$

Suppose that there exists a policy $\bar{\boldsymbol{\pi}} \neq \boldsymbol{\pi}_{new}$, such that $L_{\boldsymbol{\pi}_{old}}(\bar{\boldsymbol{\pi}}(\cdot|s)) < L_{\boldsymbol{\pi}_{old}}(\boldsymbol{\pi}_{new}(\cdot|s))$, we have

$$\mathbb{E}_{\mathbf{a} \sim \bar{\boldsymbol{\pi}}}\left[Q_{\boldsymbol{\pi}_{old}}\left(s, \mathbf{a}\right)\right] + \alpha \sum_{i=1}^{n} \mathcal{H}\left(\bar{\pi}^{i}\left(\cdot^{i}|s\right)\right) > \mathbb{E}_{\mathbf{a} \sim \boldsymbol{\pi}_{new}}\left[Q_{\boldsymbol{\pi}_{old}}(s, \mathbf{a})\right] + \alpha \sum_{i=1}^{n} \mathcal{H}\left(\pi_{new}^{i}\left(\cdot^{i}|s\right)\right). \tag{19}$$

From Equation 8, we have $L_{\pi_{old}^{i_m}}^{i_m}\left(\pi_{new}^{i_m}(\cdot|s)\right) \leq L_{\pi_{old}^{i_m}}^{i_m}\left(\bar{\pi}^{i_m}(\cdot|s)\right)$ for every $m = 1, \dots, n$, *i.e.*,

$$\mathbb{E}_{\mathbf{a}^{i_{1:m-1}} \sim \boldsymbol{\pi}_{new}^{i_{1:m-1}}, \mathbf{a}^{i_m} \sim \pi_{new}^{i_m}}\left[Q_{\boldsymbol{\pi}_{old}}^{i_{1:m}}(s, \mathbf{a}^{i_{1:m-1}}, \mathbf{a}^{i_m}) - \alpha \log \pi_{new}^{i_m}\left(\mathbf{a}^{i_m}|s\right)\right]$$
$$\geq \mathbb{E}_{\mathbf{a}^{i_{1:m-1}} \sim \boldsymbol{\pi}_{new}^{i_{1:m-1}}, \mathbf{a}^{i_m} \sim \bar{\pi}^{i_m}}\left[Q_{\boldsymbol{\pi}_{old}}^{i_{1:m}}(s, \mathbf{a}^{i_{1:m-1}}, \mathbf{a}^{i_m}) - \alpha \log \bar{\pi}^{i_m}\left(\mathbf{a}^{i_m}|s\right)\right].$$

Subtracting both sides of the inequality by $\mathbb{E}_{\mathbf{a}^{i_{1:m-1}} \sim \boldsymbol{\pi}_{\text{new}}^{i_{1:m-1}}} \left[ Q_{\boldsymbol{\pi}_{\text{old}}}^{i_{1:m-1}}(s, \mathbf{a}^{i_{1:m-1}}) \right]$ gives

$$
\begin{aligned}
&\mathbb{E}_{\mathbf{a}^{i_{1:m-1}} \sim \boldsymbol{\pi}_{\text{new}}^{i_{1:m-1}}, \mathbf{a}^{i_m} \sim \pi_{\text{new}}^{i_m}} \left[ A_{\boldsymbol{\pi}_{\text{old}}}^{i_m}(s, \mathbf{a}^{i_{1:m-1}}, \mathbf{a}^{i_m}) - \alpha \log \pi_{\text{new}}^{i_m}(\mathbf{a}^{i_m}|s) \right] \\
&\geq \mathbb{E}_{\mathbf{a}^{i_{1:m-1}} \sim \boldsymbol{\pi}_{\text{new}}^{i_{1:m-1}}, \mathbf{a}^{i_m} \sim \bar{\pi}^{i_m}} \left[ A_{\boldsymbol{\pi}_{\text{old}}}^{i_m}(s, \mathbf{a}^{i_{1:m-1}}, \mathbf{a}^{i_m}) - \alpha \log \bar{\pi}^{i_m}(\mathbf{a}^{i_m}|s) \right].
\end{aligned}
\tag{20}
$$

Combining this with Lemma A.1 gives

$$
\mathbb{E}_{\mathbf{a} \sim \boldsymbol{\pi}_{\text{new}}} \left[ A_{\boldsymbol{\pi}_{\text{old}}}(s, \mathbf{a}) + \alpha \sum_{i=1}^{n} \mathcal{H}\left( \pi_{\text{new}}^{i}(\cdot^{i}|s) \right) \right]
$$

$$
= \sum_{m=1}^{n} \left[ \mathbb{E}_{\mathbf{a}^{i_{1:m-1}} \sim \boldsymbol{\pi}_{\text{new}}^{i_{1:m-1}}, \mathbf{a}^{i_m} \sim \pi_{\text{new}}^{i_m}} \left[ A_{\boldsymbol{\pi}_{\text{old}}}^{i_m}(s, \mathbf{a}^{i_{1:m-1}}, \mathbf{a}^{i_m}) - \alpha \log \pi_{\text{new}}^{i_m}(\mathbf{a}^{i_m}|s) \right] \right]
$$

by Inequality 20

$$
\geq \sum_{m=1}^{n} \left[ \mathbb{E}_{\mathbf{a}^{i_{1:m-1}} \sim \boldsymbol{\pi}_{\text{new}}^{i_{1:m-1}}, \mathbf{a}^{i_m} \sim \bar{\pi}^{i_m}} \left[ A_{\boldsymbol{\pi}_{\text{old}}}^{i_m}(s, \mathbf{a}^{i_{1:m-1}}, \mathbf{a}^{i_m}) - \alpha \log \bar{\pi}^{i_m}(\mathbf{a}^{i_m}|s) \right] \right]
$$

$$
= \mathbb{E}_{\mathbf{a} \sim \bar{\boldsymbol{\pi}}} \left[ A_{\boldsymbol{\pi}_{\text{old}}}(s, \mathbf{a}) + \alpha \sum_{i=1}^{n} \mathcal{H}\left( \bar{\pi}^{i}(\cdot^{i}|s) \right) \right].
$$

The resulting inequality can be equivalently rewritten as

$$
\mathbb{E}_{\mathbf{a} \sim \bar{\boldsymbol{\pi}}} \left[ Q_{\boldsymbol{\pi}_{\text{old}}}(s, \mathbf{a}) \right] + \alpha \sum_{i=1}^{n} \mathcal{H}\left( \bar{\pi}^{i}(\cdot^{i}|s) \right) \leq \mathbb{E}_{\mathbf{a} \sim \boldsymbol{\pi}_{\text{new}}} \left[ Q_{\boldsymbol{\pi}_{\text{old}}}(s, \mathbf{a}) \right] + \alpha \sum_{i=1}^{n} \mathcal{H}\left( \pi_{\text{new}}^{i}(\cdot^{i}|s) \right), \tag{21}
$$

which contradicts Equation 19. Hence, for all $\boldsymbol{\pi} \in \boldsymbol{\Pi}$, $L_{\boldsymbol{\pi}_{\text{old}}}(\boldsymbol{\pi}_{\text{new}}(\cdot|s)) \leq L_{\boldsymbol{\pi}_{\text{old}}}(\boldsymbol{\pi}(\cdot|s))$, *i.e.*,

$$
\boldsymbol{\pi}_{\text{new}} = \arg\min_{\boldsymbol{\pi} \in \boldsymbol{\Pi}} \mathrm{D}_{\mathrm{KL}} \left( \boldsymbol{\pi}(\cdot|s) \, \Big\| \, \frac{\exp\left( \frac{1}{\alpha} Q_{\boldsymbol{\pi}_{\text{old}}}(s, \cdot) \right)}{Z_{\boldsymbol{\pi}_{\text{old}}}(s)} \right),
$$

which finishes the proof.

$\square$

### E.3   PROOF OF HETEROGENEOUS-AGENT SOFT POLICY IMPROVEMENT

Then the soft Q-function and joint maximum entropy objective (Equation 2) has monotonic improvement property as formalized below.

**Lemma 4.2 (Heterogeneous-Agent Soft Policy Improvement).** *Let $i_{1:n} \in \mathrm{Sym}(n)$ be an agent permutation, and for every $m = 1, \dots, n$, let policy $\pi_{old}^{i_m} \in \Pi^{i_m}$ and $\pi_{new}^{i_m}$ be the optimizer of the minimization problem defined in Equation 8. Then $Q_{\boldsymbol{\pi}_{new}}(s, \boldsymbol{a}) \geq Q_{\boldsymbol{\pi}_{old}}(s, \boldsymbol{a})$ for all $(s, \boldsymbol{a}) \in \mathcal{S} \times \boldsymbol{\mathcal{A}}$ with $|\boldsymbol{\mathcal{A}}| < \infty$ and $J(\boldsymbol{\pi}_{new}) \geq J(\boldsymbol{\pi}_{old})$.*

*Proof.* From Proposition 1, we have

$$
\boldsymbol{\pi}_{\text{new}} = \arg\min_{\boldsymbol{\pi} \in \boldsymbol{\Pi}} \mathrm{D}_{\mathrm{KL}} \left( \boldsymbol{\pi}(\cdot|s) \, \Big\| \, \frac{\exp\left( \frac{1}{\alpha} Q_{\boldsymbol{\pi}_{\text{old}}}(s, \cdot) \right)}{Z_{\boldsymbol{\pi}_{\text{old}}}(s)} \right) = \arg\min_{\boldsymbol{\pi} \in \boldsymbol{\Pi}} L_{\boldsymbol{\pi}_{\text{old}}}(\boldsymbol{\pi}(\cdot|s)).
$$

It must be the case that $L_{\boldsymbol{\pi}_{\text{old}}}(\boldsymbol{\pi}_{\text{new}}(\cdot|s)) \leq L_{\boldsymbol{\pi}_{\text{old}}}(\boldsymbol{\pi}_{\text{old}}(\cdot|s))$. Hence

$$
\mathbb{E}_{\mathbf{a} \sim \boldsymbol{\pi}_{\text{new}}} \left[ Q_{\boldsymbol{\pi}_{\text{old}}}(s, \mathbf{a}) \right] + \alpha \sum_{i=1}^{n} \mathcal{H}\left( \pi_{\text{new}}^{i}(\cdot^{i}|s) \right) \geq \mathbb{E}_{\mathbf{a} \sim \boldsymbol{\pi}_{\text{old}}} \left[ Q_{\boldsymbol{\pi}_{\text{old}}}(s, \mathbf{a}) \right] + \alpha \sum_{i=1}^{n} \mathcal{H}\left( \pi_{\text{old}}^{i}(\cdot^{i}|s) \right) = V_{\boldsymbol{\pi}_{\text{old}}}(s).
$$

$$
\tag{22}
$$

Last, considering the soft Bellman equation, the following holds:

$$
\begin{aligned}
Q_{\boldsymbol{\pi}_{\text{old}}}(s, \boldsymbol{a}) &= r(s, \boldsymbol{a}) + \gamma \mathbb{E}_{s' \sim P}\left[V_{\boldsymbol{\pi}_{\text{old}}}(s')\right] \\
&\leq r(s, \boldsymbol{a}) + \gamma \mathbb{E}_{s' \sim P}\left[\mathbb{E}_{\mathbf{a} \sim \boldsymbol{\pi}_{\text{new}}}\left[Q_{\boldsymbol{\pi}_{\text{old}}}(s', \mathbf{a})\right] + \alpha \sum_{i=1}^{n} \mathcal{H}\left(\pi_{\text{new}}^{i}\left(\cdot^{i}|s'\right)\right)\right] \quad \text{(by Inequality 22)} \\
&\vdots \\
&\leq Q_{\boldsymbol{\pi}_{\text{new}}}(s, \boldsymbol{a}),
\end{aligned}
\tag{23}
$$

where we have repeatedly expanded $Q_{\boldsymbol{\pi}_{\text{old}}}$ on the RHS by applying the soft Bellman equation and the bound in Inequality 22. Convergence to $Q_{\boldsymbol{\pi}_{\text{new}}}$ follows from Lemma 4.1.

We use it to prove the claim as follows,

$$
\begin{aligned}
V_{\boldsymbol{\pi}_{\text{new}}}(s) &= \mathbb{E}_{\mathbf{a} \sim \boldsymbol{\pi}_{\text{new}}}\left[Q_{\boldsymbol{\pi}_{\text{new}}}(s, \mathbf{a})\right] + \alpha \sum_{i=1}^{n} \mathcal{H}\left(\pi_{\text{new}}^{i}\left(\cdot^{i}|s\right)\right) \\
&\geq \mathbb{E}_{\mathbf{a} \sim \boldsymbol{\pi}_{\text{new}}}\left[Q_{\boldsymbol{\pi}_{\text{old}}}(s, \mathbf{a})\right] + \alpha \sum_{i=1}^{n} \mathcal{H}\left(\pi_{\text{new}}^{i}\left(\cdot^{i}|s\right)\right) \text{(by Inequality 23)} \\
&\geq \mathbb{E}_{\mathbf{a} \sim \boldsymbol{\pi}_{\text{old}}}\left[Q_{\boldsymbol{\pi}_{\text{old}}}(s, \mathbf{a})\right] + \alpha \sum_{i=1}^{n} \mathcal{H}\left(\pi_{\text{old}}^{i}\left(\cdot^{i}|s\right)\right) \text{(by Inequality 22)} \\
&= V_{\boldsymbol{\pi}_{\text{old}}}(s).
\end{aligned}
$$

Subsequently, the monotonic improvement property of the joint maximum entropy return follows naturally, as

$$
J\left(\boldsymbol{\pi}_{\text{new}}\right) = \mathbb{E}_{s \sim d}\left[V_{\boldsymbol{\pi}_{\text{new}}}(s)\right] \geq \mathbb{E}_{s \sim d}\left[V_{\boldsymbol{\pi}_{\text{old}}}(s)\right] = J\left(\boldsymbol{\pi}_{\text{old}}\right).
$$

$\square$

### E.4 PROOF OF **HETEROGENEOUS-AGENT SOFT POLICY ITERATION**

The full heterogeneous-agent soft policy iteration algorithm alternates between the joint soft policy evaluation and the heterogeneous-agent soft policy improvement steps, and it will provably converge to the QRE policies.

**Theorem 2** (**Heterogeneous-Agent Soft Policy Iteration**). *For any joint policy $\boldsymbol{\pi} \in \boldsymbol{\Pi}$, if we repeatedly apply joint soft policy evaluation and heterogeneous-agent soft policy improvement from $\pi^{i} \in \Pi^{i}$. Then the joint policy $\boldsymbol{\pi} = \prod_{i=1}^{n} \pi^{i}$ converges to $\boldsymbol{\pi}_{QRE}$ in Theorem 1.*

*Proof.* Let $\boldsymbol{\pi}_k$ be the joint policy at iteration $k$.

First, by Lemma 4.2, we have that $Q_{\boldsymbol{\pi}_k}(s, \boldsymbol{a}) \leq Q_{\boldsymbol{\pi}_{k+1}}(s, \boldsymbol{a})$, and that the soft Q-function is upper-bounded by $Q_{\text{max}}$ for all $\boldsymbol{\pi} \in \boldsymbol{\Pi}$ (both reward and entropy are bounded). Hence, the sequence converges to some limit point $\bar{\boldsymbol{\pi}}$.

Then, considering this limit point joint policy $\bar{\boldsymbol{\pi}}$, it must be the case that

$$
\forall i \in \mathcal{N}, \forall \pi^{i} \in \Pi^{i}, L_{\bar{\pi}^{i}}^{i}(\bar{\pi}^{i}(\cdot|s)) \leq L_{\bar{\pi}^{i}}^{i}(\pi^{i}(\cdot|s)).
$$

And we have

$$
\begin{aligned}
\bar{\pi}^{i}\left(\cdot^{i}|s\right) &= \underset{\pi^{i}(\cdot^{i}|s) \in \mathcal{P}(\mathcal{A}^{i})}{\arg\max} \mathbb{E}_{\mathbf{a}^{i} \sim \pi^{i}}\left[Q_{\bar{\boldsymbol{\pi}}}^{i}\left(s, \mathbf{a}^{i}\right) - \alpha \log \pi^{i}\left(\mathbf{a}^{i}|s\right)\right] \\
&= \underset{\pi^{i}(\cdot^{i}|s) \in \mathcal{P}(\mathcal{A}^{i})}{\arg\max} \mathbb{E}_{\mathbf{a}^{i} \sim \pi^{i}, \mathbf{a}^{-i} \sim \bar{\boldsymbol{\pi}}^{-i}}\left[Q_{\bar{\boldsymbol{\pi}}}(s, \mathbf{a})\right] \\
&\quad - \alpha \sum_{j=1}^{n} \sum_{a^{j} \in \mathcal{A}^{j}} \pi^{j}\left(a^{j}|s\right) \log \pi^{j}\left(a^{j}|s\right), \forall i \in \mathcal{N}.
\end{aligned}
\tag{24}
$$

Last, following the proof of Theorem 1, we have

$$
\bar{\pi}^{i}\left(a^{i}|s\right) := \frac{\exp\left(\alpha^{-1} \mathbb{E}_{\mathbf{a}^{-i} \sim \bar{\boldsymbol{\pi}}^{-i}}\left[Q_{\bar{\boldsymbol{\pi}}}\left(s, a^{i}, \mathbf{a}^{-i}\right)\right]\right)}{\sum_{b^{i} \in \mathcal{A}^{i}} \exp\left(\alpha^{-1} \mathbb{E}_{\mathbf{a}^{-i} \sim \bar{\boldsymbol{\pi}}^{-i}}\left[Q_{\bar{\boldsymbol{\pi}}}\left(s, b^{i}, \mathbf{a}^{-i}\right)\right]\right)}.
$$

Thus, $\bar{\boldsymbol{\pi}}$ is a quantal response equilibrium, which finishes the proof.

$\square$

# F  PSEUDOCODE OF HASAC

---

**Algorithm 2:** Heterogeneous-Agent Soft Actor-Critic

---

**Input:** temperature $\alpha$, Polyak coefficient $\tau$, batch size $B$, number of: agents $n$, episodes $K$,
    steps per episode $T$, mini-epochs $e$;

**Initialize:** the critic networks: $\phi_1$ and $\phi_2$ and policy networks: $\{\theta^i\}_{i \in \hat{\mathcal{N}}}$, replay buffer $\mathcal{B}$, Set
  target parameters equal to main parameters $\phi_{\text{targ, 1}} \leftarrow \phi_1, \phi_{\text{targ, 2}} \leftarrow \phi_2$;

**for** $k = 0, 1, \ldots, K-1$ **do**

  Observe state $o_t^i$ and select action $a_t^i \sim \pi_\theta^i(\cdot|o_t^i)$;

  Execute $a_t^i$ in the environment;

  Observe next state $o_{t+1}$, reward $r_t$;

  Push transitions $\left\{ \left( o_t^i, a_t^i, o_{t+1}^i, r_t \right), \forall i \in \mathcal{N}, t \in T \right\}$ into $\mathcal{B}$;

  Sample a random minibatch of $B$ transitions from $\mathcal{B}$;

  Compute the critic targets

$$y_t = r + \gamma \left( \min_{i=1,2} Q_{\phi_{\text{targ}, i}} \left( s_{t+1}, \mathbf{a}_{t+1} \right) - \alpha \sum_{i=1}^{n} \log \pi_\theta^i \left( a_{t+1}^i | o_{t+1}^i \right) \right), \quad \mathbf{a}_{t+1} \sim \boldsymbol{\pi}_{\boldsymbol{\theta}} \left( \cdot | s_{t+1} \right);$$

  Update Q-functions by one step of gradient descent using

$$\phi_i = \arg \min_{\phi_i} \frac{1}{B} \sum_t \left( y_t - Q_{\phi_i} \left( s_t, \mathbf{a}_t \right) \right)^2 \quad \text{for } i = 1, 2;$$

  Draw a permutation of agents $i_{1:n}$ at random;

  **for** *agent* $i_m = i_1, \ldots, i_n$ **do**

   Update agent $i_m$ by solving

$$\theta_{\text{new}}^{i_m} = \arg \max_{\hat{\theta}^{i_m}} \frac{1}{B} \sum_t \left( \min_{i=1,2} Q_{\phi_i} \left( s_t, \mathbf{a}_{\boldsymbol{\theta}_{\text{new}}^{i_{1:m-1}}}^{i_{1:m-1}} \left( \mathbf{o}_t^{i_{1:m-1}} \right), a_{\hat{\theta}^{i_m}}^{i_m} \left( o_t^{i_m} \right), \mathbf{a}_{\boldsymbol{\theta}_{\text{old}}^{i_{m+1:n}}}^{i_{m+1:n}} \left( \mathbf{o}_t^{i_{m+1:n}} \right) \right) \right.$$

$$\left. - \alpha \log \pi_{\hat{\theta}^{im}}^{i_m} \left( a_{\hat{\theta}^{im}}^{i_m} | o_t^{i_m} \right) \right)$$

   where $a_\theta^i(o_t^i)$ is a sample from $\pi_\theta^i(\cdot|o_t^i)$ which is differentiable wrt $\theta$ via the
   reparametrization trick;

   with $e$ mini-epochs of policy gradient ascent;

  **end**

  Update the target critic network smoothly

$$\phi_{\text{targ}, i} \leftarrow \rho \phi_{\text{targ}, i} + (1 - \rho) \phi_i \quad \text{for } i = 1, 2;$$

**end**

Discard $\phi$. Deploy $\{\theta^i\}_{i \in \mathcal{N}}$ in execution;

---

## G  PROOFS OF THE CORE THEOREM OF MEHAML

First, we show that enhancing the MEHAMO (Definition 2) alone is sufficient to guarantee policy improvement, as demonstrated by the following lemma.

**Lemma G.1.** *Let $\boldsymbol{\pi}_{old}$ and $\boldsymbol{\pi}_{new}$ be joint policies and let $i_{1:n} \in \mathrm{Sym}(n)$ be an agent permutation. Suppose that, for every state $s \in \mathcal{S}$ and every $m = 1, \ldots, n$,*

$$\left[ \mathcal{M}^{(\pi_{new}^{i_m})}_{\mathfrak{D}^{i_m}, \boldsymbol{\pi}_{new}^{i_{1:m-1}}} V_{\boldsymbol{\pi}_{old}} \right](s) \geq \left[ \mathcal{M}^{(\pi_{old}^{i_m})}_{\mathfrak{D}^{i_m}, \boldsymbol{\pi}_{new}^{i_{1:m-1}}} V_{\boldsymbol{\pi}_{old}} \right](s) \tag{25}$$

*Then, $\boldsymbol{\pi}_{new}$ is jointly better than $\boldsymbol{\pi}_{old}$, so that for every state $s$,*

$$V_{\boldsymbol{\pi}_{new}}(s) \geq V_{\boldsymbol{\pi}_{old}}(s).$$

*Subsequently, the monotonic improvement property of the joint return follows naturally, as*

$$J(\boldsymbol{\pi}_{new}) = \mathbb{E}_{\mathrm{s} \sim d}[V_{\boldsymbol{\pi}_{new}}(\mathrm{s})] \geq \mathbb{E}_{\mathrm{s} \sim d}[V_{\boldsymbol{\pi}_{old}}(\mathrm{s})] = J(\boldsymbol{\pi}_{old}).$$

*Proof.* By Inequality 25, we have

$$\mathbb{E}_{\mathbf{a}^{i_{1:m-1}} \sim \boldsymbol{\pi}_{new}^{i_{1:m-1}}, a^{i_m} \sim \pi_{new}^{i_m}} \left[ Q_{\boldsymbol{\pi}_{old}}^{i_{1:m}}(s, \mathbf{a}^{i_{1:m-1}}, a^{i_m}) - \alpha \log \pi_{new}^{i_m}(a^{i_m}|s) \right]$$

$$- \mathfrak{D}_{\boldsymbol{\pi}_{old}}^{i_m} \left( \pi_{new}^{i_m}|s, \boldsymbol{\pi}_{new}^{i_{1:m-1}} \right)$$

$$\geq \mathbb{E}_{\mathbf{a}^{i_{1:m-1}} \sim \boldsymbol{\pi}_{new}^{i_{1:m-1}}, a^{i_m} \sim \pi_{old}^{i_m}} \left[ Q_{\boldsymbol{\pi}_{old}}^{i_{1:m}}(s, \mathbf{a}^{i_{1:m-1}}, a^{i_m}) - \alpha \log \pi_{old}^{i_m}(a^{i_m}|s) \right]$$

$$- \mathfrak{D}_{\boldsymbol{\pi}_{old}}^{i_m} \left( \pi_{old}^{i_m}|s, \boldsymbol{\pi}_{new}^{i_{1:m-1}} \right).$$

Subtracting both sides of the inequality by $\mathbb{E}_{\mathbf{a}^{i_{1:m-1}} \sim \boldsymbol{\pi}_{new}^{i_{1:m-1}}} \left[ Q_{\boldsymbol{\pi}_{old}}^{i_{1:m-1}}(s, \mathbf{a}^{i_{1:m-1}}) \right]$ gives

$$\mathbb{E}_{\mathbf{a}^{i_{1:m-1}} \sim \boldsymbol{\pi}_{new}^{i_{1:m-1}}, a^{i_m} \sim \pi_{new}^{i_m}} \left[ A_{\boldsymbol{\pi}_{old}}^{i_m}(s, \mathbf{a}^{i_{1:m-1}}, a^{i_m}) - \alpha \log \pi_{new}^{i_m}(a^{i_m}|s) \right]$$

$$- \mathfrak{D}_{\boldsymbol{\pi}_{old}}^{i_m} \left( \pi_{new}^{i_m}|s, \boldsymbol{\pi}_{new}^{i_{1:m-1}} \right)$$

$$\geq \mathbb{E}_{\mathbf{a}^{i_{1:m-1}} \sim \boldsymbol{\pi}_{new}^{i_{1:m-1}}, a^{i_m} \sim \pi_{old}^{i_m}} \left[ A_{\boldsymbol{\pi}_{old}}^{i_m}(s, \mathbf{a}^{i_{1:m-1}}, a^{i_m}) - \alpha \log \pi_{old}^{i_m}(a^{i_m}|s) \right] \tag{26}$$

$$- \mathfrak{D}_{\boldsymbol{\pi}_{old}}^{i_m} \left( \pi_{old}^{i_m}|s, \boldsymbol{\pi}_{new}^{i_{1:m-1}} \right).$$

Let $\widetilde{\mathfrak{D}}_{\boldsymbol{\pi}_{old}}(\boldsymbol{\pi}_{new}|s) \triangleq \sum_{m=1}^{n} \mathfrak{D}_{\boldsymbol{\pi}_{old}}^{i_m} \left( \pi_{new}^{i_m}|s, \boldsymbol{\pi}_{new}^{i_{1:m-1}} \right)$. Combining this with Lemma A.1 gives

$$\mathbb{E}_{\mathbf{a} \sim \boldsymbol{\pi}_{new}} \left[ A_{\boldsymbol{\pi}_{old}}(s, \mathbf{a}) + \alpha \sum_{i=1}^{n} \mathcal{H} \left( \pi_{new}^i(\cdot^i|s) \right) \right] - \widetilde{\mathfrak{D}}_{\boldsymbol{\pi}_{old}}(\boldsymbol{\pi}_{new}|s)$$

$$= \sum_{m=1}^{n} \left[ \mathbb{E}_{\mathbf{a}^{i_{1:m-1}} \sim \boldsymbol{\pi}_{new}^{i_{1:m-1}}, a^{i_m} \sim \pi_{new}^{i_m}} \left[ A_{\boldsymbol{\pi}_{old}}^{i_m}(s, \mathbf{a}^{i_{1:m-1}}, a^{i_m}) - \alpha \log \pi_{new}^{i_m}(a^{i_m}|s) \right] \right.$$

$$\left. - \mathfrak{D}_{\boldsymbol{\pi}_{old}}^{i_m} \left( \pi_{new}^{i_m}|s, \boldsymbol{\pi}_{new}^{i_{1:m-1}} \right) \right]$$

by Inequality 20

$$\geq \sum_{m=1}^{n} \left[ \mathbb{E}_{\mathbf{a}^{i_{1:m-1}} \sim \boldsymbol{\pi}_{new}^{i_{1:m-1}}, a^{i_m} \sim \pi_{old}^{i_m}} \left[ A_{\boldsymbol{\pi}_{old}}^{i_m}(s, \mathbf{a}^{i_{1:m-1}}, a^{i_m}) - \alpha \log \pi_{old}^{i_m}(a^{i_m}|s) \right] \right.$$

$$\left. - \mathfrak{D}_{\boldsymbol{\pi}_{old}}^{i_m} \left( \pi_{old}^{i_m}|s, \boldsymbol{\pi}_{new}^{i_{1:m-1}} \right) \right]$$

$$= \mathbb{E}_{\mathbf{a} \sim \boldsymbol{\pi}_{old}} \left[ A_{\boldsymbol{\pi}_{old}}(s, \mathbf{a}) + \alpha \sum_{i=1}^{n} \mathcal{H} \left( \pi_{old}^i(\cdot^i|s) \right) \right] - \widetilde{\mathfrak{D}}_{\boldsymbol{\pi}_{old}}(\boldsymbol{\pi}_{old}|s).$$

The resulting inequality can be equivalently rewritten as

$$
\mathbb{E}_{\mathbf{a}\sim\boldsymbol{\pi}_{\mathrm{new}}} \left[ Q_{\boldsymbol{\pi}_{\mathrm{old}}} (s, \mathbf{a}) \right] + \alpha \sum_{i=1}^{n} \mathcal{H} \left( \pi^{i}_{\mathrm{new}} \left( \cdot^{i}|s \right) \right) - \widetilde{\mathfrak{D}}_{\boldsymbol{\pi}_{\mathrm{old}}} \left( \boldsymbol{\pi}_{\mathrm{new}} | s \right)
$$

$$
\geq \mathbb{E}_{\mathbf{a}\sim\boldsymbol{\pi}_{\mathrm{old}}} \left[ Q_{\boldsymbol{\pi}_{\mathrm{old}}} (s, \mathbf{a}) \right] + \alpha \sum_{i=1}^{n} \mathcal{H} \left( \pi^{i}_{\mathrm{old}} \left( \cdot^{i}|s \right) \right) - \widetilde{\mathfrak{D}}_{\boldsymbol{\pi}_{\mathrm{old}}} \left( \boldsymbol{\pi}_{\mathrm{old}} | s \right), \forall s \in \mathcal{S}. \tag{27}
$$

We use it to prove the claim as follows,

$$
V_{\boldsymbol{\pi}_{\mathrm{new}}} (s) = \mathbb{E}_{\mathbf{a}\sim\boldsymbol{\pi}_{\mathrm{new}}} \left[ Q_{\boldsymbol{\pi}_{\mathrm{new}}} (s, \mathbf{a}) \right] + \alpha \sum_{i=1}^{n} \mathcal{H} \left( \pi^{i}_{\mathrm{new}} \left( \cdot^{i}|s \right) \right)
$$

$$
= \mathbb{E}_{\mathbf{a}\sim\boldsymbol{\pi}_{\mathrm{new}}} \left[ Q_{\boldsymbol{\pi}_{\mathrm{old}}} (s, \mathbf{a}) \right] + \alpha \sum_{i=1}^{n} \mathcal{H} \left( \pi^{i}_{\mathrm{new}} \left( \cdot^{i}|s \right) \right) - \widetilde{\mathfrak{D}}_{\boldsymbol{\pi}_{\mathrm{old}}} \left( \boldsymbol{\pi}_{\mathrm{new}} | s \right)
$$

$$
+ \widetilde{\mathfrak{D}}_{\boldsymbol{\pi}_{\mathrm{old}}} \left( \boldsymbol{\pi}_{\mathrm{new}} | s \right) + \mathbb{E}_{\mathbf{a}\sim\boldsymbol{\pi}_{\mathrm{new}}} \left[ Q_{\boldsymbol{\pi}_{\mathrm{new}}} (s, \mathbf{a}) - Q_{\boldsymbol{\pi}_{\mathrm{old}}} (s, \mathbf{a}) \right],
$$
by Inequality 27

$$
\geq \mathbb{E}_{\mathbf{a}\sim\boldsymbol{\pi}_{\mathrm{old}}} \left[ Q_{\boldsymbol{\pi}_{\mathrm{old}}} (s, \mathbf{a}) \right] + \alpha \sum_{i=1}^{n} \mathcal{H} \left( \pi^{i}_{\mathrm{old}} \left( \cdot^{i}|s \right) \right) - \widetilde{\mathfrak{D}}_{\boldsymbol{\pi}_{\mathrm{old}}} \left( \boldsymbol{\pi}_{\mathrm{old}} | s \right)
$$

$$
+ \widetilde{\mathfrak{D}}_{\boldsymbol{\pi}_{\mathrm{old}}} \left( \boldsymbol{\pi}_{\mathrm{new}} | s \right) + \mathbb{E}_{\mathbf{a}\sim\boldsymbol{\pi}_{\mathrm{new}}} \left[ Q_{\boldsymbol{\pi}_{\mathrm{new}}} (s, \mathbf{a}) - Q_{\boldsymbol{\pi}_{\mathrm{old}}} (s, \mathbf{a}) \right],
$$

$$
= V_{\boldsymbol{\pi}_{\mathrm{old}}} (s) + \widetilde{\mathfrak{D}}_{\boldsymbol{\pi}_{\mathrm{old}}} \left( \boldsymbol{\pi}_{\mathrm{new}} | s \right) + \mathbb{E}_{\mathbf{a}\sim\boldsymbol{\pi}_{\mathrm{new}}} \left[ Q_{\boldsymbol{\pi}_{\mathrm{new}}} (s, \mathbf{a}) - Q_{\boldsymbol{\pi}_{\mathrm{old}}} (s, \mathbf{a}) \right]
$$

$$
= V_{\boldsymbol{\pi}_{\mathrm{old}}} (s) + \widetilde{\mathfrak{D}}_{\boldsymbol{\pi}_{\mathrm{old}}} \left( \boldsymbol{\pi}_{\mathrm{new}} | s \right) + \mathbb{E}_{\mathbf{a}\sim\boldsymbol{\pi}_{\mathrm{new}}, \mathbf{s}'\sim P} \left[ r(s, \mathbf{a}) + \gamma V_{\boldsymbol{\pi}_{\mathrm{new}}} (\mathbf{s}') - r(s, \mathbf{a}) - \gamma V_{\boldsymbol{\pi}_{\mathrm{old}}} (\mathbf{s}') \right]
$$

$$
= V_{\boldsymbol{\pi}_{\mathrm{old}}} (s) + \widetilde{\mathfrak{D}}_{\boldsymbol{\pi}_{\mathrm{old}}} \left( \boldsymbol{\pi}_{\mathrm{new}} | s \right) + \gamma \mathbb{E}_{\mathbf{a}\sim\boldsymbol{\pi}_{\mathrm{new}}, \mathbf{s}'\sim P} \left[ V_{\boldsymbol{\pi}_{\mathrm{new}}} (\mathbf{s}') - V_{\boldsymbol{\pi}_{\mathrm{old}}} (\mathbf{s}') \right]
$$

$$
\geq V_{\boldsymbol{\pi}_{\mathrm{old}}} (s) + \gamma \inf_{\mathbf{s}'} \left[ V_{\boldsymbol{\pi}_{\mathrm{new}}} (\mathbf{s}') - V_{\boldsymbol{\pi}_{\mathrm{old}}} (\mathbf{s}') \right].
$$

Hence $\quad V_{\boldsymbol{\pi}_{\mathrm{new}}} (s) - V_{\boldsymbol{\pi}_{\mathrm{old}}} (s) \geq \gamma \inf_{\mathbf{s}'} \left[ V_{\boldsymbol{\pi}_{\mathrm{new}}} (\mathbf{s}') - V_{\boldsymbol{\pi}_{\mathrm{old}}} (\mathbf{s}') \right].$

Taking infimum over s and simplifying

$(1 - \gamma) \inf_{s} \left[ V_{\boldsymbol{\pi}_{\mathrm{new}}} (s) - V_{\boldsymbol{\pi}_{\mathrm{old}}} (s) \right] \geq 0$

Therefore, $\inf_{s} \left[ V_{\boldsymbol{\pi}_{\mathrm{new}}} (s) - V_{\boldsymbol{\pi}_{\mathrm{old}}} (s) \right] \geq 0$, which proves the lemma. $\qquad\square$

Then, any algorithm derived from Algorithm 1 ensures that the resulting policies satisfy Condition 25, as demonstrated by the following lemma.

**Lemma G.2.** *Suppose an agent $i_m$ maximizes the expected MEHAMO*

$$
\pi^{i_m}_{new} = \underset{\pi^{i_m} \in \mathcal{U}^{i_m}_{\boldsymbol{\pi}_{old}} \left( \pi^{i_m}_{old} \right)}{\arg\max} \mathbb{E}_{s\sim\beta_{\boldsymbol{\pi}_{old}}} \left[ \left[ \mathcal{M}^{\left( \pi^{i_m} \right)}_{\mathfrak{D}^{i_m}, \boldsymbol{\pi}^{i_{1:m-1}}_{new}} V_{\boldsymbol{\pi}_{old}} \right] (s) \right]. \tag{28}
$$

*Then, for every state $s \in \mathcal{S}$*

$$
\left[ \mathcal{M}^{\left( \pi^{i_m}_{new} \right)}_{\mathfrak{D}^{i_m}, \boldsymbol{\pi}^{i_{1:m-1}}_{new}} V_{\boldsymbol{\pi}_{old}} \right] (s) \geq \left[ \mathcal{M}^{\left( \pi^{i_m}_{old} \right)}_{\mathfrak{D}^{i_m}, \boldsymbol{\pi}^{i_{1:m-1}}_{new}} V_{\boldsymbol{\pi}_{old}} \right] (s). \tag{29}
$$

*Hence, $\boldsymbol{\pi}_{new}$ attains the properties provided by Lemma G.1.*

*Proof.* We will prove this statement by contradiction. Suppose that there exists $s_0 \in \mathcal{S}$ such that

$$
\left[ \mathcal{M}^{\left( \pi^{i_m}_{new} \right)}_{\mathfrak{D}^{i_m}, \boldsymbol{\pi}^{i_{1:m-1}}_{new}} V_{\boldsymbol{\pi}_{old}} \right] (s_0) < \left[ \mathcal{M}^{\left( \pi^{i_m}_{old} \right)}_{\mathfrak{D}^{i_m}, \boldsymbol{\pi}^{i_{1:m-1}}_{new}} V_{\boldsymbol{\pi}_{old}} \right] (s_0).
$$

Let us define the following policy $\hat{\pi}^{i_m}$.

$$
\hat{\pi}^{i_m} \left( \cdot^{i_m}|s \right) = \begin{cases} \pi^{i_m}_{old} \left( \cdot^{i_m}|s \right), & \text{at } s = s_0 \\ \pi^{i_m}_{new} \left( \cdot^{i_m}|s \right), & \text{at } s \neq s_0 \end{cases}
$$

Note that $\hat{\pi}^{i_m}$ is (weakly) closer to $\pi_{\text{old}}^{i_m}$ than $\pi_{\text{new}}^{i_m}$ at $s_0$, and at the same distance at other states. Together with $\pi_{\text{new}}^{i_m} \in \mathcal{U}_{\pi_{\text{old}}}^{i_m}\left(\pi_{\text{old}}^{i_m}\right)$, this implies that $\hat{\pi}^{i_m} \in \mathcal{U}_{\pi_{\text{old}}}^{i_m}\left(\pi_{\text{old}}^{i_m}\right)$. Further,

$$\mathbb{E}_{\text{s}\sim\beta_{\boldsymbol{\pi}_{\text{old}}}}\left[\left[\mathcal{M}_{\mathfrak{D}^{i_m},\boldsymbol{\pi}_{\text{new}}^{i_{1:m-1}}}^{(\hat{\pi}^{i_m})} V_{\boldsymbol{\pi}_{\text{old}}}\right](\text{s})\right] - \mathbb{E}_{\text{s}\sim\beta_{\boldsymbol{\pi}_{\text{old}}}}\left[\left[\mathcal{M}_{\mathfrak{D}^{i_m},\boldsymbol{\pi}_{\text{new}}^{i_{1:m-1}}}^{(\pi_{\text{new}}^{i_m})} V_{\boldsymbol{\pi}_{\text{old}}}\right](\text{s})\right]$$

$$= \beta_{\boldsymbol{\pi}_{\text{old}}}\left(s_0\right)\left(\left[\mathcal{M}_{\mathfrak{D}^{i_m},\boldsymbol{\pi}_{\text{new}}^{i_{1:m-1}}}^{(\hat{\pi}^{i_m})} V_{\boldsymbol{\pi}_{\text{old}}}\right](s_0) - \left[\mathcal{M}_{\mathfrak{D}^{i_m},\boldsymbol{\pi}_{\text{new}}^{i_{1:m-1}}}^{(\pi_{\text{new}}^{i_m})} V_{\boldsymbol{\pi}_{\text{old}}}\right](s_0)\right) > 0.$$

The above contradicts $\pi_{\text{new}}^{i_m}$ as being the argmax of Equality 28, as $\hat{\pi}^{i_m}$ is strictly better. The contradiction finishes the proof. $\qquad\square$

Next, we prove the most fundamental theorem of MEHAML.

**Theorem 3 (The Core Theorem of MEHAML).** *Let $\boldsymbol{\pi}_0 \in \Pi$, and the sequence of joint policies $(\boldsymbol{\pi}_k)_{k=0}^{\infty}$ be obtained by a MEHAML algorithm 1 induced by $\mathfrak{D}^i, \mathcal{U}^i, \forall i \in \mathcal{N}$, and $\beta_{\boldsymbol{\pi}}$. Then, the joint policies induced by the algorithm enjoy the following list of properties (1) Attain the monotonic improvement property $J\left(\boldsymbol{\pi}_{k+1}\right) \geq J\left(\boldsymbol{\pi}_k\right)$, (2) Their value functions converge to a quantal response value function $\lim_{k\to\infty} V_{\boldsymbol{\pi}_k} = V^{QRE}$, (3) Their expected returns converge to a quantal response return $\lim_{k\to\infty} J\left(\boldsymbol{\pi}_k\right) = J^{QRE}$, (4) Their $\omega$-limit set consists of quantal response equilibria.*

*Proof.* **Proof of Property 1.**
It follows from combining Lemma G.1 & G.2.

**Proof of Properties 2, 3 & 4.**

**Step 1: convergence of the value function.** By Lemma G.1, we have that $V_{\boldsymbol{\pi}_k}(s) \leq V_{\boldsymbol{\pi}_{k+1}}(s), \forall s \in \mathcal{S}$, and that the value function is upper-bounded by $V_{\max}$. Hence, the sequence of value functions $(V_{\boldsymbol{\pi}_k})_{k\in\mathbb{N}}$ converges. We denote its limit by $V$.

**Step 2: characterisation of limit points.** As the joint policy space $\Pi$ is bounded, by Bolzano-Weierstrass theorem, we know that the sequence $(\boldsymbol{\pi}_k)_{k\in\mathbb{N}}$ has a convergent subsequence. Therefore, it has at least one limit point policy. Let $\bar{\boldsymbol{\pi}}$ be such a limit point. We introduce an auxiliary notation: for a joint policy $\boldsymbol{\pi}$ and a permutation $i_{1:n}$, let $\text{HU}\left(\boldsymbol{\pi}, i_{1:n}\right)$ be a joint policy obtained by a MEHAML update from $\boldsymbol{\pi}$ along the permutation $i_{1:n}$.

**Claim:** For any permutation $z_{1:n} \in \text{Sym}(n)$,

$$\bar{\boldsymbol{\pi}} = \text{HU}\left(\bar{\boldsymbol{\pi}}, z_{1:n}\right)$$

**Proof of Claim.** Let $\hat{\boldsymbol{\pi}} = \text{HU}\left(\bar{\boldsymbol{\pi}}, z_{1:n}\right) \neq \bar{\boldsymbol{\pi}}$ and $(\boldsymbol{\pi}_{k_r})_{r\in\mathbb{N}}$ be a subsequence converging to $\bar{\boldsymbol{\pi}}$. Let us recall that the limit value function is unique and denoted as $V$. Writing $\mathbb{E}_{i_{1:n}^{0:\infty}}[\cdot]$ for the expectation operator under the stochastic process $(i_{1:n}^k)_{k\in\mathbb{N}}$ of update orders, for a state $s \in \mathcal{S}$, we have

$$0 = \lim_{r\to\infty} \mathbb{E}_{i_{1:n}^{0:\infty}}\left[V_{\boldsymbol{\pi}_{k_r+1}}(s) - V_{\boldsymbol{\pi}_{k_r}}(s)\right]$$

as every choice of permutation improves the value function

$$\geq \lim_{r\to\infty} \text{P}\left(i_{1:n}^{k_r} = z_{1:n}\right)\left[V_{\text{HU}(\boldsymbol{\pi}_{k_r},z_{1:n})}(s) - V_{\boldsymbol{\pi}_{k_r}}(s)\right]$$

$$= p\left(z_{1:n}\right) \lim_{r\to\infty}\left[V_{\text{HU}(\boldsymbol{\pi}_{k_r},z_{1:n})}(s) - V_{\boldsymbol{\pi}_{k_r}}(s)\right].$$

By the continuity of the expected MEHAMO (following from the continuity of the state-action value function (Lemma A.2), the entropy term, HADFs, neighbourhood operators, and the sampling distribution) we obtain that the first component of $\text{HU}\left(\boldsymbol{\pi}_{k_r}, z_{1:n}\right)$, which is $\pi_{k_r+1}^{z_1}$, is continuous in $\boldsymbol{\pi}_{k_r}$ by Berge's Maximum Theorem (1). Applying this argument recursively for $z_2, \ldots, z_n$, we have that $\text{HU}\left(\boldsymbol{\pi}_{k_r}, z_{1:n}\right)$ is continuous in $\boldsymbol{\pi}_{k_r}$. Hence, as $\boldsymbol{\pi}_{k_r}$ converges to $\bar{\boldsymbol{\pi}}$, its HU converges to the HU of $\bar{\boldsymbol{\pi}}$, which is $\hat{\boldsymbol{\pi}}$. Hence, we continue writing the above derivation as

$$= p\left(z_{1:n}\right)\left[V_{\hat{\boldsymbol{\pi}}}(s) - V_{\bar{\boldsymbol{\pi}}}(s)\right] \geq 0, \text{ by Lemma G.1.}$$

As $s$ was arbitrary, the state-value function of $\hat{\boldsymbol{\pi}}$ is the same as that of $\boldsymbol{\pi}$: $V_{\hat{\boldsymbol{\pi}}} = V_{\bar{\boldsymbol{\pi}}}$, by the Bellman equation 5: $Q(s, \boldsymbol{a}) = r(s, \boldsymbol{a}) + \gamma\mathbb{E}\left[V\left(s'\right)\right]$, this also implies that their state-action value functions

are the same: $Q_{\hat{\boldsymbol{\pi}}} = Q_{\bar{\boldsymbol{\pi}}}$. Let $m$ be the smallest integer such that $\hat{\pi}^{z_m} \neq \bar{\pi}^{z_m}$. This means that $\hat{\pi}^{z_m}$ achieves a greater expected MEHAMO than $\bar{\pi}^{z_m}$. Hence,

$$\mathbb{E}_{s \sim \beta_{\bar{\boldsymbol{\pi}}}}\left[\left[\mathcal{M}_{\mathfrak{D}^{z_m}, \bar{\boldsymbol{\pi}}^{z_{1:m-1}}}^{(\hat{\pi}^{z_m})} V_{\bar{\boldsymbol{\pi}}}\right](\mathrm{s})\right] > \mathbb{E}_{s \sim \beta_{\bar{\boldsymbol{\pi}}}}\left[\left[\mathcal{M}_{\mathfrak{D}^{z_m}, \bar{\boldsymbol{\pi}}^{z_{1:m-1}}}^{(\bar{\pi}^{z_m})} V_{\bar{\boldsymbol{\pi}}}\right](\mathrm{s})\right]$$

then for some state $s$,

$$\left[\mathcal{M}_{\mathfrak{D}^{z_m}, \bar{\boldsymbol{\pi}}^{z_{1:m-1}}}^{(\hat{\pi}^{z_m})} V_{\bar{\boldsymbol{\pi}}}\right](s) > \left[\mathcal{M}_{\mathfrak{D}^{z_m}, \bar{\boldsymbol{\pi}}^{z_{1:m-1}}}^{(\bar{\pi}^{z_m})} V_{\bar{\boldsymbol{\pi}}}\right](s)$$

which can be written as

$$\mathbb{E}_{\mathbf{a}^{z_{1:m-1}} \sim \bar{\boldsymbol{\pi}}^{z_{1:m-1}}, \mathbf{a}^{z_m} \sim \hat{\pi}^{z_m}}\left[Q_{\bar{\boldsymbol{\pi}}}^{z_{1:m}}(s, \mathbf{a}^{z_{1:m-1}}, \mathbf{a}^{z_m}) - \alpha \log \hat{\pi}^{z_m}(\mathbf{a}^{z_m}|s)\right] - \mathfrak{D}_{\bar{\boldsymbol{\pi}}}^{z_m}(\hat{\pi}^{z_m}|s, \bar{\boldsymbol{\pi}}^{z_{1:m-1}})$$

$$= \mathbb{E}_{\mathbf{a}^{z_{1:m-1}} \sim \bar{\boldsymbol{\pi}}^{z_{1:m-1}}, \mathbf{a}^{z_m} \sim \hat{\pi}^{z_m}}\left[Q_{\hat{\boldsymbol{\pi}}}^{z_{1:m}}(s, \mathbf{a}^{z_{1:m-1}}, \mathbf{a}^{z_m}) - \alpha \log \hat{\pi}^{z_m}(\mathbf{a}^{z_m}|s)\right] - \mathfrak{D}_{\bar{\boldsymbol{\pi}}}^{z_m}(\hat{\pi}^{z_m}|s, \bar{\boldsymbol{\pi}}^{z_{1:m-1}})$$

$$> \mathbb{E}_{\mathbf{a}^{z_{1:m-1}} \sim \bar{\boldsymbol{\pi}}^{z_{1:m-1}}, \mathbf{a}^{z_m} \sim \bar{\pi}^{z_m}}\left[Q_{\bar{\boldsymbol{\pi}}}^{z_{1:m}}(s, \mathbf{a}^{z_{1:m-1}}, \mathbf{a}^{z_m}) - \alpha \log \bar{\pi}^{z_m}(\mathbf{a}^{z_m}|s)\right] - \mathfrak{D}_{\bar{\boldsymbol{\pi}}}^{z_m}(\bar{\pi}^{z_m}|s, \bar{\boldsymbol{\pi}}^{z_{1:m-1}})$$

$$= \mathbb{E}_{\mathbf{a}^{z_{1:m-1}} \sim \bar{\boldsymbol{\pi}}^{z_{1:m-1}}, \mathbf{a}^{z_m} \sim \bar{\pi}^{z_m}}\left[Q_{\bar{\boldsymbol{\pi}}}^{z_{1:m}}(s, \mathbf{a}^{z_{1:m-1}}, \mathbf{a}^{z_m}) - \alpha \log \bar{\pi}^{z_m}(\mathbf{a}^{z_m}|s)\right].$$

Adding both sides of the inequality by $\alpha \sum_{i=1}^{m-1} \mathcal{H}\left(\bar{\pi}^{z_i}(\cdot|s)\right)$ and using the equation $V_{\boldsymbol{\pi}}(s) = \mathbb{E}_{\mathbf{a} \sim \boldsymbol{\pi}}\left[Q_{\boldsymbol{\pi}}(s, \mathbf{a}) + \alpha \sum_{i=1}^{n} \mathcal{H}\left(\pi^i(\cdot|s)\right)\right]$ gives

$$V_{\hat{\boldsymbol{\pi}}}(s) = \mathbb{E}_{\mathbf{a} \sim \hat{\boldsymbol{\pi}}}\left[Q_{\hat{\boldsymbol{\pi}}}(s, \mathbf{a}) + \alpha \sum_{i=1}^{n} \mathcal{H}\left(\hat{\pi}^i(\cdot|s)\right)\right]$$

$$\geq \mathbb{E}_{\mathbf{a} \sim \hat{\boldsymbol{\pi}}}\left[Q_{\hat{\boldsymbol{\pi}}}(s, \mathbf{a}) + \alpha \sum_{i=1}^{n} \mathcal{H}\left(\hat{\pi}^i(\cdot|s)\right)\right] - \mathfrak{D}_{\bar{\boldsymbol{\pi}}}^{z_m}(\hat{\pi}^{z_m}|s, \bar{\boldsymbol{\pi}}^{z_{1:m-1}})$$

$$> \mathbb{E}_{\mathbf{a} \sim \bar{\boldsymbol{\pi}}}\left[Q_{\bar{\boldsymbol{\pi}}}(s, \mathbf{a}) + \alpha \sum_{i=1}^{n} \mathcal{H}\left(\bar{\pi}^i(\cdot|s)\right)\right]$$

$$= V_{\bar{\boldsymbol{\pi}}}(s).$$

However, we have $V_{\hat{\boldsymbol{\pi}}} = V_{\boldsymbol{\pi}}$ which yields a contradiction, proving the claim.

**Step 3: dropping the HADF.** Consider an arbitrary limit point joint policy $\bar{\boldsymbol{\pi}}$. By Step 2, for any permutation $i_{1:n}$, considering the first component of the HU,

$$\bar{\pi}^{i_1} = \underset{\pi^{i_1} \in \mathcal{U}_{\bar{\boldsymbol{\pi}}}^{i_1}(\bar{\pi}^{i_1})}{\arg\max} \; \mathbb{E}_{s \sim \beta_{\bar{\boldsymbol{\pi}}}}\left[\left[\mathcal{M}_{\mathfrak{D}^{i_1}}^{(\pi^{i_1})} V_{\bar{\boldsymbol{\pi}}}\right](\mathrm{s})\right]$$

$$= \underset{\pi^{i_1} \in \mathcal{U}_{\bar{\boldsymbol{\pi}}}^{i_1}(\bar{\pi}^{i_1})}{\arg\max} \; \mathbb{E}_{s \sim \beta_{\bar{\boldsymbol{\pi}}}}\left[\mathbb{E}_{\mathbf{a}^{i_1} \sim \pi^{i_1}}\left[Q_{\bar{\boldsymbol{\pi}}}^{i_1}(\mathrm{s}, \mathbf{a}^{i_1}) - \alpha \log \pi^{i_1}(\mathbf{a}^{i_1}|\mathrm{s})\right] - \mathfrak{D}_{\bar{\boldsymbol{\pi}}}^{i_1}(\pi^{i_1}|\mathrm{s})\right]$$

$$= \underset{\pi^{i_1} \in \mathcal{U}_{\bar{\boldsymbol{\pi}}}^{i_1}(\bar{\pi}^{i_1})}{\arg\max} \; \mathbb{E}_{s \sim \beta_{\bar{\boldsymbol{\pi}}}}\left[\mathbb{E}_{\mathbf{a}^{i_1} \sim \pi^{i_1}}\left[A_{\bar{\boldsymbol{\pi}}}^{i_1}(\mathrm{s}, \mathbf{a}^{i_1}) - \alpha \log \pi^{i_1}(\mathbf{a}^{i_1}|\mathrm{s})\right] - \mathfrak{D}_{\bar{\boldsymbol{\pi}}}^{i_1}(\pi^{i_1}|\mathrm{s})\right].$$

Suppose that there exists a policy $\pi' \neq \bar{\pi}^{i_1}$, and a state $s$, such that

$$\pi' = \underset{\pi^{i_1} \in \mathcal{U}_{\bar{\boldsymbol{\pi}}}^{i_1}(\bar{\pi}^{i_1})}{\arg\max} \; \mathbb{E}_{\mathbf{a}^{i_1} \sim \pi^{i_1}}\left[A_{\bar{\boldsymbol{\pi}}}^{i_1}(s, \mathbf{a}^{i_1}) - \alpha \log \pi^{i_1}(\mathbf{a}^{i_1}|s)\right], \tag{30}$$

implies

$$\mathbb{E}_{\mathbf{a}^{i_1} \sim \pi'}\left[A_{\bar{\boldsymbol{\pi}}}^{i_1}(s, \mathbf{a}^{i_1}) - \alpha \log \pi'(\mathbf{a}^{i_1}|s)\right] > \mathbb{E}_{\mathbf{a}^{i_1} \sim \bar{\pi}^{i_1}}\left[A_{\bar{\boldsymbol{\pi}}}^{i_1}(s, \mathbf{a}) - \alpha \log \bar{\pi}^{i_1}(\mathbf{a}^{i_1}|s)\right]$$

which can be written as

$$\mathbb{E}_{\mathbf{a}^{i_1} \sim \pi'}\left[A_{\bar{\boldsymbol{\pi}}}^{i_1}(s, \mathbf{a}^{i_1})\right] + \alpha \mathcal{H}\left(\pi'(\cdot^{i_1}|s)\right) > \alpha \mathcal{H}\left(\bar{\pi}^{i_1}(\cdot^{i_1}|s)\right).$$

For any policy $\pi^{i_1}$, consider the canonical parameterisation $\pi^{i_1}(\cdot^{i_1}|s) = \left(x_1, \ldots, x_{m-1}, 1 - \sum_{i=1}^{m-1} x_i\right)$, where $m$ is the size of the action space. We have that

$$\mathbb{E}_{\mathrm{a}^{i_1} \sim \pi^{i_1}}\left[A_{\bar{\pi}}^{i_1}(s, \mathrm{a}^{i_1})\right] = \sum_{i=1}^{m} \pi^{i_1}\left(a_i^{i_1}|s\right) A_{\bar{\pi}}^{i_1}\left(s, a_i^{i_1}\right)$$

$$= \sum_{i=1}^{m-1} x_i A_{\bar{\pi}}^{i_1}\left(s, a_i^{i_1}\right) + \left(1 - \sum_{j=1}^{m-1} x_j\right) A_{\bar{\pi}}^{i_1}\left(s, a_m^{i_1}\right)$$

$$= \sum_{i=1}^{m-1} x_i \left[A_{\bar{\pi}}^{i_1}\left(s, a_i^{i_1}\right) - A_{\bar{\pi}}^{i_1}\left(s, a_m^{i_1}\right)\right] + A_{\bar{\pi}}^{i_1}\left(s, a_m^{i_1}\right).$$

This means that $\mathbb{E}_{\mathrm{a}^{i_1} \sim \pi^{i_1}}\left[A_{\bar{\pi}}^{i_1}(s, \mathrm{a}^{i_1})\right]$ is an affine function of $\pi^{i_1}(\cdot^{i_1}|s)$, and thus, its Gâteaux derivatives are constant in $\mathcal{P}(\mathcal{A})$ for fixed directions. Hence, we can obtain that $\mathbb{E}_{\mathrm{a}^{i_1} \sim \pi^{i_1}}\left[A_{\bar{\pi}}^{i_1}(s, \mathrm{a}^{i_1})\right] + \alpha \mathcal{H}\left(\pi^{i_1}\left(\cdot^{i_1}|s\right)\right)$ is a strict concave function of $\pi^{i_1}(\cdot^{i_1}|s)$ (following from the affinity of $\mathbb{E}_{\mathrm{a}^{i_1} \sim \pi^{i_1}}\left[A_{\bar{\pi}}^{i_1}(s, \mathrm{a}^{i_1})\right]$ and the strict concavity of $\mathcal{H}\left(\pi^{i_1}\left(\cdot^{i_1}|s\right)\right)$). Therefore, by combining the Equation 30 and the strict concavity of $\mathbb{E}_{\mathrm{a}^{i_1} \sim \pi^{i_1}}\left[A_{\bar{\pi}}^{i_1}(s, \mathrm{a}^{i_1})\right] + \alpha \mathcal{H}\left(\pi^{i_1}\left(\cdot^{i_1}|s\right)\right)$, Gâteaux derivative of $\mathbb{E}_{\mathrm{a}^{i_1} \sim \pi^{i_1}}\left[A_{\bar{\pi}}^{i_1}(s, \mathrm{a}^{i_1})\right] + \alpha \mathcal{H}\left(\pi^{i_1}\left(\cdot^{i_1}|s\right)\right)$, in the direction from $\bar{\pi}$ to $\pi'$, is strictly positive.

Furthermore, the Gâteaux derivatives of $\mathfrak{D}_{\bar{\pi}}^{i_1}(\pi^{i_1}|s)$ are zero at $\pi^{i_1}(\cdot^{i_1}|s) = \bar{\pi}^{i_1}(\cdot^{i_1}|s)$ by its definition (zero gradient). Hence, the Gâteaux derivative of $\mathbb{E}_{\mathrm{a}^{i_1} \sim \pi^{i_1}}\left[A_{\bar{\pi}}^{i_1}(s, \mathrm{a})\right] + \alpha \mathcal{H}\left(\pi^{i_1}\left(\cdot^{i_1}|s\right)\right) - \mathfrak{D}_{\bar{\pi}}^{i_1}\left(\pi^{i_1}|s\right)$ is strictly positive. Therefore, for conditional policies $\hat{\pi}^{i_1}(\cdot^{i_1}|s)$ sufficiently close to $\bar{\pi}^{i_1}(\cdot^{i_1}|s)$ in the direction towards $\pi'(\cdot^{i_1}|s)$, we have

$$\mathbb{E}_{\mathrm{a}^{i_1} \sim \hat{\pi}^{i_1}}\left[A_{\bar{\pi}}^{i_1}(s, \mathrm{a}^{i_1})\right] + \alpha \mathcal{H}\left(\hat{\pi}^{i_1}\left(\cdot^{i_1}|s\right)\right) - \mathfrak{D}_{\bar{\pi}}^{i_1}(\hat{\pi}^{i_1}|s)$$

$$> \mathbb{E}_{\mathrm{a}^{i_1} \sim \bar{\pi}^{i_1}}\left[A_{\bar{\pi}}^{i_1}(s, \mathrm{a}^{i_1})\right] + \alpha \mathcal{H}\left(\bar{\pi}^{i_1}\left(\cdot^{i_1}|s\right)\right) - \mathfrak{D}_{\bar{\pi}}^{i_1}(\bar{\pi}^{i_1}|s). \tag{31}$$

Let us construct a policy $\tilde{\pi}^{i_1}$ as follows. For all states $y \neq s$, we set $\tilde{\pi}^{i_1}(\cdot^{i_1}|y) = \bar{\pi}^{i_1}(\cdot^{i_1}|y)$. Moreover, for $\tilde{\pi}^{i_1}(\cdot^{i_1}|s)$ we choose $\hat{\pi}^{i_1}(\cdot^{i_1}|s)$ as in Inequality 31, sufficiently close to $\bar{\pi}^{i_1}(\cdot^{i_1}|s)$, so that $\tilde{\pi}^{i_1} \in \mathcal{U}_{\bar{\pi}^{i_1}}^{i_1}\left(\bar{\pi}^{i_1}\right)$. Then, we have

$$\mathbb{E}_{\mathrm{s} \sim \beta_{\bar{\pi}}, \mathrm{a}^{i_1} \sim \tilde{\pi}^{i_1}}\left[A_{\bar{\pi}}^{i_1}(\mathrm{s}, \mathrm{a}^{i_1})\right] + \mathbb{E}_{\mathrm{s} \sim \beta_{\bar{\pi}}}\left[\alpha \mathcal{H}\left(\tilde{\pi}^{i_1}\left(\cdot^{i_1}|\mathrm{s}\right)\right) - \mathfrak{D}_{\bar{\pi}}^{i_1}(\tilde{\pi}^{i_1}|\mathrm{s})\right]$$

$$> \mathbb{E}_{\mathrm{s} \sim \beta_{\bar{\pi}}, \mathrm{a}^{i_1} \sim \bar{\pi}^{i_1}}\left[A_{\bar{\pi}}^{i_1}(\mathrm{s}, \mathrm{a}^{i_1})\right] + \mathbb{E}_{\mathrm{s} \sim \beta_{\bar{\pi}}}\left[\alpha \mathcal{H}\left(\bar{\pi}^{i_1}\left(\cdot^{i_1}|\mathrm{s}\right)\right) - \mathfrak{D}_{\bar{\pi}}^{i_1}(\bar{\pi}^{i_1}|\mathrm{s})\right],$$

which yields a contradiction. Hence, the assumption was false. Thus, we have proved that, for every state $s$,

$$\bar{\pi}^{i_1}\left(\cdot^{i_1}|s\right) = \underset{\pi^{i_1} \in \mathcal{U}_{\bar{\pi}}^{i_1}(\bar{\pi}^{i_1})}{\arg\max} \ \mathbb{E}_{\mathrm{a}^{i_1} \sim \pi^{i_1}}\left[A_{\bar{\pi}}^{i_1}\left(s, \mathrm{a}^{i_1}\right) - \alpha \log \pi^{i_1}\left(a^{i_1}|s\right)\right]$$

$$= \underset{\pi^{i_1} \in \mathcal{U}_{\bar{\pi}}^{i_1}(\bar{\pi}^{i_1})}{\arg\max} \ \mathbb{E}_{\mathrm{a}^{i_1} \sim \pi^{i_1}}\left[Q_{\bar{\pi}}^{i_1}\left(s, \mathrm{a}^{i_1}\right) - \alpha \log \pi^{i_1}\left(a^{i_1}|s\right)\right].$$

**Step 4: Quantal response equilibrium.** We have proved that $\bar{\pi}$ satisfies

$$\bar{\pi}^i\left(\cdot^i|s\right) = \underset{\pi^i(\cdot^i|s) \in \mathcal{P}(\mathcal{A}^i)}{\arg\max} \ \mathbb{E}_{\mathrm{a}^i \sim \pi^i}\left[Q_{\bar{\pi}}^i\left(s, \mathrm{a}^i\right) - \alpha \log \pi^i\left(\mathrm{a}^i|s\right)\right]$$

$$= \underset{\pi^i(\cdot^i|s) \in \mathcal{P}(\mathcal{A}^i)}{\arg\max} \ \mathbb{E}_{\mathrm{a}^i \sim \pi^i, \mathbf{a}^{-i} \sim \bar{\pi}^{-i}}\left[Q_{\bar{\pi}}(s, \mathbf{a})\right]$$

$$- \alpha \sum_{j=1}^{n} \sum_{a^j \in \mathcal{A}^j} \pi^j\left(a^j|s\right) \log \pi^j\left(a^j|s\right), \forall i \in \mathcal{N}, s \in \mathcal{S}.$$

Then following the proof of Theorem 1, we have

$$\bar{\pi}^i\left(a^i|s\right) := \frac{\exp\left(\alpha^{-1}\mathbb{E}_{\mathbf{a}^{-i} \sim \bar{\pi}^{-i}}\left[Q_{\bar{\pi}}\left(s, a^i, \mathbf{a}^{-i}\right)\right]\right)}{\sum_{b^i \in \mathcal{A}^i} \exp\left(\alpha^{-1}\mathbb{E}_{\mathbf{a}^{-i} \sim \bar{\pi}^{-i}}\left[Q_{\bar{\pi}}\left(s, b^i, \mathbf{a}^{-i}\right)\right]\right)}.$$

Thus, $\bar{\pi}$ is a quantal response equilibrium. Lastly, this implies that the value function corresponds to a quantal response value function $V^{\mathrm{QRE}}$, the return corresponds to a quantal response return $J^{\mathrm{QRE}}$.

$\square$

# H EXPERIMENTAL DETAILS

## H.1 AUTOMATICALLY ADJUSTING TEMPERATURE

We implement an automated temperature tuning method for HASAC which draws on the auto-tuned temperature extension of SAC (9). Thus, we adjust $\alpha$ with the following objective:

$$J(\alpha) = \mathbb{E}_{\mathrm{s}_t \sim \mathcal{D}, \mathbf{a}_t \sim \boldsymbol{\pi}}[-\alpha \log \boldsymbol{\pi}(\mathbf{a}_t|\mathrm{s}_t) - \alpha \bar{\mathcal{H}}],$$

where $\bar{\mathcal{H}}$ is the target entropy.

## H.2 EXPERIMENTAL SETUP AND ADDITIONAL RESULTS

### H.2.1 BI-DEXHANDS

Bi-DexHands (2) offers numerous bimanual manipulation tasks that are designed to match various human skill levels. Building on the Isaac Gym simulator, Bi-DexHands supports running thousands of environments simultaneously. This increases the number of samples generated in the same time interval, thus significantly alleviating the sample efficiency problem of on-policy algorithms.

We evaluate HASAC on nine tasks simulating human behaviors across different ages and compare it with on-policy algorithms HAPPO, MAPPO, PPO, and off-policy algorithms HATD3 and SAC, as shown in Figure 7. Despite leveraging GPU parallelization, we observe that the sample efficiency of on-policy algorithms remains significantly lower than that of off-policy algorithms. In the most challenging Catch Abreast, Two Catch Underarm, and Lift Underarm tasks, on-policy algorithms fail to learn useful policies within 10m steps. While the off-policy algorithm HATD3 demonstrates higher sample efficiency compared to on-policy algorithms, it exhibits substantial variance during training due to the deterministic policies it learned, eventually converging to local optima. In contrast, our algorithm HASAC outperforms the other five methods by a large margin, showcasing faster convergence speed and lower variance.

### H.2.2 MULTI-AGENT MUJOCO

MuJoCo tasks challenge a robot to learn an optimal way of motion; Multi-Agent MuJoCo (MAMu-JoCo) models each part of a robot as an independent agent, for example, a leg for a spider or an arm for a swimmer. With the increasing variety of body parts, improving each agent's exploration becomes necessary. Although the easier tasks with fewer agents can be solved by a wide range of different algorithms, the more complex benchmarks, such as the HumanoidStandup 17x1 and ManyAgentSwimmer 10x2, are difficult to solve with current MARL algorithms.

We compare our method to several algorithms that show the current state-of-the-art performance in 10 tasks of 5 scenarios in MAMuJoCo, including HAPPO, a sequential-update on-policy algorithm; MAPPO, a simultaneous-update on-policy algorithm; and HATD3 (45), an off-policy algorithm that outperforms HADDPG and MADDPG. Figure 8 demonstrates that, in all scenarios, HASAC enjoys superior performance over the three rivals both in terms of reward values and learning speed.

### H.2.3 STARCRAFT MULTI-AGENT CHALLENGE

StarCraft Multi-Agent Challenge (SMAC) (25) contains a set of StarCraft maps in which a team of ally units aims to defeat the opponent team. Notably, MAPPO (42), HAPPO, and HATRPO have demonstrated remarkable performance on this benchmark through the utilization of five influential factors that significantly impact algorithm performance.

We evaluate our method on four hard maps and four super-hard. Our experimental results, illustrated in Figure 9, reveal that HASAC achieves over 90% win rates in 7 out of 8 maps and significantly outperforms other strong baselines in most maps. Notably, in particularly challenging tasks such as 5m_vs_6m, 3s5z_vs_3s5z, and 6h_vs_8z, we observe that HAPPO and HATRPO would converge towards suboptimal NE. In addition, FOP is unable to learn meaningful joint policy in super-hard tasks due to its reliance on the IGO assumption. By contrast, HASAC consistently achieves superior performance and shows the ability to identify higher reward equilibria due to its extensive exploration. We also observe that HASAC has better stability and higher learning speed across most maps.

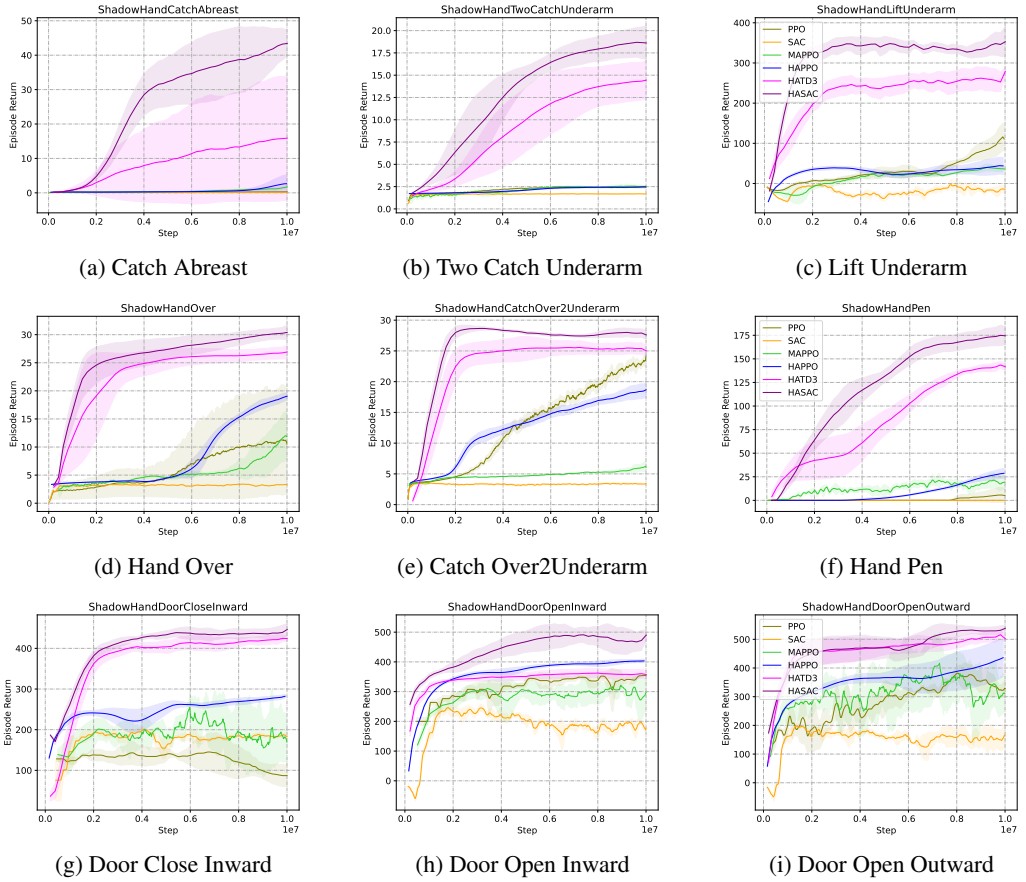

Figure 7: Comparisons of average episode return on nine Bi-DexHands tasks.

### H.2.4 GOOGLE RESEARCH FOOTBALL

Google Research Football Environment (GRF) contains a set of cooperative multi-agent challenges in which a team of agents plays a team of bots in various football scenarios. Recent works (42; 45) have conducted experiments on academic scenarios and achieved nearly 100% winning rate on each scenario except two very challenging tasks: run pass and shoot with keeper (RPS) and corner. We apply HASAC to these two academy tasks of GRF, with QMIX and several SOTA methods, including HAPPO and MAPPO as baselines. Since GRF lacks a global state interface, we propose a solution to address this limitation by implementing a global state based on agents' observations following the Simple115StateWrapper of GRF. Concretely, the global state consists of common components in agents' observations and the concatenation of agent-specific parts and is taken as input by the centralized critic for value prediction. Additionally, we employ the dense-reward setting. All methods are trained for 20 million environment steps in the RPS task and for 25 million environment steps in the corner task.

As shown in Figure 10a and 10b, we generally observe that both MAPPO and HAPPO tend to converge to a non-optimal NE on the two challenging tasks with a winning rate of approximately 80%. This suboptimal convergence can be attributed to the insufficient level of exploration of these algorithms. In contrast, HASAC exhibits the ability to attain a higher reward equilibrium by learning stochastic policies, which effectively enhance exploration and robustness. This finding highlights the crucial role of stochastic policies in improving exploration, thereby enabling agents to converge toward a higher reward equilibrium.

### H.2.5 MULTI-AGENT PARTICLE ENVIRONMENT

We evaluate HASAC on the Spread, Reference, and Speaker_Listener tasks of the Multi-Agent Particle Environment (MPE) (19), which were implemented in PettingZoo (31). PettingZoo incorporates

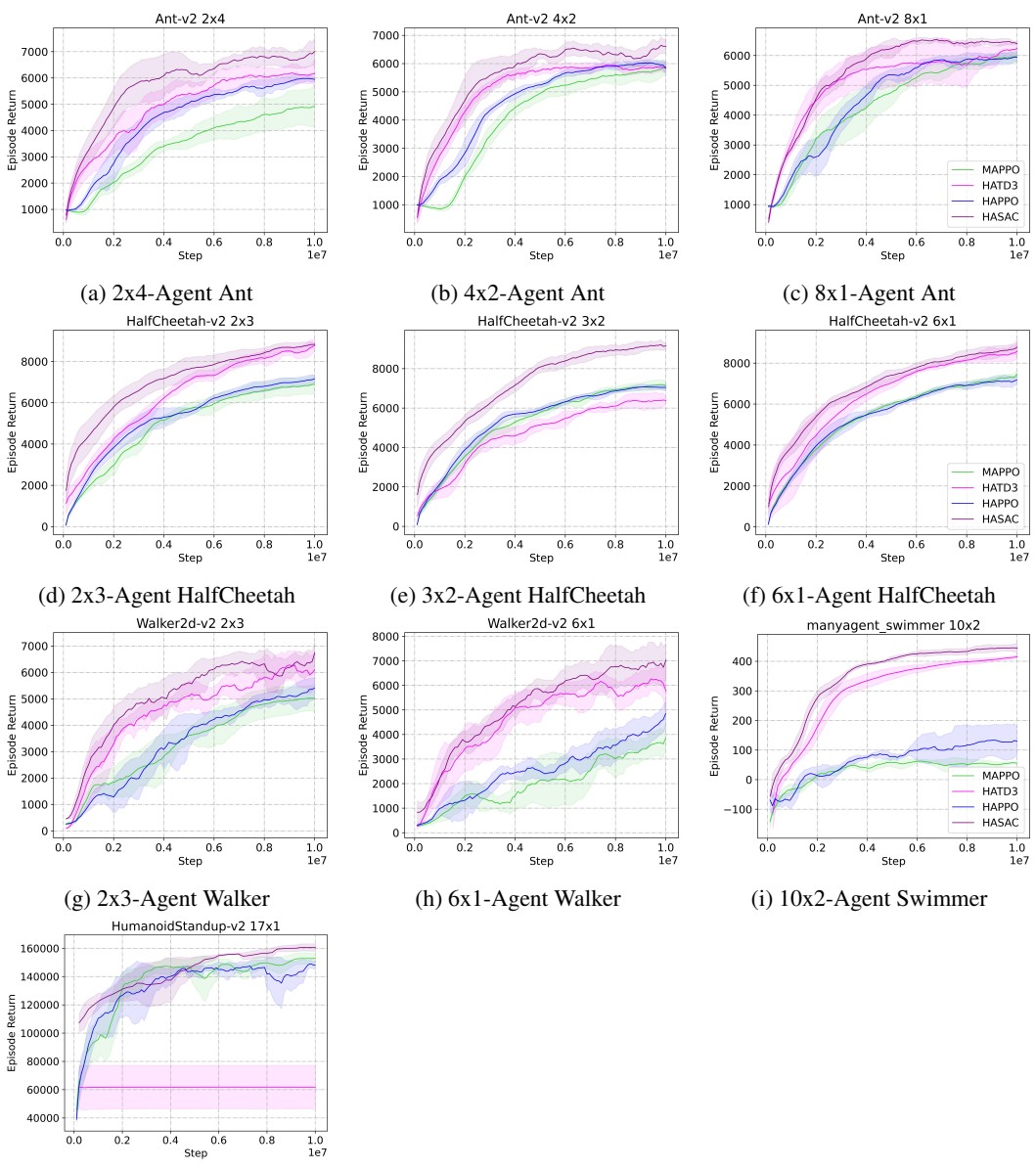

(a) 2x4-Agent Ant     (b) 4x2-Agent Ant     (c) 8x1-Agent Ant

(d) 2x3-Agent HalfCheetah     (e) 3x2-Agent HalfCheetah     (f) 6x1-Agent HalfCheetah

(g) 2x3-Agent Walker     (h) 6x1-Agent Walker     (i) 10x2-Agent Swimmer

(j) 17x1-Agent HumanoidStandup

Figure 8: Comparisons of average episode return on multiple Multi-Agent MuJoCo tasks.

MPE with some minor adjustments and allows for customizing the number of agents and landmarks, as well as the global and local rewards. To adapt these tasks to fully cooperative MARL settings, we sum up the individual rewards of agents to form a joint reward. The results presented in Figure 11 shows that HASAC consistently outperforms the baselines in terms of both average return and sample efficiency.

### H.2.6 LIGHT AIRCRAFT GAME

In addition to the previous three well-established benchmarks, we extend our experiments to include a novel environment called Light Aircraft Game (LAG) (23). LAG is a recently developed cooperative-competitive environment for red and blue aircraft games, offering various settings such as single control, 1v1, and 2v2 scenarios. In the context of multi-agent scenarios, LAG currently supports self-play only for 2v2 settings. To address this limitation, we introduce a novel cooperative non-weapon task where two agents collaborate to combat two opponents controlled by the built-in AI. Specifically, the agents are trained to fly towards the tail of their opponents and maintain a suitable distance.

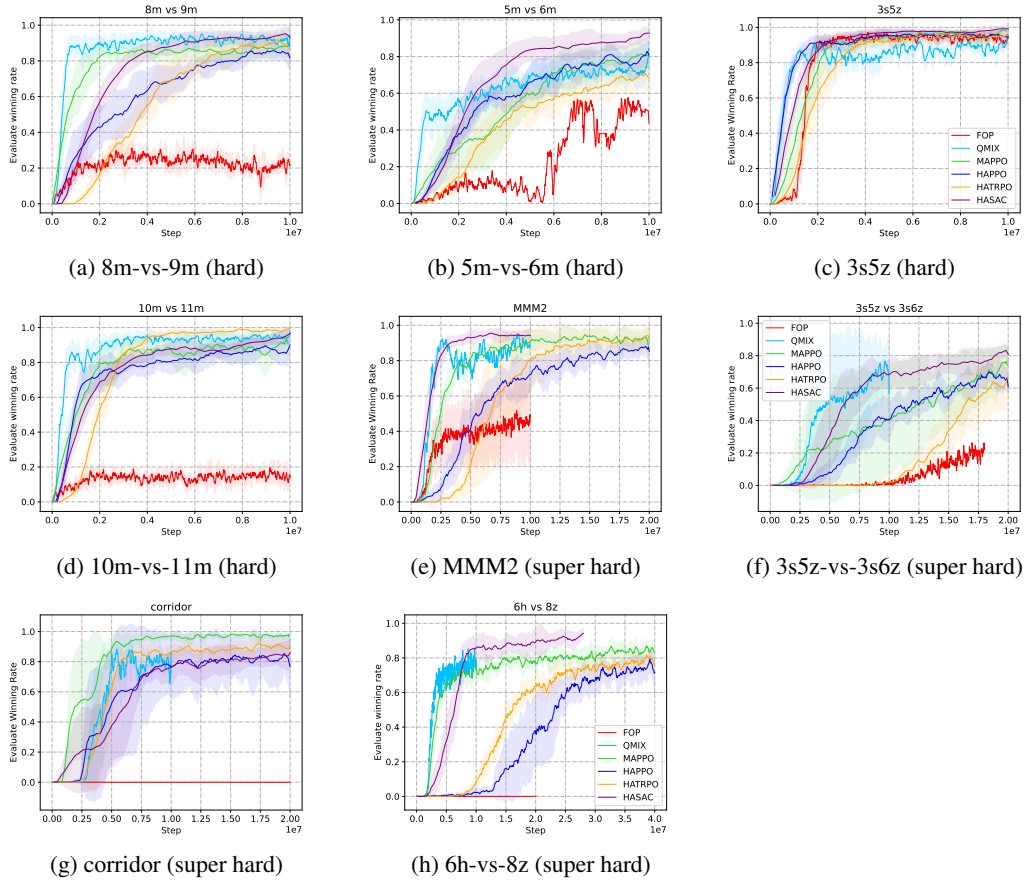

Figure 9: Performance comparison on eight SMAC tasks. We generally observe that HASAC consistently outperforms other baselines in most tasks, exhibiting higher convergence speed and better stability.

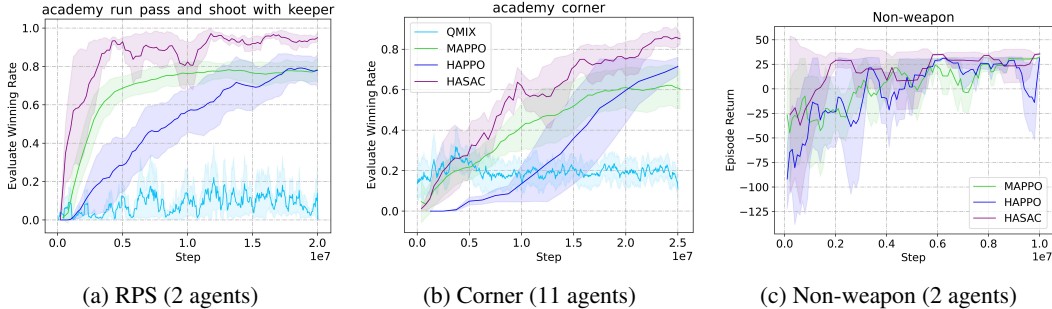

Figure 10: Performance comparison on two GRF tasks and one LAG task. HASAC achieves superior performance to the other methods.

We compare our method to MAPPO and HAPPO on the cooperative non-weapon task involving 2 agents. Figure 10c demonstrates that HASAC outperforms both MAPPO and HAPPO in terms of learning speed and stability. Specifically, HASAC exhibits faster convergence and achieves a higher level of stability throughout the learning process. In contrast, MAPPO and HAPPO exhibit considerable variability in their performance and display slower learning speeds.

## H.3 HYPER-PARAMETER SETTINGS FOR EXPERIMENTS

Before presenting the hyperparameters employed in our experiments, we would like to clarify the reporting conventions we adhere to. First, we introduce a boolean variable **auto_alpha** to indicate whether the temperature is auto-tuned or not. Also, the hyperparameters will only take effect when

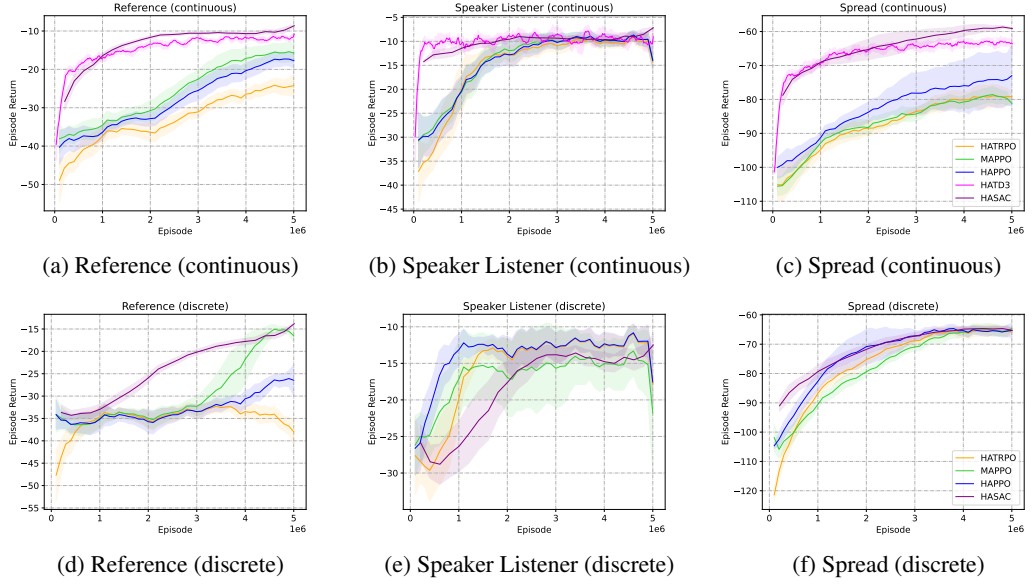

Figure 11: Comparisons of average episode return on six MPE tasks.

they are used. For instance, the temperature parameter $\alpha$ can be assigned any numerical value, but it is taken into consideration only when the boolean value **auto_alpha** is set to *False*. Similarly, the **target_entropy** and **alpha_lr** are applicable when **auto_alpha** is set to *True*.

### H.3.1 COMMON HYPER-PARAMETERS ACROSS ALL ENVIRONMENTS

We implement the HASAC based on the HARL framework (45) and employ the existing implementations of other algorithms, including HATD3, HAPPO, HATRPO, and MAPPO, as described in the HARL literature. In Google Research Football, we use the results of QMIX in Yu et al. (42). In StarCraft Multi-Agent Challenge, we use the results of FOP in Zhang et al. (44). To ensure comprehensive evaluation, we conduct training using a minimum of four different random seeds for each algorithm. Next, we offer the hyperparameters used for HASAC in Table 4 across all environments, which are kept comparable with the HATD3 for fairness purposes.

Table 4: Common hyperparameters used for off-policy algorithms HASAC and HATD3 across all environments.

| hyperparameters | value | hyperparamerters | value |
|---|---|---|---|
| proper time limits | True | warmup steps | 1e4 |
| activation | ReLU | final activation | Tanh |
| buffer size | 1e6 | polyak | 0.005 |
| hidden sizes | [256, 256] | update per train | 1 |
| train interval | 50 | target entropy | $-\dim(\mathcal{A}^i)$ |
| policy noise | 0.2 | noise clip | 0.5 |
| policy update frequency | 2 | linear lr decay | False |

### H.3.2 BI-DEXHANDS

In the Bi-DexHands domain, for SAC, PPO, MAPPO baselines we adopt the implementation and tuned hyperparameters reported in the paper (43). And for HAPPO we adopt the implementation and tuned hyperparameters reported in the HARL paper (45). Here we report the hyperparameters for HASAC and HATD3 in Table 5 and 6.

### H.3.3 MULTI-AGENT MUJOCO (MAMUJOCO)

In this part, we report the hyperparameters used in MAMuJoCo tasks for HASAC and HATD3 in Table 7, 8, 9, and 10. For the other three baselines, we utilize the implementation and tuned hyperparameters reported in the HARL paper (45).

Table 5: Common hyperparameters used for HASAC and HATD3 in the Bi-DexHands domain.

| hyperparameters | value | hyperparameters | value | hyperparameters | value |
|---|---|---|---|---|---|
| rollout threads | 20 | gamma | 0.95 | batch size | 1000 |
| actor lr | 5e - 4 | critic lr | 5e - 4 | n step | 20 |
| use huber loss | False | use valuenorm | False | exploration noise | 0.1 |

Table 6: Different hyperparameters used for HASAC in the Bi-DexHands domain.

| scenarios | auto alpha | alpha | alpha lr |
|---|---|---|---|
| ShadowHandCatchAbreast | True | / | 3e - 4 |
| ShadowHandTwoCatchUnderarm | False | 5e - 5 | / |
| ShadowHandLiftUnderarm | True | / | 3e - 4 |
| ShadowHandOver | True | / | 3e - 4 |
| ShadowHandCatchOver2Underarm | True | / | 3e - 4 |
| ShadowHandPen | True | / | 3e - 4 |
| ShadowHandDoorCloseInward | True | / | 3e - 4 |
| ShadowHandDoorOpenInward | True | / | 3e - 4 |
| ShadowHandDoorOpenOutward | True | / | 3e - 4 |

Table 7: Common hyperparameters used for HASAC and HATD3 in the MAMuJoCo domain.

| hyperparameters | value | hyperparameters | value |
|---|---|---|---|
| rollout threads | 10 | train interval | 50 |
| critic lr | 1e - 3 | gamma | 0.99 |
| use huber loss | False | use valuenorm | False |

Table 8: Common hyperparameters used for HATD3 in the MAMuJoCo domain.

| hyperparameters | value | hyperparameters | value |
|---|---|---|---|
| actor lr | 5e - 4 | exploration noise | 0.1 |

Table 9: Parameter **n_step** used for HASAC and HATD3 in the MAMuJoCo domain.

| scenarios | value | scenarios | value |
|---|---|---|---|
| Ant 2x4 | 5 | HalfCheetah 2x3 | 10 |
| Ant 4x2 | 5 | HalfCheetah 3x2 | 10 |
| Ant 8x1 | 5 | HalfCheetah 6x1 | 10 |
| Walker 2x3 | 20 | Walker 6x1 | 20 |
| manyagent_swimmer 10x2 | 10 | HumanoidStandup 17x1 | 10 |

Table 10: Different hyperparameters used for HASAC in the MAMuJoCo domain.

| scenarios | actor lr | auto alpha | alpha | alpha lr | batch size |
|---|---|---|---|---|---|
| Ant 2x4 | 3e - 4 | False | 0.2 | / | 2200 |
| Ant 4x2 | 3e - 4 | False | 0.2 | / | 1000 |
| Ant 8x1 | 3e - 4 | False | 0.2 | / | 2200 |
| HalfCheetah 2x3 | 1e - 3 | True | / | 3e - 4 | 1000 |
| HalfCheetah 3x2 | 1e - 3 | True | / | 3e - 4 | 1000 |
| HalfCheetah 6x1 | 1e - 3 | True | / | 3e - 4 | 1000 |
| Walker 2x3 | 5e - 4 | False | 0.2 | / | 1000 |
| Walker 6x1 | 5e - 4 | False | 0.2 | / | 1000 |
| manyagent_swimmer 10x2 | 1e - 3 | True | / | 3e - 4 | 1000 |
| HumanoidStandup 17x1 | 1e - 3 | True | / | 3e - 4 | 1000 |

### H.3.4   STARCRAFT MULTI-AGENT CHALLENGE (SMAC)

In the SMAC domain, for MAPPO we adopt the implementation and tuned hyperparameters reported in the MAPPO paper (42). And for HAPPO and HATRPO we adopt the implementation and tuned

Table 11: Common hyperparameters used for HAPPO and MAPPO in the MAMuJoCo domain.

| hyperparameters | value | hyperparameters | value |
|---|---|---|---|
| batch size | 4000 | network | MLP |
| gamma | 0.99 | hidden sizes | [256, 256] |

Table 12: Different hyperparameters used for HAPPO and MAPPO in the MAMuJoCo domain.

| scenarios | linear lr decay | actor/critic lr | ppo/critic epoch | clip param | actor/critic mini batch | entropy coef |
|---|---|---|---|---|---|---|
| Ant | False | 5e - 4 | 5 | 0.1 | 1 | 0 |
| HalfCheetah | False | 5e - 4 | 15 | 0.05 | 1 | 0.01 |
| Walker 2x3 | True | 1e - 3 | 5 | 0.05 | 2 | 0 |
| Walker 6x1 | False | 5e - 4 | 5 | 0.1 | 1 | 0.01 |
| manyagent_swimmer 10x2 | False | 5e - 4 | 5 | 0.2 | 1 | 0.01 |
| HumanoidStandup 17x1 | True | 5e - 4 | 5 | 0.1 | 1 | 0 |

hyperparameters reported in the HARL paper (45). Here we report the hyperparameters for HASAC in Table 13 and 14.

Table 13: Common hyperparameters used for HASAC in the SMAC domain.

| hyperparameters | value | hyperparameters | value | hyperparameters | value |
|---|---|---|---|---|---|
| rollout threads | 20 | state type | FP | batch size | 1000 |
| use huber loss | False | use valuenorm | False | use policy active masks | true |

Table 14: Different hyperparameters used for HASAC in the SMAC domain.

| Map | critic lr | actor lr | gamma | auto alpha | alpha | alpha lr | n step |
|---|---|---|---|---|---|---|---|
| 8m_vs_9m | 5e - 4 | 3e - 4 | 0.95 | True | / | 3e - 4 | 5 |
| 5m_vs_6m | 5e - 4 | 3e - 4 | 0.95 | True | / | 3e - 4 | 20 |
| 3s5z | 5e - 4 | 3e - 4 | 0.99 | True | / | 3e - 4 | 20 |
| 10m_vs_11m | 5e - 4 | 3e - 4 | 0.95 | True | / | 3e - 4 | 20 |
| MMM2 | 5e - 4 | 3e - 4 | 0.95 | True | / | 3e - 4 | 20 |
| 3s5z_vs_3s6z | 5e - 4 | 3e - 4 | 0.99 | True | / | 3e - 4 | 10 |
| corridor | 5e - 4 | 5e - 4 | 0.99 | False | 2e - 3 | / | 5 |
| 6h_vs_8z | 5e - 4 | 3e - 4 | 0.99 | True | / | 3e - 4 | 5 |

H.3.5 GOOGLE RESEARCH FOOTBALL (GRF)

In the GRF domain, for MAPPO and QMIX baselines we adopt the implementation and tuned hyperparameters reported in the MAPPO paper (42). And for HAPPO we adopt the implementation and tuned hyperparameters reported in the HARL paper (45). Here we report the hyperparameters for HASAC in Table 15 and 16.

Table 15: Common hyperparameters used for HASAC in the GRF domain.

| hyperparameters | value | hyperparameters | value | hyperparameters | value |
|---|---|---|---|---|---|
| rollout threads | 20 | gamma | 0.99 | batch size | 1000 |
| actor lr | 5e - 4 | critic lr | 5e - 4 | n step | 10 |
| use huber loss | False | use valuenorm | False | | |

Table 16: Different hyperparameters used for HASAC in the GRF domain.

| scenarios | auto alpha | alpha | alpha lr |
|---|---|---|---|
| RPS | False | 1e - 4 | / |
| Corner | False | 1e - 3 | / |

### H.3.6   MULTI-AGENT PARTICLE ENVIRONMENT (MPE)

In this part, we report the hyperparameters for HASAC in Table 17.

Table 17: Hyperparameters used for HASAC in the MPE domain.

| hyperparameters | value | hyperparameters | value | hyperparameters | value |
|---|---|---|---|---|---|
| rollout threads | 20 | batch_size | 1000 | critic lr | 5e - 4 |
| gamma | 0.99 | actor lr | 5e - 4 | n_step | 20 |
| auto alpha | True | alpha lr | 3e - 4 | | |
| use huber loss | False | use valuenorm | False | | |

### H.3.7   LIGHT AIRCRAFT GAME (LAG)

In this part, we report the hyperparameters for HASAC in Table 18 and the hyperparameters for MAPPO and HAPPO in Table 19.

Table 18: Hyperparameters used for HASAC in the LAG domain.

| hyperparameters | value | hyperparameters | value | hyperparameters | value |
|---|---|---|---|---|---|
| rollout threads | 20 | batch_size | 1000 | critic lr | 5e - 4 |
| gamma | 0.99 | actor lr | 5e - 4 | n_step | 10 |
| auto alpha | True | alpha lr | 3e - 4 | | |

Table 19: Hyperparameters used for MAPPO and HAPPO in the LAG domain.

| hyperparameters | value | hyperparameters | value | hyperparameters | value |
|---|---|---|---|---|---|
| batch size | 4000 | linear lr decay | False | hidden sizes | [256, 256] |
| actor/critic lr | 5e - 4 | gamma | 0.99 | network | MLP |
| ppo epoch | 5 | entropy coef | 0 | clip param | 0.05 |
| critic epoch | 5 | actor mini batch | 2 | critic mini batch | 2 |

# I    ABLATION STUDY ON SEQUENTIAL UPDATES

In this section, we conduct an ablation study to investigate the effect of sequential updates. We compare the performance of the original HASAC with sequential update scheme, and HASAC without sequential updates (or MASAC), which updates each policy according to the following objective:

$$J_{\pi^i}(\phi^i) = \mathbb{E}_{s_t \sim \mathcal{D}} \left[ \mathbb{E}_{a_t^i \sim \pi_{\phi^i}^i, \mathbf{a}_t^{-i} \sim \boldsymbol{\pi}_{\phi_{\text{old}}^{-i}}^{-i}} \left[ \alpha \log \pi_{\phi^i}^i \left( a_t^i | s_t \right) - Q_{\boldsymbol{\pi}_{\text{old}};\theta}^i \left( s_t, a_t^i, \mathbf{a}_t^{-i} \right) \right] \right].$$

We run the experiments on nine Bi-DexHands tasks. For a fair comparison, we set the hyperparameters of MASAC to be the same as those of HASAC. Experiments results show that HASAC with sequential updates consistently achieves better performance.

| Tasks | HASAC | MASAC |
|---|---|---|
| ShadowHandCatchAbreast | **45.9(4.3)** | 36.4(5.2) |
| ShadowHandTwoCatchUnderarm | **20.6(1.9)** | 17.2(1.1) |
| ShadowHandLiftUnderarm | **361.0(23.2)** | **345.0(9.7)** |
| ShadowHandOver | **31.1(0.7)** | **30.5(1.1)** |
| ShadowHandCatchOver2Underarm | **28.7(0.4)** | 27.0(0.9) |
| ShadowHandPen | **181.5(10.4)** | 163.8(18.6) |
| ShadowHandDoorCloseInward | **429.8(16.7)** | **425.9(7.1)** |
| ShadowHandDoorOpenInward | **475.6(23.8)** | 428.8(14.1) |
| ShadowHandDoorOpenOutward | **542.1(13.1)** | 528.1(20.4) |

Table 20: Averaged final performance on nine Bi-DexHands tasks.

