# OpenReview forum: "Maximum Entropy Heterogeneous-Agent Reinforcement Learning"
_ICLR.cc/2024/Conference — ICLR 2024 spotlight_

### Official Review · Reviewer_9JkX · 2023-10-28

**Soundness:** 3 good
**Presentation:** 4 excellent
**Contribution:** 3 good
**Rating:** 8
**Confidence:** 4

**Summary:**

This work addresses cooperative multiagent reinforcement learning (MARL). It builds on the idea of MaxEnt RL, and proposes the maximum entropy heterogeneous-agent reinforcement learning (MEHARL) framework for learning stochastic policies in MARL. On the theoretical/technical side, it uses the PGM to derive the MaxEnt MARL objective, and prove monotonic improvement and QRE convergence properties for the corresponding HASAC algorithm, as well as the unified MEHAML template. On the empirical side, HASAC has been implemented on six benchmark MARL tasks and it achieves the best performance in 31 out of 35 tasks across all benchmarks.

**Strengths:**

1. Excellent presentation. The methodology is well motivated with an illustrative matrix game example. The related work is well discussed and the contribution of this paper is clear. The paper is overall well structured and easy to follow.

2. Clear technical contribution. The algorithmic framework MEHAML as well as the specific practical algorithm HASAC are novel and theoreticall grouned with proofs on the nonotonicity improvement and convergence to QRE. The method is not a simple combination of MaxEnt RL and MARL, but the derivation of the MARL version is built on the PGM formulation connecting from the idea of control as an inference task.

3. Superior empirical performance over a wide spectrum of benchmark tasks -- HASAC has been implemented on six benchmark MARL tasks and it achieves the best performance in 31 out of 35 tasks across all benchmarks.

**Weaknesses:**

I don't see a major weakness.

A minor point: the IGO assumption is not explained when first being introduced. I don't think people outside MARL are familiar with this term.

**Questions:**

The MEHAML framework (or HASAC) is proved to converge to QRE, but not the optimal NE. But it seems that empirically HASAC does learn a good equilibrium due to the stochastic policies. I am curious if is it possible at all to have some sort of guarantees to reach the optimal NE?

---

> ### Author Response · Authors · 2023-11-16
> **Response to Reviewer 9JkX**
>
> Thank you sincerely for appreciating our work. Your acknowledgment and encouragement have motivated us to work even harder to achieve a thorough refinement of our current work.
>
> Q1: The IGO assumption is not explained.
>
> > A1: Thank you for pointing it out. We have included the definition of IGO and the explanation of its limitations in Appendix A.2.
>
> Q2: HASAC is proved to converge to QRE, but not the optimal NE. Is it possible to have guarantees to reach the optimal NE?
>
> > A2: We agree that in the cooperative setting, there exists an optimal joint policy (i.e., Pareto optimality) to which our proposed methods cannot guarantee convergence. It is because although our setting is cooperative, the agents' policies are decentralized. Consequently, finding a Pareto-optimal policy may prove infeasible, as it could involve joint exploration and joint optimization—an impractical task for agents unaware of other agents' policies. Therefore, our method cannot guarantee convergece to Pareto-optimal policies, and obtaining such policies requires directly learning the joint policy.

---

> > ### Comment · Reviewer_9JkX · 2023-11-23
> >
> > Thanks for the authors' response. I will be happy to see the paper bing accepted.

---

> > > ### Author Response · Authors · 2023-11-23
> > > **Thanks**
> > >
> > > Thank you again for helping us improve the paper. Should you have any further comments, please let us know and we are happy to improve.

---

### Official Review · Reviewer_iakN · 2023-10-28

**Soundness:** 3 good
**Presentation:** 3 good
**Contribution:** 4 excellent
**Rating:** 8
**Confidence:** 3

**Summary:**

This paper proposes a MaxEnt MARL approach that employs sequential policy updates and uses a centralized Q-function. They provide the representation of QRE policies maximizing the MaxEnt objective of MARL and demonstrate that the joint policy updated through the sequential policy updates converges to the joint QRE policy. The proposed practical algorithm is simple and outperforms the baselines on various benchmarks.

**Strengths:**

1.	(Theoretical) With the multi-agent soft Q-value function defined in the main paper, the authors demonstrate that through the sequential policy updates, the joint policy converges to a QRE policy while monotonically improving the objectives for the policy and Q-value function.

2.	(Contribution) The authors provide remarkable combinations for the MaxEnt problem of those that exist in previous works; the monotonic improvement and the convergence to a QRE policy for the MaxEnt objective seem to be rigorous, and they extend the MaxEnt MARL problem to the general one with possible constraints.

3.	(Simplicity) The proposed practical algorithm is straightforward. The algorithm doesn’t require recomputing Q-value estimation for the sequential policy updates.

4.	(Experimental) On a bunch of benchmarks, the proposed algorithm consistently achieves superior performance and high sample efficiency, compared to the baselines.

**Weaknesses:**

1. (Unclear effect of the sequential updates) Although there is an example [1] to show why the sequential policy updates for the standard MARL objective are needed, an example or technical explanation for the MaxEnt MARL objective is also needed. It is because the authors define the multi-agent soft Q-function (eq. (7)) and local policy update (eq. (8)). Also, the practical objective for the policy (eq. (10)) can be reduced to the expectation of $\alpha\log\pi_{\phi_{i_m}}^{i_m}-Q_{\pi_{old}}^{i_{1:n}}$, which is consistent with the pseudocode of HASAC, since the MA soft Q-function consists of the centralized Q-function $Q_{\pi_{old}}^{i_{1:n}}$ and the entropy term of its complementary agents, which is not subject to optimize. So, in the proposed algorithm, the sequential policy update may be just additional sampling actions of some agents.

2. (Sensitive to the entropy temperature) The Ablation study shows that the proposed algorithm may not be robust to the entropy temperature and converge to different policies according to the temperature. In this paper, each domain has a different entropy temperature; one domain has a fixed temperature, and another has an automatic temperature with a fixed target entropy. A more effective method to tune entropy terms is needed, like ADER[2].

[1] Kuba et al., Trust Region Policy Optimisation in Multi-Agent Reinforcement Learning, ICLR 2022.

[2] Kim et al., An Adaptive Entropy-Regularization Framework for Multi-Agent Reinforcement Learning, ICML 2023.

**Questions:**

For the weaknesses, could you provide results of MASAC, which is the HASAC without the sequential policy updates, on the benchmarks and the idea of an adaptive entropy temperature tuning method for better exploration?

---

> ### Author Response · Authors · 2023-11-16
> **Response to Reviewer iakN**
>
> Thank you sincerely for appreciating our work. As for the weaknesses you mentioned, please allow us to address them in the most respectful manner.
>
> Q1: The effect of the sequential updates is unclear. Could the authors provide results of MASAC on the benchmarks?
>
> > A1: Thank you for the valuable suggestion. The advantages of sequential updates compared to simultaneous updates in MaxEnt MARL are similar to those mentioned in standard MARL in [1]. MASAC cannot provide a rigorous guarantee of monotonic improvement during training. To illustrate this, let's consider two agents $(1, 2)$ that take actions $a^1, a^2 \in \\{-2, -1, 1, 2\\}$. They receive the reward $r(a^1,a^2) = a^1 \cdot a^2$. Every agent $i$ takes its action $a^i$ from its policy $\pi^i$. We initialize $\pi^1 = (0, 1, 0, 0)$ and $\pi^2 = (0, 0, 1, 0)$. Under this setting, the initial reward is $r(-1, 1) = -1 \cdot 1 = -1$. The maximizing action for agent $1$ is $a_\text{max}^1 = 2$, and for agent $2$ it is $a_{\max}^{2}=-2$. Consequently, MASAC will simultaneously update $\pi^1$ and $\pi^2$, where  $\pi^1$ will assign a highest probability to action 2, and $\pi^2$ will assign a highest probability to action $-2$ (the specific probabilities are influenced by $\alpha$), resulting in a worse outcome. In contrast, sequential updates could avoid such situation, ensuring monotonic improvement.
>
> > Following your suggestion, we have conducted an additional ablation study to investigate the effect of sequential updates in the updated submission version. We compare performance of  the original HASAC and HASAC without sequential updates (or MASAC) on Bi-DexHands tasks. As shown in Appendix I, experimental results show that HASAC consistently achieves better performance than MASAC.
>
>
> Q2: Could the authors implement an adaptive entropy temperature tuning method for better exploration?
>
> > A2: We respectfully disagree with the statement that our automatically tuning temperature method is not effective. The automatically adjusting temperature mechanism we implement for HASAC draws on the method proposed in [2], as stated in Appendix H.1. It is similar to the Equation 13 in ADER [3]. However, regarding the method mentioned in ADER for learning individual target entropies, since we only have a centralized Q function, unlike ADER, which has both a centralized value function and local value functions, we cannot directly apply their approach. Nevertheless, we believe that their idea is indeed enlightening and aids in better temperature adjustment. We will explore how to adjust temperature more effectively in future work. Besides this, our method is already fairly effective in practice. As shown in Appendix H.3, HASAC performs well with automatically adjusting $\alpha$ in most tasks; only a few tasks require additional manual tuning of $\alpha$. As shown in the Appendix C2 of ADER, ADER needs to set different values of $k_{\text{ratio}}, \xi, \alpha^i_\text{init}$ for almost every task on SMAC. In contrast, HASAC achieves good performance across various tasks with auto-tuned  $\alpha$, except for setting $\alpha$ to 0.002 on the 'corridor' task.
> >
>
> [1] Kuba et al., Trust Region Policy Optimisation in Multi-Agent Reinforcement Learning.
>
> [2] Haarnoja et al., Soft actor-critic algorithms and applications.
>
> [3] Kim et al., An Adaptive Entropy-Regularization Framework for Multi-Agent Reinforcement Learning.

---

> > ### Comment · Reviewer_iakN · 2023-11-17
> > **Response to the Rebuttal**
> >
> > I'd like to thank the authors for addressing my concerns and for conducting additional experiments. My concerns have been addressed with additional materials and your comments. I raised the overall rating accordingly.

---

> > > ### Author Response · Authors · 2023-11-17
> > > **Thanks**
> > >
> > > Thank you sincerely for your deep engagement with our work, especially the directions you have suggested that will surely improve the quality of our paper.

---

### Official Review · Reviewer_ZAWK · 2023-10-30

**Soundness:** 3 good
**Presentation:** 3 good
**Contribution:** 2 fair
**Rating:** 6
**Confidence:** 5

**Summary:**

The authors propose a novel algorithm, Heterogeneous-Agent Soft Actor-Critic (HASAC), based on the Maximum Entropy (MaxEnt) framework. The paper theoretically proves the monotonic improvement and convergence properties of HASAC to a quantal response equilibrium (QRE). The authors also introduce a generalized template, Maximum Entropy Heterogeneous-Agent Mirror Learning (MEHAML), which provides any induced method with the same guarantees as HASAC. The proposed methods are evaluated on six benchmarks, demonstrating superior performance in terms of sample efficiency, robustness, and exploration.

**Strengths:**

1. The paper is well-structured and clearly written, making it easy to follow the authors' line of thought.
2. The authors provide a comprehensive theoretical analysis of the proposed methods, including proofs of monotonic improvement and convergence to QRE.
3. The proposed methods are evaluated on a variety of benchmarks, demonstrating their versatility and effectiveness.

**Weaknesses:**

1. The novelty of the paper is limited, as the main contribution is the application of the Soft Actor-Critic (SAC) algorithm to the multi-agent setting.
2. The authors should have tested their method in scenarios where sample efficiency is crucial (such as real robots, stock exchange, etc), given that their proposed method is off-policy.
3. The validity of the experimental results is questionable. The training curves show significant fluctuations, and the authors only present a selection of results in the main paper, which may give a biased view of the method's performance.
4. The authors should provide more comprehensive experimental results, including results from a larger number of seeds, to fully demonstrate the effectiveness of their method.

**Questions:**

1. Could the authors provide more details on how sensitive is the performance of HASAC to the choice of α?
2. How does the proposed method perform in scenarios where sample efficiency is crucial? Could the authors provide experimental results in such scenarios?
3. Could the authors provide more comprehensive experimental results, including results from a larger number of seeds ?

---

> ### Author Response · Authors · 2023-11-16
> **Response to Reviewer ZAWK**
>
> Thank you so much for your time and effort in reading our code and paper. We really appreciate your deep engagement that will surely turn our paper into a better shape.
>
> Q1: The novelty and contribution are limited.
>
> > A1: We respectfully disagree with the statement that the novelty and contribution of our work are limited. Firstly, our novelties are three-fold: We first use PGM to derive MaxEnt objective of MARL and provide a representation of QRE in the context of MaxEnt MARL. Then we develop the theoretically-justified HASAC algorithm to maximize this objective. Finally, we introduced a versatile template, MEHAML, for rigorous algorithmic design.
>
> > Secondly, our application of SAC algorithm to MARL is non-trivial. We not only prove rigorous theoretical properties but also demonstrate strong experimental results.
>
> > Lastly, ****Reviewer 9JkX**** also agrees with us on the novelty and theoretical soundness, and believes that our work is not a simple combination of MaxEnt RL and MARL.
>
>
> Q2: Could the authors provide more details about the sensitivity of HASAC to $\alpha$?
>
> > A2: Thank you for raising the question. Actually, HASAC is indeed sensitive to $\alpha$, as shown in Section 5.2. Since $\alpha$ affects the stochasticity of each policy, it leads to the convergence to different QREs. Figure 5 and 4(c) intuitively illustrate the impact of different $\alpha$ on the final QREs and performance. Therefore, setting different $\alpha$ for various tasks is necessary to enable policies to discover and converge to higher reward equilibria. To mitigate such sensitivity to $\alpha$, we draw on the method of auto-tuned $\alpha$ mentioned in [1] and have implemented it for HASAC. As shown in our Appendix H, auto-tuned $\alpha$ does achieve this goal. In most tasks, HASAC performs well with automatically adjusting $\alpha$; only a few tasks require additional manual tuning of $\alpha$. This significantly alleviates sensitivity of HASAC to $\alpha$.
>
> Q3: How about the performance of HASAC in scenarios where sample efficiency is important? Can the authors provide experimental results in such settings?
>
> > A3: We regret that we may not currently have the capability to test our algorithm on real robots. However, Bi-DexHands [2] and MAMuJoCo [3] are both simulated environments for robotic control where sample efficiency is crucial. As shown in our experimental results, on-policy algorithms exhibit low sample efficiency in these environments, especially in Bi-DexHands, where on-policy algorithms fail to learn any meaningful joint policy within 1e7 steps. In contrast, HASAC shows high sample efficiency in these tasks, demonstrating sample-efficient even compared to other off-policy algorithms. Therefore, we believe that our current experiments could showcase the sampling efficiency of HASAC.
>
> Q4: The validity of the experimental results is questionable. The authors only present a selection of results in the main paper.
>
> > A4: The reason why we only present a selection of results in the main paper is the 9-page limit and the abundance of experimental results. Therefore, we have to selectively include a portion of our extensive experiments in the main paper. Our selection is primarily based on the difficulty of tasks, choosing one or two difficult tasks from each benchmark that other baselines cannot solve effectively, while our algorithm still perform well. This allows for a better demonstration of the advantages of our algorithm. Despite this, we recommend you referring to Appendix H for a comprehensive view of all experimental results. In 31 out of the 35 tasks across all benchmarks, our method achieve the best performance. Furthermore, our method demonstrate better stability, smaller fluctuation and variance in most tasks.
>
> Q5: Could the authors provide more experimental results from a larger number of seeds?
>
> > A5: Thank you for the precious suggestion. We initially ran each algorithm with at least four seeds. Due to the large number of experiments and algorithms, and the fact that each method's performance difference across the four random seeds were not substantial, we did not run additional random seeds.
>
> > Following your suggestion, we rerun each algorithm with eight random seeds on three Bi-DexHands tasks, namely ShadowHandCatchAbreast, ShadowHandTwoCatchUnderarm, and ShadowHandLiftUnderarm. We have updated the experimental results in the updated submission version. From the results of these three tasks, the difference between the results from eight seeds and four seeds is minimal, indicating that our original experiments are sufficiently comprehensive to statistically demonstrate the advantages of our algorithm.
>
>
> [1] Haarnoja et al., Soft actor-critic algorithms and applications.
>
> [2] Chen et al., Towards Human-Level Bimanual Dexterous Manipulation with Reinforcement Learning.
>
> [3] Peng et al., FACMAC: Factored Multi-Agent Centralised Policy Gradients.

---

### Official Review · Reviewer_YGsv · 2023-11-06

**Soundness:** 3 good
**Presentation:** 3 good
**Contribution:** 3 good
**Rating:** 8
**Confidence:** 2

**Summary:**

This paper considers the problem of co-operative Multi-Agent Reinforcement Learning, where issues of sample complexity, training instability, and sub-optimal exploration affect leading methods. The authors propose a method for learning stochastic policies to overcome these limitations, by drawing a connection with Graphical models and deriving a familiar Maximum Entropy solution optimization approach.

The paper is well written, seems comprehensive in it's theoretical establishment of the new method, and thorough in the range of depth of empirical evaluations.

I am familiar with single-agent RL (and have a background in Inverse Reinforcement Learning theory), however am only tangentially aware of work in the multi-agent RL setting. As such, I may have overlooked details or not been aware of relevant prior work when reviewing this paper. I have read the paper, and skimmed the appendices, however did not do a detailed check of the proofs.

**Strengths:**

* Well written, easy to follow the argument development. Seems to engage thoroughly with prior work.
 * Empirical evaluations are strong, and results support the conclusions

**Weaknesses:**

* The contribution of the method in part hinges on the limitations induced by the 'IGO' assumption from prior work (Sec 2, p2), but this is never elaborated on in the paper. Can you define IGO more clearly and explain exactly what limitations this assumption introduces? This will help the reader not intimately familiar with MARL.
* The proposed methods introduces hyper-parameters, notably the temperature term $\alpha$ and the drift functional and neighborhood operator. However any alternate method will also have hyper-parameters, so this isn't a big drawback. Some elaboration of the 'automatically adjusted' $\alpha$ schedule (citation #9) just before the heading for Sec. 6 might be helpful for the reader here.

**Questions:**

# Questions and comments

* It seems the design of the drift functional and neighborhood operators will be key to the success of the proposed HASAC, or MEHAML based methods (as you note in Sec. 6). Can you provide any comment on what factors should be taken into consideration in the construction of these terms? E.g. in what ways will this depend on the nature or definition of the MARL task? Some discussion of the design/selection of these terms for your empirical experiments might be helpful here.
 * The core method (e.g. end of Sec.4 on p7) seems to have high-level similarities to PPO methods for single-agent RL (e.g. constraint to keep the policies from drifting too far) - do you see any connection to this family of methods? Is this something that could be explored further in the literature or has been already?
 * The notion of Quantal Response Equilibrium is key to this optimization objective, but I'm not familiar with this term. You provide a citation (#20, also #6), but the paper would be strengthened with a little bit more explanation of this notion in Sec 4.1. E.g. can you give some intuition for what this objective means in practice compared to regular Nash Equilibrium? In what situations is QRE to be preferred over NE?
 * What is the $\omega$-limit (Point 4 in Theorem 3) - I could not find a definition and am not familiar with this terminology.

# Minor and grammatical points

 * Under heading 5.1 - '2 hundred' - could write as '200'
 * There are a lot of acronyms in this paper - please consider adding a table of acronym definitions in the appendix to aid readers.

---

> ### Author Response · Authors · 2023-11-16
> **Response to Reviewer YGsv**
>
> Thank you very much for your careful review. We deeply appreciate your acknowledgment and the helpful comments you have given. Please allow us to address them as follows.
>
> Q1: IGO assumption should have been defined and explained more clearly.
>
> > A1: Thanks for pointing this out. Due to the 9-page limit, we have included the definition of IGO and the explanation of its limitations in Appendix A.2.
>
> Q2: Lack of elaboration of the automatically-adjusted $\alpha$ schedule.
>
> > A2: Thank you for the helpful suggestion. We implement the automatically-adjusted $\alpha$ for HASAC following the method proposed in [2]. And we have included the optimization objective for $\alpha$ in Appendix H.1.
>
> Q3&Q4: The design of the drift functional and neighborhood operators is key to the MEHAML based methods. Could the authors provide some discussion of the design of these terms? What is the connection between MEHAML and PPO methods?
>
> > A3&A4: Thank you for raising the question. We would like to answer it as follows. First, HASAC can be seen as the most natural instance of MEHAML template with a drift functional of $0$ and neighborhood of $\Pi^{i_m}$.  Nevertheless, HASAC has demonstrated effective performance across a majority of tasks.
>
> > Furthermore, when designing more sophisticated drift functionals and neighborhoods, the most crucial factor is undoubtedly to design operators that satisfy the definition. As discussed in [3], drift functional can be designed as KL-divergence, squared L2 distance, etc., while neighborhood can be designed as KL-ball, like in TRPO. What's more, drift functional can be obtained by meta-learning, as shown in [4]. The objective of our theory is primarily theoretical, aiming to provide theoretical guarantees for all these potential methods in MaxEnt MARL. The specifics of which drift functional and neighborhood lead to better performance in a certain task, as well as how to design practical algorithms with different operators, remain for future research.
>
> > In conclusion, the broad scope of drift functionals and neighborhoods encompasses abstractions for TRPO, PPO, and many other methods. Hence, the MEHAML template offers theoretical guarantees for applying policy updates with additional constraints, including PPO and TRPO, to MaxEnt MARL.
>
>
> Q5: Can the authors explain what QRE means in practice compared to NE? In what situations is QRE more favorable than NE?
>
> > A5: We apologize if the explanation of QRE is not clear enough. In fact, QRE is a generalization of NE. In NE, each player is perfectly rational, deterministically selecting the strategy with highest payoff. In QRE, however, each player is bounded rational, resulting in their strategies being probabilistic distributions related to payoffs. Put simply, QRE represents the equilibria of stochastic behavior, while NE represents the equilibria of deterministic behavior.
>
> > In our case, due to the additional entropy regularization term in the MaxEnt objective, each agent's policy at equilibrium becomes stochastic (Boltzmann distribution) rather than a deterministic one, as demonstrated by Theorem 1. Therefore, it corresponds to QRE. As $\alpha \rightarrow 0$, the MaxEnt objective is reduced to the standard objective, the Boltzmann distribution is reduced to a one-hot distribution, and QRE is reduced to NE. Hence, it is not accurate to claim any inherent advantages of QRE over NE; the observed advantage in experimental results arises from the effectiveness of stochastic policies, which inherently facilitate more efficient exploration, leading to the discovery and convergence of equilibria with higher rewards.
>
>
> Q6: What is the $\omega$-limit set?
>
> > A6: The $\omega$-limit set is a concept used in the study of dynamic systems, which describes the set of states that a system tends to reach as time progresses indefinitely. In our case, the sequence of joint policies would reach fixed points of soft policy iteration, and these fixed points are QRE policies. Therefore, we refer to these fixed points as the $\omega$-limit set of joint policies, and their $\omega$-limits set consists of QRE.
>
> Q7: Some minors.
>
> > Thank you for the valuable suggestion. We have rewritten '2 hundred' as '200' and added a table of acronyms in Appendix A.1.
>
> [1] Wang et al., More centralized training, still decentralized execution: Multi-agent conditional policy factorization.
>
> [2] Haarnoja et al., Soft actor-critic algorithms and applications.
>
> [3] Kuba et al., Mirror learning: A unifying framework of policy optimisation.
>
> [4] Kuba et al., Discovered Policy Optimisation.

---

### Author Response · Authors · 2023-11-17
**Summary of responses and major modifications**

Dear reviewers,

Thanks a lot for your thorough investigation of our paper, and for your valuable comments. We have made proper modifications according to your suggestions. All modifications are marked in red in our revised paper. Here, we summarize the modifications.

- We added a table of acronyms in Appendix A.1.
- We included the missing definition and limitations of IGO assumption in the Appendix A.2.
- We provided the results from eight random seeds on three Bi-DexHands tasks,namely ShadowHandCatchAbreast, ShadowHandTwoCatchUnderarm, and ShadowHandLiftUnderarm.
- We provided the optimization objective for automatically adjusting temperature in Appendix H.1.
- We conducted an additional ablation study to investigate the effect of sequential updates in Appendix I.

We hope that these changes answer the reviewers' concerns, and make our paper more readable and insightful.

Authors

---

### Meta-Review · Area_Chair_F59c · 2023-12-07

**Metareview:**

The paper introduces HASAC, a novel algorithm in the Maximum Entropy (MaxEnt) framework, addressing challenges in cooperative Multi-Agent Reinforcement Learning (MARL). Theoretical proofs validate HASAC's effectiveness, demonstrating monotonic improvement and convergence to quantal response equilibrium (QRE). The paper also presents MEHAML, a generalized template. Empirical evaluations on six benchmarks consistently show HASAC outperforming baselines, emphasizing enhanced sample efficiency, robustness, and exploration. The work contributes a comprehensive solution, bridging theory and practice in cooperative MARL.

The paper stands out for its excellent presentation, well-motivated methodology, and clear discussion of related work. The technical contributions, specifically the novel algorithmic framework MEHAML and the practical algorithm HASAC, are noteworthy. Unlike a simple combination of MaxEnt RL and MARL, the paper's approach is uniquely derived from the Probabilistic Graphical Models (PGM) formulation. Empirically, HASAC consistently outperforms baselines across various benchmarks, achieving the best performance in 31 out of 35 tasks. Overall, the paper contributes significantly to cooperative Multi-Agent Reinforcement Learning (MARL) with its clarity, innovation, and empirical success.

While the paper has several strengths, a few weaknesses have been identified. The identified weaknesses in the paper include concerns about limited novelty due to applying the Soft Actor-Critic (SAC) algorithm, the absence of testing in crucial sample efficiency scenarios, and doubts about the validity of experimental results due to significant fluctuations in training curves and selective result presentation. Additionally, there are calls for a clearer explanation of the effect of sequential updates in the MaxEnt MARL objective and a more robust method to tune entropy terms, given the algorithm's sensitivity to entropy temperature in the ablation study. Addressing these concerns would enhance the overall quality and credibility of the paper.

Given the consensus among reviewers, we propose accepting the paper for publication, with the understanding that the authors will address the identified weaknesses and incorporate any necessary clarifications or improvements in the final revision. We are confident that the suggested changes will further enhance the paper's quality and contribution to the field.

**Justification For Why Not Higher Score:**

While the paper demonstrates notable strengths, there are aspects (like the novelty) that should be stronger in a paper accepted for oral presentation.

**Justification For Why Not Lower Score:**

The paper's substantial theoretical contribution, superior empirical performance, clear presentation, and potential for broader impact collectively make it a strong candidate for a spotlight presentation rather than being limited to a poster.

---

### Decision · Program_Chairs · 2024-01-16

Accept (spotlight)